# Guarantees for Tuning the Step Size using a Learning-to-Learn Approach

## Abstract

Learning-to-learn—using optimization algorithms to learn a new optimizer—has successfully trained efficient optimizers in practice. This approach relies on meta-gradient descent on a meta-objective based on the trajectory that the optimizer generates. However, there were few theoretical guarantees on how to avoid meta-gradient explosion/vanishing problems, or how to train an optimizer with good generalization performance. In this paper we study the learning-to-learn approach on a simple problem of tuning the step size for quadratic loss. Our results show that although there is a way to design the meta-objective so that the meta-gradient remain polynomially bounded, computing the meta-gradient directly using backpropagation leads to numerical issues that look similar to gradient explosion/vanishing problems. We also characterize when it is necessary to compute the meta-objective on a separate validation set instead of the original training set. Finally, we verify our results empirically and show that a similar phenomenon appears even for more complicated learned optimizers parametrized by neural networks.

## 1 Introduction

Choosing the right optimization algorithm and related hyper-parameters is important for training a deep neural network. Recently, a series of works (e.g., Andrychowicz et al. (2016); Wichrowska et al. (2017)) proposed to use learning algorithms to find a better optimizer. These papers use a learning-to-learn approach: they design a class of possible optimizers (often parametrized by a neural network), and then optimize the parameters of the optimizer (later referred to as meta-parameters) to achieve better performance. We refer to the optimization of the optimizer as the meta optimization problem, and the application of the learned optimizer as the inner optimization problem. The learning-to-learn approach solves the meta optimization problem by defining a meta-objective function based on the trajectory that the inner-optimizer generates, and then using back-propagation to compute the meta-gradient (Franceschi et al., 2017).

Although the learning-to-learn approach has shown empirical success, there are very few theoretical guarantees for learned optimizers. In particular, since the optimization for meta-parameters is usually a nonconvex problem, does it have bad local optimal solutions? Current ways of optimizing meta-parameters rely on unrolling the trajectory of the inner-optimizer, which is very expensive and often lead to exploding/vanishing gradient problems. Is there a way to alleviate these problems? Can we have a provable way of designing meta-objective to make sure that the inner optimizers can achieve good generalization performance?

In this paper we answer some of these problems in a simple setting, where we use the learning-to-learn approach to tune the step size of the standard gradient descent/stochastic gradient descent algorithm. We will see that even in this simple setting, many of the challenges still remain and we can get better learned optimizers by choosing the right meta-objective function. Though our results are proved only in the simple setting, we empirically verify the results using complicated learned optimizers with neural network parametrizations.

## 1.1 CHALLENGES OF LEARNING-TO-LEARN APPROACH AND OUR RESULTS

Metz et al. (2019) highlighted several challenges in the meta-optimization for learning-to-learn approach. First, they observed that the optimal parameters for the learned optimizer (or even just the step size for gradient descent) can depend on the number of training steps $t$ of the inner-optimization problem, which is also observed by Wu et al. (2018). Ge et al. (2019) theoretically proved this in a least-squares setting. Because of this, one needs to ensure that the inner training has enough number of steps (similar to the number of steps that it would take when we apply the learned optimizer). However, when the number of steps is large, the meta-gradient can often explode or vanish, which makes it difficult to solve the meta-optimization problem.

Our first result shows that this is still true in the case of tuning step size for gradient descent on a simple quadratic objective. In this setting, we show that there is a unique local and global minimizer for the step size, and we also give a simple way to get rid of the gradient explosion/vanishing problem.

**Theorem 1** (Informal). *For tuning the step size of gradient descent on a quadratic objective, if the meta-objective is the loss of the last iteration, then the meta-gradient can explode/vanish. If the meta-objective is the $\log$ of the loss of the last iteration, then the meta-gradient is polynomially bounded. Further, doing meta-gradient descent with a meta step size of $1/\sqrt{k}$ (where $k$ is the number of meta-gradient steps) provably converges to the optimal step size for the inner-optimizer.*

Surprisingly, even though taking the $\log$ of the objective solves the gradient explosion/vanishing problem, one cannot simply implement such an algorithm using auto-differentiation tools such as those used in TensorFlow (Abadi et al., 2016). The reason is that even though the meta-gradient is polynomially bounded, if we compute the meta-gradient using the standard back-propagation algorithm, the meta-gradient will be the ratio of two exponentially large/small numbers, which causes numerical issues. Detailed discussion for the first result appears in Section 3 (Theorem 3 and Theorem 4).

The generalization performance of the learned optimizer is another challenge. If one just tries to optimize the performance of the learned optimizer on the training set (we refer to this as the train-by-train approach), then the learned optimizer might overfit. Metz et al. (2019) proposed to use a train-by-validation approach instead, where the meta-objective is defined to be the performance of the learned optimizer on a separate validation set.

Our second result considers a simple least squares setting where $y = \langle w^*, x \rangle + \xi$ and $\xi \sim \mathcal{N}(0, \sigma^2)$. We show that when the number of samples is small and the noise is large, it is important to use train-by-validation; while when the number of samples is much larger train-by-train can also learn a good optimizer.

**Theorem 2** (Informal). *For a simple least squares problem in $d$ dimensions, if the number of samples $n$ is a constant fraction of $d$ (e.g., $d/2$), and the samples have large noise, then the train-by-train approach performs much worse than train-by-validation. On the other hand, when number of samples $n$ is large, train-by-train can get close to error $d\sigma^2/n$, which is optimal.*

We discuss the details in Section 4 (Theorem 5 and Theorem 6). In Section 5 we show that such observations also hold empirically for more complicated learned optimizers—an optimizer parametrized by neural network.

## 1.2 RELATED WORK

**Learning-to-learn for supervised learning** Hochreiter et al. (2001) introduced the application of gradient descent method to meta-learning. The idea of using a neural network to parametrize an optimizer started in Andrychowicz et al. (2016), which used an LSTM to directly learn the update rule. Before that, the idea of using optimization to tune parameters for optimzers also appeared in Maclaurin et al. (2015). Later, Li & Malik (2016); Bello et al. (2017) applied techniques from reinforcement learning to learn an optimizer. Wichrowska et al. (2017) used a hierarchical RNN as the optimizer. Metz et al. (2019) adopted a small MLP as the optimizer and used dynamic weighting of two gradient estimators to stabilize and speedup the meta-training process.

**Learning-to-learn in other settings** Ravi & Larochelle (2016) used LSTM as a meta-learner to learn the update rule for training neural networks in the few-shot learning setting, Wang et al. (2016) learned an RL algorithm by another meta-learning RL algorithm, and Duan et al. (2016) learned a general-purpose RNN that can adapt to different RL tasks.

**Gradient-based meta-learning** Finn et al. (2017) proposed Model-Agnostic Meta-Learning (MAML) where they parameterize the update rule for network parameters and learn a shared initialization for the optimizer using the tasks sampled from some distribution. Subsequent works generalized or improved MAML, e.g., Rusu et al. (2018) learned a low-dimensional latent representation for gradient-based meta-learning, and Li et al. (2017) enabled the concurrent learning of learning rate and update direction. Chen et al. (2020) studied a model with an optimization solver stacked on another neural component. They computed Rademacher complexity of the model, but didn't give any optimization guarantee or study train-by-train versus train-by-validation.

**Learning assisted algorithms design** Similar ideas can also be extended to develop a meta-algorithm selecting an algorithm from a family of parametrized algorithms. Gupta & Roughgarden (2017) first modeled the algorithm-selection process as a statistical learning problem and bounded the number of tasks it takes to tune a step size for gradient descent. However, they didn't consider the meta-optimization problem. Based on Gupta & Roughgarden (2017), people have developed and analyzed the meta-algorithms in many problems (Balcan et al., 2016; 2018a;c;b; Denevi et al., 2018; Alabi et al., 2019; Denevi et al., 2019)

**Tuning step size/step size schedule for SGD** Shamir & Zhang (2013) showed that SGD with polynomial step size scheduling can almost match the minimax rate in convex non-smooth settings, which was later tightened by Harvey et al. (2018) for standard step size scheduling. Assuming that the horizon $T$ is known to the algorithm, the information-theoretically optimal bound in convex non-smooth setting was later achieved by Jain et al. (2019) which used another step size schedule, and Ge et al. (2019) showed that exponentially decaying step size scheduling can achieve near optimal rate for least squares regression. There are also a line of works that investigate methods which adapt a vector of step sizes (Sutton, 1992; Schraudolph, 1999; Kearney et al., 2018; Günther et al., 2019; Jacobsen et al., 2019).

## 2 PRELIMINARIES

In this section, we first introduce some notations, then formulate the learning-to-learn framework.

### 2.1 NOTATIONS

For any integer $n$, we use $[n]$ to denote $\{1, 2, \cdots, n\}$. We use $\|\cdot\|$ to denote the $\ell_2$ norm for a vector and the spectral norm for a matrix. We use $\langle \cdot, \cdot \rangle$ to denote the inner product of two vectors. For a symmetric matrix $A \in \mathbb{R}^{d \times d}$, we denote its eigenvalues as $\lambda_1(A) \geq \cdots \geq \lambda_d(A)$. We denote the $d$-dimensional identity matrix as $I_d$. We also denote the identity matrix simply as $I$ when the dimension is clear from the context. We use $O(\cdot), \Omega(\cdot), \Theta(\cdot)$ to hide constant factor dependencies. We use $\text{poly}(\cdot)$ to represent a polynomial on the relevant parameters with constant degree. We say an event happens with high probability if it happens with probability $1 - c$ for small constant $c$.

### 2.2 LEARNING-TO-LEARN FRAMEWORK

We consider the learning-to-learn approach applied to training a distribution of learning tasks. Each task is specified by a tuple $(\mathcal{D}, S_{\text{train}}, S_{\text{valid}}, \ell)$. Here $\mathcal{D}$ is a distribution of samples in $X \times Y$, where $X$ is the domain for the sample and $Y$ is the domain for the label/value. The sets $S_{\text{train}}$ and $S_{\text{valid}}$ are samples generated independently from $\mathcal{D}$, which serve as the training and validation set (the validation set is optional). The learning task looks to find a parameter $w \in W$ that minimizes the loss function $\ell(w, x, y) : W \times X \times Y \to \mathbb{R}$, which gives the loss of the parameter $w$ for sample $(x, y)$. The training loss for this task is $\hat{f}(w) := \frac{1}{|S_{\text{train}}|} \sum_{(x,y) \in S_{\text{train}}} \ell(w, x, y)$, while the population loss is $f(w) := \mathbb{E}_{(x,y) \sim \mathcal{D}}[\ell(w, x, y)]$.

The goal of inner-optimization is to minimize the population loss $f(w)$. For the learned optimizer, we consider it as an update rule $u(\cdot)$ on weight $w$. The update rule is a parameterized function that maps the weight at step $\tau$ and its history to the step $\tau + 1$ : $w_{\tau+1} = u(w_\tau, \nabla \hat{f}(w_\tau), \nabla \hat{f}(w_{\tau-1}), \cdots ; \theta)$. In most parts of this paper, we consider the update rule $u$ as gradient descent mapping with step size as the trainable parameter (here $\theta = \eta$ which is the step size for gradient descent). That is, $u_\eta(w) = w - \eta \nabla \hat{f}(w)$ for gradient descent and $u_\eta(w) = w - \eta \nabla_w \ell(w, x, y)$ for stochastic gradient descent where $(x, y)$ is a sample randomly chosen from the training set $S_{\text{train}}$.

In the outer (meta) level, we consider a distribution $\mathcal{T}$ of tasks. For each task $P \sim \mathcal{T}$, we can define a meta-loss function $\Delta(\theta, P)$. The meta-loss function measures the performance of the optimizer on this learning task. The meta objective, for example, can be chosen as the target training loss $\hat{f}$ at the last iteration (train-by-train), or the loss on the validation set (train-by-validation).

The training loss for the meta-level is the average of the meta-loss across $m$ different specific tasks $P_1, P_2, ..., P_m$, that is, $\hat{F}(\theta) = \frac{1}{m} \sum_{i=1}^m \Delta(\theta, P_k)$. The population loss for the meta-level is the expectation over all the possible specific tasks $F(\theta) = \mathbb{E}_{P \sim \mathcal{T}}[\Delta(\theta, P)]$.

In order to train an optimizer by gradient descent, we need to compute the gradient of meta-objective $\hat{F}$ in terms of meta parameters $\theta$. The meta parameter is updated once after applying the optimizer on the inner objective $t$ times to generate the trajectory $w_0, w_1, ..., w_t$. The meta-gradient is then computed by unrolling the optimization process and back-propagating through the $t$ applications of the optimizer. As we will see later, this unroll procedure is costly and can introduce meta-gradient explosion/vanishing problems.

## 3 ALLEVIATING GRADIENT EXPLOSION/VANISHING PROBLEMS

First we consider the meta-gradient explosion/vanishing problem. More precisely, we say the meta-gradient explodes/vanishes if it is exponentially large/small with respect to the number of steps $t$ of the inner-optimizer.

In this section, we consider a very simple instance of the learning-to-learn approach, where the distribution $\mathcal{T}$ only contains a single task $P$, and the task also just defines a single loss function $f$[1]. Therefore, in this section $\hat{F}(\eta) = F(\eta) = \Delta(\eta, P)$. We will simplify notation and only use $\hat{F}(\eta)$.

The inner task $P$ is a simple quadratic problem, where the starting point is fixed at $w_0$, and the loss function is $f(w) = \frac{1}{2} w^\top H w$ for some fixed positive definite matrix $H$. Without loss of generality, assume $w_0$ has unit $\ell_2$ norm. Suppose the eigenvalue decomposition of $H$ is $\sum_{i=1}^d \lambda_i u_i u_i^\top$. Throughout this section we assume $L = \lambda_1(H)$ and $\alpha = \lambda_d(H)$ are the largest and smallest eigenvalues of $H$ with $L > \alpha$. For each $i \in [d]$, let $c_i$ be $\langle w_0, u_i \rangle$ and let $c_{\min} = \min(|c_1|, |c_d|)$. We assume $c_{\min} > 0$ for simplicity. Note that if $w_0$ is uniformly sampled from the unit sphere, with high probability $c_{\min}$ is at least $\Omega(1/\sqrt{d})$; if $H$ is $XX^\top$ with $X \in \mathbb{R}^{d \times 2d}$ as a random Gaussian matrix, with constant probability, both $\alpha$ and $L - \alpha$ are at least $\Omega(d)$.

Let $\{w_{\tau, \eta}\}$ be the GD sequence running on $f(w)$ starting from $w_0$ with step size $\eta$. We consider several ways of defining meta-objective, including using the loss of the last point directly, or using the log of this value. We first show that although choosing $\hat{F}(\eta) = f(w_{t, \eta})$ does not have any bad local optimal solution, it has the gradient explosion/vanishing problem. We use $\hat{F}'(\eta)$ to denote the derivative of $\hat{F}$ in $\eta$.

**Theorem 3.** *Let the meta objective be $\hat{F}(\eta) = f(w_{t, \eta}) = \frac{1}{2} w_{t, \eta}^\top H w_{t, \eta}$. We know $\hat{F}(\eta)$ is a strictly convex function in $\eta$ with an unique minimizer. However, for any step size $\eta < 2/L$, $|\hat{F}'(\eta)| \leq t \sum_{i=1}^d c_i^2 \lambda_i^2 |1 - \eta \lambda_i|^{2t-1}$; for any step size $\eta > 2/L$, $|\hat{F}'(\eta)| \geq c_1^2 L^2 t (\eta L - 1)^{2t-1} - L^2 t$.*

Note that in Theorem 3, when $\eta < 2/L$, $|\hat{F}'(\eta)|$ is exponentially small because $|1 - \eta \lambda_i| < 1$ for all $i \in [d]$; when $\eta > 2/L$, $|\hat{F}'(\eta)|$ is exponentially large because $\eta L - 1 > 1$. Intuitively, gradient

---

[1] In the notation of Section 2, one can think that $\mathcal{D}$ contains a single point $(0, 0)$ and the loss function $f(w) = \ell(w, 0, 0)$.

explosion/vanishing happens because the meta-loss function becomes too small or too large. A natural idea to fix the problem is to take the $\log$ of the meta-loss function to reduce its range. We show that this indeed works. More precisely, if we choose $\hat{F}(\eta) = \frac{1}{t} \log f(w_{t,\eta})$, then we have

**Theorem 4.** *Let the meta objective be $\hat{F}(\eta) = \frac{1}{t} \log f(w_{t,\eta})$. We know $\hat{F}(\eta)$ has a unique minimizer $\eta^*$ and $\hat{F}'(\eta) = O\left(\frac{L^3}{c_{\min}^2 \alpha(L-\alpha)}\right)$ for all $\eta \geq 0$. Let $\{\eta_k\}$ be the GD sequence running on $\hat{F}$ with meta step size $\mu_k = 1/\sqrt{k}$. Suppose the starting step size $\eta_0 \leq M$. Given any $1/L > \epsilon > 0$, there exists $k' = \frac{M^6}{\epsilon^2} poly(\frac{1}{c_{\min}}, L, \frac{1}{\alpha}, \frac{1}{L-\alpha})$ such that for all $k \geq k'$, $|\eta_k - \eta^*| \leq \epsilon$.*

For convenience, in the above algorithmic result, we reset $\eta$ to zero once $\eta$ goes negative. Note that although we show the gradient is bounded and there is a unique optimizer, the problem of optimizing $\eta$ is still not convex because the meta-gradient is not monotone. We use ideas from quasi-convex optimization to show that meta-gradient descent can find the unique optimal step size for this problem.

Surprisingly, even though we showed that the meta-gradient is bounded, it cannot be effectively computed by doing back-propagation due to numerical issues. More precisely:

**Corollary 1.** *If we choose the meta-objective as $\hat{F}(\eta) = \frac{1}{t} \log f(w_{t,\eta})$, when computing the meta-gradient using back-propagation, there are intermediate results that are exponentially large/small in number of inner-steps $t$.*

Indeed, in Section 5 we empirically verify that standard auto-differentiation tools can still fail in this setting. This suggests that one should be more careful about using standard back-propagation in the learning-to-learn approach. The proofs of the results in this section are deferred into Appendix A.

## 4 TRAIN-BY-TRAIN VS. TRAIN-BY-VALIDATION

Next we consider the generalization ability of simple optimizers. In this section we consider a simple family of least squares problems. Let $\mathcal{T}$ be a distribution of tasks where every task $(\mathcal{D}(w^*), S_{\text{train}}, S_{\text{valid}}, \ell)$ is determined by a parameter $w^* \in \mathbb{R}^d$ which is chosen uniformly at random on the unit sphere. For each individual task, $(x, y) \sim \mathcal{D}(w^*)$ is generated by first choosing $x \sim \mathcal{N}(0, I_d)$ and then computing $y = \langle w^*, x \rangle + \xi$ where $\xi \sim \mathcal{N}(0, \sigma^2)$ with $\sigma \geq 1$. The loss function $\ell(w, x, y)$ is just the squared loss $\ell(w, x, y) = \frac{1}{2}(y - \langle w, x \rangle)^2$. That is, the tasks are just standard least-squares problems with ground-truth equal to $w^*$ and noise level $\sigma^2$.

For the meta-loss function, we consider two different settings. In the train-by-train setting, the training set $S_{\text{train}}$ contains $n$ independent samples, and the meta-loss function is chosen to be the training loss. That is, in each task $P$, we first choose $w^*$ uniformly at random, then generate $(x_1, y_1), ..., (x_n, y_n)$ as the training set $S_{\text{train}}$. The meta-loss function $\Delta_{TbT(n)}(\eta, P)$ is defined to be

$$\Delta_{TbT(n)}(\eta, P) = \frac{1}{2n} \sum_{i=1}^{n} (y_i - \langle w_{t,\eta}, x_i \rangle)^2.$$

Here $w_{t,\eta}$ is the result of running $t$ iterations of gradient descent starting from point $0$ with step size $\eta$. Note we truncate a sequence and declare the meta loss is high once the weight norm exceeds certain threshold. We can safely do this because we assume the ground truth weight $w^*$ has unit norm, so if the weight norm of our model is too high, it means the inner training has diverged and the step size is too large. Specifically, if at the $\tau$-th step, $\|w_{\tau,\eta}\| \geq 40\sigma$, we freeze the training on this task and set $w_{\tau',\eta} = 40\sigma u$ for all $\tau \leq \tau' \leq t$, for some arbitrary vector $u$ with unit norm. Setting the weight to a large vector is just one way to declare the loss is high; we choose this particular way for some proof convenience.

As before, the empirical meta objective in train-by-train setting is the average of the meta-loss across $m$ different specific tasks $P_1, P_2, ..., P_m$, that is,

$$\hat{F}_{TbT(n)}(\eta) = \frac{1}{m} \sum_{k=1}^{m} \Delta_{TbT(n)}(\eta, P_k). \tag{1}$$

In the train-by-validation setting, the specific tasks are generated by sampling $n_1$ training samples and $n_2$ validation samples for each task, and the meta-loss function is chosen to be the validation loss. That is, in each specific task $P$, we first choose $w^*$ uniformly at random, then generate $(x_1, y_1), ..., (x_{n_1}, y_{n_1})$ as the training set $S_{\text{train}}$ and $(x'_1, y'_1), ..., (x'_{n_2}, y'_{n_2})$ as the validation set $S_{\text{valid}}$. The meta-loss function $\Delta_{TbV(n_1,n_2)}(\eta, P)$ is defined to be

$$\Delta_{TbV(n_1,n_2)}(\eta, P) = \frac{1}{2n_2} \sum_{i=1}^{n_2} (y'_i - \langle w_{t,\eta}, x'_i \rangle)^2.$$

Here again $w_{t,\eta}$ is the result of running $t$ iterations of the gradient descent on the training set starting from point 0, and we use the same truncation as before. The empirical meta objective is defined as

$$\hat{F}_{TbV(n_1,n_2)}(\eta) = \frac{1}{m} \sum_{k=1}^{m} \Delta_{TbV(n_1,n_2)}(\eta, P_k), \tag{2}$$

where each $P_k$ is independently sampled according to the described procedure.

We first show that when the number of samples is small (in particular $n < d$) and the noise is a large enough constant, train-by-train can be much worse than train-by-validation, even when $n_1 + n_2 = n$ (the total number of samples used in train-by-validation is the same as train-by-train)

**Theorem 5.** *Let $\hat{F}_{TbT(n)}(\eta)$ and $\hat{F}_{TbV(n_1,n_2)}(\eta)$ be as defined in Equation (1) and Equation (2) respectively. Assume $n, n_1, n_2 \in [d/4, 3d/4]$. Assume noise level $\sigma$ is a large constant $c_1$. Assume unroll length $t \geq c_2$, number of training tasks $m \geq c_3 \log(mt)$ and dimension $d \geq c_4 \log(mt)$ for certain constants $c_2, c_3, c_4$. With high probability in the sampling of training tasks, we have*

$$\eta^*_{train} = \Theta(1) \text{ and } \mathbb{E} \left\| w_{t,\eta^*_{train}} - w^* \right\|^2 = \Omega(1)\sigma^2,$$

*for all $\eta^*_{train} \in \arg\min_{\eta \geq 0} \hat{F}_{TbT(n)}(\eta)$;*

$$\eta^*_{valid} = \Theta(1/t) \text{ and } \mathbb{E} \left\| w_{t,\eta^*_{valid}} - w^* \right\|^2 = \|w^*\|^2 - \Omega(1)$$

*for all $\eta^*_{valid} \in \arg\min_{\eta \geq 0} \hat{F}_{TbV(n_1,n_2)}(\eta)$. In both equations the expectation is taken over new tasks.*

Note that in this case, the number of samples $n$ is smaller than $d$, so the least square problem is under-determined and the optimal training loss would go to 0 (there is always a way to simultaneously satisfy all $n$ equations). This is exactly what train-by-train would do—it will choose a large constant learning rate which guarantees the optimizer converges exponentially to the empirical risk minimizer (ERM). However, when the noise is large making the training loss go to 0 will overfit to the noise and hurt the generalization performance. Train-by-validation on the other hand will choose a smaller learning rate which allows it to leverage the information in the training samples without overfitting to noise. Theorem 5 is proved in Appendix B. We also prove similar results for SGD in Appendix D

We emphasize that neural networks are often over-parameterized, which corresponds to the case when $d > n$. Indeed Liu & Belkin (2018) showed that variants of stochastic gradient descent can converge to the empirical risk minimizer with exponential rate in this case. Therefore in order to train neural networks, it is better to use train-by-validation. On the other hand, we show when the number of samples is large ($n \gg d$), train-by-train can also perform well.

**Theorem 6.** *Let $\hat{F}_{TbT(n)}(\eta)$ be as defined in Equation 1. Assume noise level is a constant $c_1$. Given any $1 > \epsilon > 0$, assume training set size $n \geq \frac{cd}{\epsilon^2} \log(\frac{nm}{\epsilon d})$, unroll length $t \geq c_2 \log(\frac{n}{\epsilon d})$, number of training tasks $m \geq \frac{c_3 n^2}{\epsilon^4 d^2} \log(\frac{tnm}{\epsilon d})$ and dimension $d \geq c_4$ for certain constants $c, c_2, c_3, c_4$. With high probability in the sampling of training tasks, we have*

$$\mathbb{E} \left\| w_{t,\eta^*_{train}} - w^* \right\|^2 \leq (1 + \epsilon) \frac{d\sigma^2}{n},$$

*for all $\eta^*_{train} \in \arg\min_{\eta \geq 0} \hat{F}_{TbT(n)}(\eta)$, where the expectation is taken over new tasks.*

Therefore if the learning-to-learn approach is applied to a traditional optimization problem that is not over-parameterized, train-by-train can work well. In this case, the empirical risk minimizer (ERM) already has good generalization performance, and train-by-train optimizes the convergence towards the ERM. We defer the proof of Theorem 6 into Appendix C.

## 5 EXPERIMENTS

**Optimizing step size for quadratic objective** We first validate the results in Section 3. We fixed a 20-dimensional quadratic objective as the inner problem and vary the number of inner steps $t$ and initial value $\eta_0$. We compute the meta-gradient directly using a formula which we derive in Appendix A. In this way, we avoid the computation of exponentially small/large intermediate terms. We use the algorithm suggested in Theorem 4, except we choose the meta-step size to be $1/(100\sqrt{k})$ as the constants in the theorem were not optimized.

An example training curve of $\eta$ for $t = 80$ and $\eta_0 = 0.1$ is shown in Figure 1, and we can see that $\eta$ converges quickly within 300 steps. Similar convergence also holds for larger $t$ or much larger initial $\eta_0$. In contrast, we also implemented the meta-training with Tensorflow, where the code was adapted from the previous work of Wichrowska et al. (2017). Experiments show that in many settings (especially with large $t$ and large $\eta_0$) the implementation does not converge. In Figure 1, under the TensorFlow implementation, the step size is stuck at the initial value throughout the meta training because the meta gradient explodes and gives NaN value.

In Figure 2, we verify the observation from Metz et al. (2019) that the optimal step size depends on inner training length.

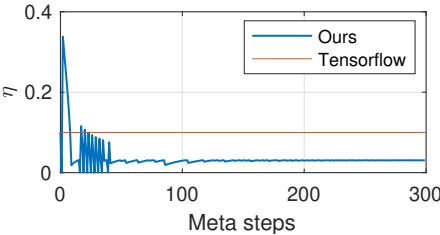

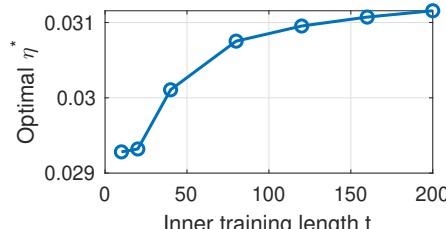

Figure 1: Training $\eta$ ($t = 80$, $\eta_0 = 0.1$)    Figure 2: Optimal $\eta^*$ for different $t$

**Train-by-train vs. train-by-validation, synthetic data** Here we validate our theoretical results in Section 4 using the least-squares model defined there. We fix the input dimension $d$ to be 1000.

In the first experiment, we fix the size of the data ($n = 500$ for train-by-train, $n_1 = n_2 = 250$ for train-by-validation). Under different noise levels, we find the optimal $\eta^*$ by a grid search on its meta-objective for train-by-train and train-by-validation settings respectively. We then use the optimal $\eta^*$ found in each of these two settings to test on 10 new least-squares problem. The mean RMSE, as well as its range over the 10 test cases, are shown in Figure 3. We can see that for all of these cases, the train-by-train model overfits easily, while the train-by-validation model performs much better and does not overfit. Also, when the noise becomes larger, the difference between these two settings becomes more significant.

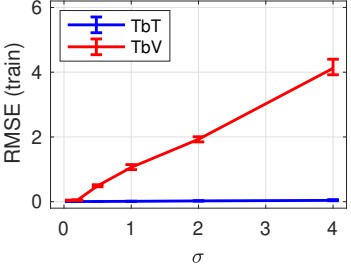

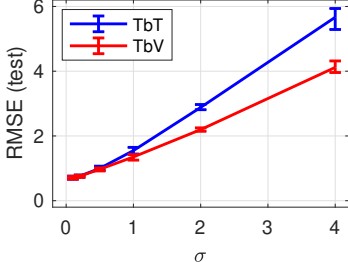

Figure 3: Training and testing RMSE for different $\sigma$ values (500 samples)

In the next experiment, we fix $\sigma = 1$ and change the sample size. For train-by-validation, we always split the samples evenly into training and validation set. From Figure 4, we can see that the gap between these two settings is decreasing as we use more data, as expected by Theorem 6.

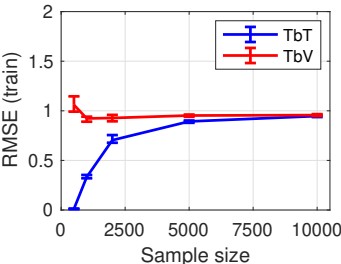 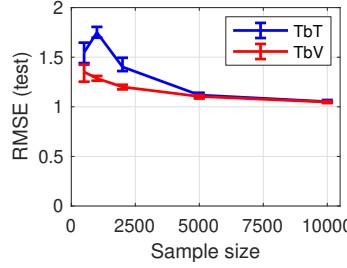

Figure 4: Training and testing RMSE for different samples sizes ($\sigma = 1$)

**Train-by-train vs. train-by-validation, MLP optimizer on MNIST** Finally we consider a more complicated multi-layer perceptron (MLP) optimizer on MNIST data set. We use the same MLP optimizer as in Metz et al. (2019), details of this optimizer is discussed in Appendix F. As the inner problem, we use a two-layer fully-connected network of 100 and 20 hidden units with ReLU activations. The inner objective is the classic 10-class cross entropy loss, and we use mini-batches of 32 samples at inner training. In all the following experiments, we use SGD as a baseline with step size tuned by grid search against validation loss.

To see whether the comparison between train-by-train and train-by-validation behave similarly to our theoretical results, we consider different number of samples and different levels of label noise. For each optimizer, we run 5 independent tests and collect training accuracy and test accuracy for evaluation. The plots show the mean of the 5 tests. We didn't show the measure of the spread because the results of these 5 tests are so close to each other, such that the range or standard deviation marks will not be readable in the plots.

First, consider optimizing the MNIST dataset with small number of samples. In this case, the train-by-train setting uses 1,000 samples (denoted as "TbT1000"), and we use another 1,000 samples as the validation set for the train-by-validation case (denoted as "TbV1000+1000"). To be fair to train-by-train we also consider TbT2000 where the train-by-train algorithm has access to 2000 data points. Figure 5 shows the results—all the models have training accuracy close to 1, but both TbT1000 and TbT2000 overfits the data significantly, whereas TbV1000+1000 performs well.

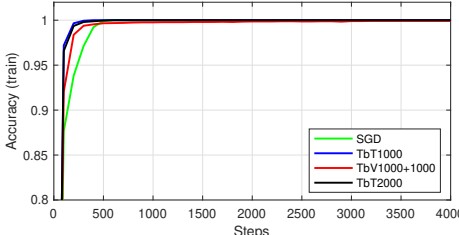 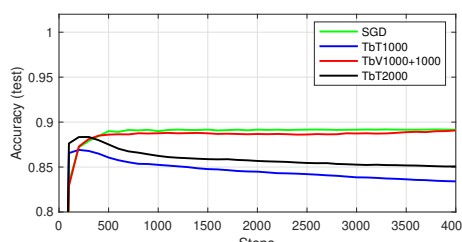

Figure 5: Training and testing accuracy for different models (1000 samples, no noise)

To show that when the noise is higher, the advantage of train-by-validation increases, we keep the same sample size and consider a "noisier" version of MNIST, where we randomly change the label of a sample with probability 0.2 (the new label is chosen uniformly at random, including the original label). The results are shown in Figure 6. We can see that both train-by-train models, as well as SGD, overfit easily with training accuracy close to 1 and their test performances are low. The train-by-validation model performs much better.

Finally we run experiments on the complete MNIST data set (without label noise). For the train-by-validation setting, we split the data set to 50,000 training samples and 10,000 validation samples. As shown in Figure 7, in this case train-by-train and train-by-validation performs similarly (in fact both are slightly weaker than the tuned SGD baseline). This shows that when the sample size is sufficiently large, train-by-train can get comparable results as train-by-validation.

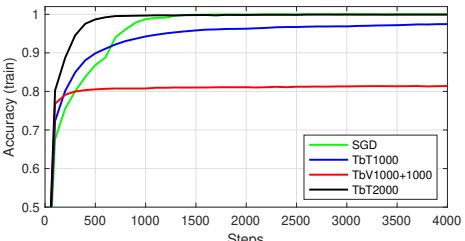 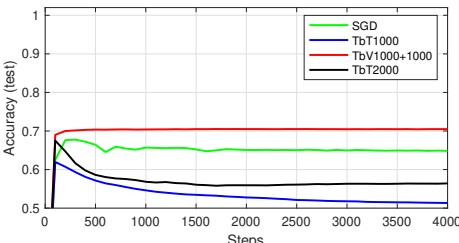

Figure 6: Training and testing accuracy for different models (1000 samples, 20% noise)

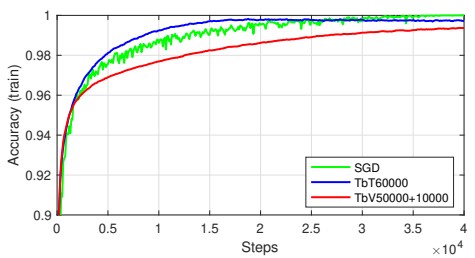 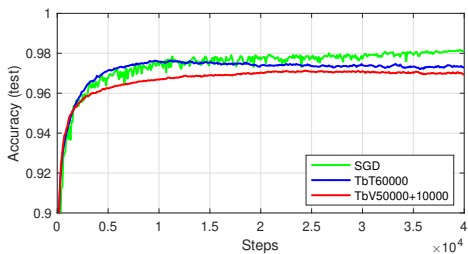

Figure 7: Training and testing accuracy for different models (all samples, no noise)

## 6 CONCLUSION AND FUTURE WORKS

In this paper, we have proved optimization and generalization guarantees for tuning the step size for quadratic loss. From the optimization perspective, we considered a simple task whose objective is a quadratic function. We proved that the meta-gradient can explode/vanish if the meta-objective is simply the loss of the last iteration; we then showed that the log-transformed meta-objective has polynomially bounded meta-gradient and can be successfully optimized. To study the generalization issues, we considered the least squares problem. We showed that when the number of samples is small and the noise is large, train-by-validation approach generalizes better than train-by-train; while when the number of samples is large, train-by-train can also work well. Although our theoretical results are proved for quadratic loss, this simple setting already yields interesting phenomenons and requires non-trivial techniques to analyze. We have also verified our theoretical results on an optimizer parameterized by neural networks and MNIST dataset.

Since this is a very first work studying the learning to learn approach, there are many potential future works. One immediate future work is to extend the result for least squares to log-transformed meta objective (as in Section 3). This is probably doable because compositing the log function with the current meta-objective should not change its minimizer. For the least squares problem, we only studied the generalization properties of the optimal step size under the meta-objective, it's unclear if meta-gradient descent can converge to such an optimal step size. We believe our techniques for the meta-optimization of the simple quadratic objective function (Section 3) can also be useful in this analysis. More broadly, we are also interested in analyzing more complicated optimizers on more complicated tasks.

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

In the appendix, we first give the missing proofs for the theorems in the main paper. Later in Appendix F we give details for the experiments.

**Notations:** Besides the notations defined in Section 2, we define more notations that will be used in the proofs.

For a matrix $X \in \mathbb{R}^{n \times d}$ with $n \leq d$, we denote its singular values as $\sigma_1(X) \geq \cdots \geq \sigma_n(X)$.

For a positive semi-definite matrix $A \in \mathbb{R}^{d \times d}$, we denote $u^\top A u$ as $\|u\|_A^2$. For a matrix $X \in \mathbb{R}^{d \times n}$, let $\text{Proj}_X \in \mathbb{R}^{d \times d}$ be the projection matrix onto the column span of $X$. That means, $\text{Proj}_X = SS^\top$, where the columns of $S$ form an orthonormal basis for the column span of $X$.

For any event $\mathcal{E}$, we use $\mathbb{1}\{\mathcal{E}\}$ to denote its indicator function: $\mathbb{1}\{\mathcal{E}\}$ equals 1 when $\mathcal{E}$ holds and equals 0 otherwise. We use $\bar{\mathcal{E}}$ to denote the complementary event of $\mathcal{E}$.

# A  PROOFS FOR SECTION 3 – ALLEVIATING GRADIENT EXPLOSION/VANISHING PROBLEM FOR QUADRATIC OBJECTIVE

In this section, we prove the results in Section 3. Recall the meta learning problem as follows:

The inner task is a fixed quadratic problem, where the starting point is fixed at $w_0$, and the loss function is $f(w) = \frac{1}{2} w^\top H w$ for some fixed positive definite matrix $H \in \mathbb{R}^{d \times d}$. Suppose the eigenvalue decomposition of $H$ is $\sum_{i=1}^d \lambda_i u_i u_i^\top$. In this section, we assume $L = \lambda_1(H)$ and $\alpha = \lambda_d(H)$ are the largest and smallest eigenvalues of $H$ with $L > \alpha$. We assume the starting point $w_0$ has unit $\ell_2$ norm. For each $i \in [d]$, let $c_i$ be $\langle w_0, u_i \rangle$ and let $c_{\min} = \min(|c_1|, |c_d|)$. We assume $c_{\min} > 0$ for simplicity, which is satisfied if $w_0$ is chosen randomly from the unit sphere.

Let $\{w_{\tau, \eta}\}$ be the GD sequence running on $f(w)$ starting from $w_0$ with step size $\eta$. For the meta-objective, we consider using the loss of the last point directly, or using the $\log$ of this value. In Section A.1, we first show that although choosing $\hat{F}(\eta) = f(w_{t,\eta})$ does not have any bad local optimal solution, it has the gradient explosion/vanishing problem (Theorem 3). Then, in Section A.2, we show choosing $\hat{F}(\eta) = \frac{1}{t} \log f(w_{t,\eta})$ leads to polynomially bounded meta-gradient and further show meta-gradient descent converges to the optimal step size (Theorem 4). Although the meta-gradient is polynomially bounded, if we simply use back-propogation to compute the meta-gradient, the intermediate results can still be exponentially large/small (Corollary 1). This is also proved in Section A.2.

## A.1  META-GRADIENT VANISHING/EXPLOSION

In this section, we show although choosing $\hat{F}(\eta) = f(w_{t,\eta})$ does not have any bad local optimal solution, it has the meta-gradient explosion/vanishing problem. Recall Theorem 3 as follows.

**Theorem 3.** *Let the meta objective be $\hat{F}(\eta) = f(w_{t,\eta}) = \frac{1}{2} w_{t,\eta}^\top H w_{t,\eta}$. We know $\hat{F}(\eta)$ is a strictly convex function in $\eta$ with an unique minimizer. However, for any step size $\eta < 2/L$, $|\hat{F}'(\eta)| \leq t \sum_{i=1}^d c_i^2 \lambda_i^2 |1 - \eta \lambda_i|^{2t-1}$; for any step size $\eta > 2/L$, $|\hat{F}'(\eta)| \geq c_1^2 L^2 t (\eta L - 1)^{2t-1} - L^2 t$.*

Intuitively, if we write $w_{t,\eta}$ in the basis of the eigen-decomposition of $H$, then each coordinate evolve exponentially in $t$. The gradient of the standard objective is therefore also exponential in $t$.

**Proof of Theorem 3.** According to the gradient descent iterations, we have

$$w_{t,\eta} = w_{t-1,\eta} - \eta \nabla f(w_{t-1,\eta}) = w_{t-1,\eta} - \eta H w_{t-1,\eta} = (I - \eta H) w_{t-1,\eta} = (I - \eta H)^t w_0.$$

Therefore, $\hat{F}(\eta) := f(w_{t,\eta}) = \frac{1}{2} w_0^\top (I - \eta H)^{2t} H w_0$. Taking the derivative of $\hat{F}(\eta)$,

$$\hat{F}'(\eta) = -t w_0^\top (I - \eta H)^{2t-1} H^2 w_0 = -t \sum_{i=1}^d c_i^2 \lambda_i^2 (1 - \eta \lambda_i)^{2t-1},$$

where $c_i = \langle w_0, u_i \rangle$. Taking the second derivative of $F(\eta)$,

$$F''(\eta) = t(2t-1)w_0^\top (I - \eta H)^{2t-2} H^3 w_0 = t(2t-1) \sum_{i=1}^d c_i^2 \lambda_i^3 (1 - \eta \lambda_i)^{2t-2}.$$

Since $L > \alpha$, we have $\hat{F}''(\eta) > 0$ for any $\eta$. That means $\hat{F}(\eta)$ is a strictly convex function in $\eta$ with a unique minimizer.

For any fixed $\eta \in (0, 2/L)$ we know $|1 - \eta \lambda_i| < 1$ for all $i \in [d]$. We have

$$\left| \hat{F}'(\eta) \right| \le t \sum_{i=1}^d c_i^2 \lambda_i^2 |1 - \eta \lambda_i|^{2t-1}.$$

For any fixed $\eta \in (2/L, \infty)$, we know $\eta L - 1 > 1$. We have

$$\hat{F}'(\eta)$$
$$= -tc_1^2 L^2 (1 - \eta L)^{2t-1} - t \sum_{i \neq 1:(1-\eta\lambda_i)\le 0} c_i^2 \lambda_i^2 (1 - \eta \lambda_i)^{2t-1} - t \sum_{i \neq 1:(1-\eta\lambda_i)>0} c_i^2 \lambda_i^2 (1 - \eta \lambda_i)^{2t-1}$$
$$\ge tc_1^2 L^2 (\eta L - 1)^{2t-1} - t \sum_{i=1}^d c_i^2 \lambda_i^2 \ge tc_1^2 L^2 (\eta L - 1)^{2t-1} - L^2 t,$$

where the last inequality uses $\sum_{i=1}^d c_i^2 = 1$. $\qquad\square$

### A.2 Alleviating meta-gradient vanishing/explosion

We prove when the the meta objective is chosen as $\frac{1}{t} \log f(w_{t,\eta})$, the meta-gradient is polynomially bounded. Furthermore, we show meta-gradient descent can converge to the optimal step size within polynomial iterations. Recall Theorem 4 as follows.

**Theorem 4.** *Let the meta objective be* $\hat{F}(\eta) = \frac{1}{t} \log f(w_{t,\eta})$. *We know* $\hat{F}(\eta)$ *has a unique minimizer* $\eta^*$ *and* $\hat{F}'(\eta) = O\left( \frac{L^3}{c_{\min}^2 \alpha (L-\alpha)} \right)$ *for all* $\eta \ge 0$. *Let* $\{\eta_k\}$ *be the GD sequence running on* $\hat{F}$ *with meta step size* $\mu_k = 1/\sqrt{k}$. *Suppose the starting step size* $\eta_0 \le M$. *Given any* $1/L > \epsilon > 0$, *there exists* $k' = \frac{M^6}{\epsilon^2} poly(\frac{1}{c_{\min}}, L, \frac{1}{\alpha}, \frac{1}{L-\alpha})$ *such that for all* $k \ge k'$, $|\eta_k - \eta^*| \le \epsilon$.

When we take the $\log$ of the function value, the derivative of the function value with respect to $\eta$ becomes much more stable. We will first show some structural result on $\hat{F}(\eta)$ – it has a unqiue minimizer and the gradient is polynomially bounded. Further the gradient is only close to 0 when the point $\eta$ is close to the unique minimizer. Then using such structural result we prove that meta-gradient descent converges.

**Proof of Theorem 4.** The proof consists of three claims. In the first claim, we show that $\hat{F}$ has a unique minimizer and the minus meta derivative always points to the minimizer. In the second claim, we show that $\hat{F}$ has bounded derivative. In the last claim, we show that for any $\eta$ that is outside the $\epsilon$-neighborhood of $\eta^*$, $|\hat{F}'(\eta)|$ is lower bounded. Finally, we combine these three claims to finish the proof.

**Claim 1.** *The meta objective* $\hat{F}$ *has only one stationary point that is also its unique minimizer* $\eta^*$. *For any* $\eta \in [0, \eta^*)$, $\hat{F}'(\eta) < 0$ *and for any* $\eta \in (\eta^*, \infty)$, $\hat{F}'(\eta) > 0$. *Furthermore, we know* $\eta^* \in [1/L, 1/\alpha]$.

We can compute the derivative of $\hat{F}$ in $\eta$ as follows,

$$\hat{F}'(\eta) = \frac{-2w_0^\top (I - \eta H)^{2t-1} H^2 w_0}{w_0^\top (I - \eta H)^{2t} H w_0} = \frac{-2 \sum_{i=1}^d c_i^2 \lambda_i^2 (1 - \eta \lambda_i)^{2t-1}}{\sum_{i=1}^d c_i^2 \lambda_i (1 - \eta \lambda_i)^{2t}}. \tag{3}$$

It's not hard to verify that the denominator $\sum_{i=1}^d c_i^2 \lambda_i (1 - \eta \lambda_i)^{2t}$ is always positive. Denote the numerator $-2 \sum_{i=1}^d c_i^2 \lambda_i^2 (1 - \eta \lambda_i)^{2t-1}$ as $g(\eta)$. Since $g'(\eta) > 0$ for any $\eta \in [0, \infty)$, we know $g(\eta)$

is strictly increasing in $\eta$. Combing with the fact that $g(0) < 0$ and $g(\infty) > 0$, we know there is a unique point (denoted as $\eta^*$) where $g(\eta^*) = 0$ and $g(\eta) < 0$ for all $\eta \in [0, \eta^*)$ and $g(\eta) > 0$ for all $\eta \in (\eta^*, \infty)$. Since the denominator in $\hat{F}'(\eta)$ is always positive and the numerator equals $g(\eta)$, we know there is a unique point $\eta^*$ where $\hat{F}'(\eta^*) = 0$ and $\hat{F}'(\eta) < 0$ for all $\eta \in [0, \eta^*)$ and $\hat{F}'(\eta) > 0$ for all $\eta \in (\eta^*, \infty)$. It's clear that $\eta^*$ is the minimizer of $\hat{F}$.

Also, it's not hard to verify that for any $\eta \in [0, 1/L)$, $\hat{F}'(\eta) < 0$ and for any $\eta \in (1/\alpha, \infty)$, $\hat{F}'(\eta) > 0$. This implies that $\eta^* \in [1/L, 1/\alpha]$.

**Claim 2.** *For any $\eta \in [0, \infty)$, we have*

$$|\hat{F}'(\eta)| \leq \frac{4L^3}{c_{\min}^2 \alpha (L - \alpha)} := D_{\max}.$$

For any $\eta \in [0, \frac{2}{\alpha + L}]$, we have $|1 - \eta\lambda_i| \leq 1 - \eta\alpha$ for all $i$. Dividing the numerator and denominator in $\hat{F}'(\eta)$ by $(1 - \eta\alpha)^{2t}$, we have

$$\left|\hat{F}'(\eta)\right| = 2 \frac{\left|\sum_{i=1}^{d} \frac{c_i^2 \lambda_i^2}{1 - \eta\alpha} \left(\frac{1 - \eta\lambda_i}{1 - \eta\alpha}\right)^{2t-1}\right|}{c_d^2 \alpha + \sum_{i=1}^{d-1} c_i^2 \lambda_i \left(\frac{1 - \eta\lambda_i}{1 - \eta\alpha}\right)^{2t}} \leq \frac{2 \sum_{i=1}^{d} c_i^2 \lambda_i^2}{c_d^2 \alpha (1 - \eta\alpha)} \leq \frac{2(\alpha + L) \sum_{i=1}^{d} c_i^2 \lambda_i^2}{c_d^2 \alpha (L - \alpha)} \leq \frac{4L^3}{c_d^2 \alpha (L - \alpha)},$$

where the second last inequality uses $\eta \leq \frac{2}{\alpha + L}$.

Similarly for any $\eta \in (\frac{2}{\alpha + L}, \infty)$, we have $|1 - \eta\lambda_i| \leq \eta L - 1$ for all $i$. Dividing the numerator and denominator in $\hat{F}'(\eta)$ by $(\eta L - 1)^{2t}$, we have

$$\hat{F}'(\eta) = 2 \frac{\left|\sum_{i=1}^{d} \frac{c_i^2 \lambda_i^2}{\eta L - 1} \left(\frac{1 - \eta\lambda_i}{\eta L - 1}\right)^{2t-1}\right|}{c_1^2 L + \sum_{i=2}^{d} c_i^2 \lambda_i \left(\frac{1 - \eta\lambda_i}{\eta L - 1}\right)^{2t}} \leq \frac{2 \sum_{i=1}^{d} c_i^2 \lambda_i^2}{c_1^2 L(\eta L - 1)} \leq \frac{2(\alpha + L) \sum_{i=1}^{d} c_i^2 \lambda_i^2}{c_1^2 L(L - \alpha)} \leq \frac{4L^3}{c_1^2 L(L - \alpha)}$$

where the last inequality uses $\eta \geq \frac{2}{\alpha + L}$.

Overall, we know for any $\eta \geq 0$,

$$|\hat{F}'(\eta)| \leq \frac{4L^3}{L - \alpha} \max\left(\frac{1}{c_d^2 \alpha}, \frac{1}{c_1^2 L}\right) \leq \frac{4L^3}{c_{\min}^2 \alpha (L - \alpha)}.$$

**Claim 3.** *Given $\hat{M} \geq 2/\alpha$ and $1/L > \epsilon > 0$, for any $\eta \in [0, \eta^* - \epsilon] \cup [\eta^* + \epsilon, \hat{M}]$, we have*

$$|F'(\eta)| \geq \min\left(\frac{2\epsilon c_d^2 \alpha^3}{L}, \frac{2\epsilon c_1^2 L^2}{(\hat{M}L - 1)^2}\right) \geq 2\epsilon c_{\min}^2 \min\left(\frac{\alpha^3}{L}, \frac{1}{\hat{M}^2}\right) := D_{\min}(\hat{M}).$$

If $\eta \in [0, \eta^* - \epsilon]$ and $\eta \leq \frac{2}{\alpha + L}$, we have

$$\hat{F}'(\eta) = -2 \frac{\sum_{i=1}^{d} c_i^2 \lambda_i^2 (1 - \eta\lambda_i)^{2t-1}}{\sum_{i=1}^{d} c_i^2 \lambda_i (1 - \eta\lambda_i)^{2t}} = -2 \frac{\sum_{i=1}^{d} c_i^2 \lambda_i^2 (1 - \eta\lambda_i)^{2t-1} - \sum_{i=1}^{d} c_i^2 \lambda_i^2 (1 - \eta^*\lambda_i)^{2t-1}}{\sum_{i=1}^{d} c_i^2 \lambda_i (1 - \eta\lambda_i)^{2t}},$$

where the second equality holds because $\sum_{i=1}^{d} c_i^2 \lambda_i^2 (1 - \eta^*\lambda_i)^{2t-1} = 0$. For the numerator, we have

$$\sum_{i=1}^{d} c_i^2 \lambda_i^2 (1 - \eta\lambda_i)^{2t-1} - \sum_{i=1}^{d} c_i^2 \lambda_i^2 (1 - \eta^*\lambda_i)^{2t-1} \geq c_d^2 \alpha^2 \left((1 - \eta\alpha)^{2t-1} - (1 - \eta^*\alpha)^{2t-1}\right)$$

$$\geq c_d^2 \alpha^2 \left((1 - \eta\alpha)^{2t-1} - (1 - \eta\alpha - \epsilon\alpha)^{2t-1}\right);$$

for the denominator, we have

$$\sum_{i=1}^{d} c_i^2 \lambda_i (1 - \eta\lambda_i)^{2t} \leq \left(\sum_{i=1}^{d} c_i^2 \lambda_i\right) (1 - \eta\alpha)^{2t},$$

where the second inequality holds because $|1 - \eta\lambda_i| \leq 1 - \eta\alpha$ for all $i$. Overall, we have when $\eta \in [0, \eta^* - \epsilon]$ and $\eta \leq \frac{2}{\alpha+L}$,

$$
\begin{aligned}
\left|\hat{F}'(\eta)\right| &\geq 2\frac{c_d^2\alpha^2\left((1-\eta\alpha)^{2t-1} - (1-\eta\alpha-\epsilon\alpha)^{2t-1}\right)}{\left(\sum_{i=1}^d c_i^2\lambda_i\right)(1-\eta\alpha)^{2t}} \\
&\geq \frac{2\epsilon c_d^2\alpha^3}{\left(\sum_{i=1}^d c_i^2\lambda_i\right)(1-\eta\alpha)} \geq \frac{2\epsilon c_d^2\alpha^3}{L},
\end{aligned}
$$

where the last inequality holds because $(1-\eta\alpha) \leq 1$ and $\sum_i^d c_i^2\lambda_i \leq L$.

Similarly, if $\eta \in [0, \eta^* - \epsilon]$ and $\eta \geq \frac{2}{\alpha+L}$, we have

$$
\begin{aligned}
\left|\hat{F}'(\eta)\right| &\geq 2\frac{c_1^2L^2\left((1-\eta L)^{2t-1} - (1-\eta L - \epsilon L)^{2t-1}\right)}{\left(\sum_{i=1}^d c_i^2\lambda_i\right)(1-\eta L)^{2t}} \\
&= 2\frac{c_1^2L^2\left((\eta L + \epsilon L - 1)^{2t-1} - (\eta L - 1)^{2t-1}\right)}{\left(\sum_{i=1}^d c_i^2\lambda_i\right)(\eta L - 1)^{2t}} \\
&\geq \frac{2\epsilon c_1^2L^3}{\left(\sum_{i=1}^d c_i^2\lambda_i\right)(\eta L - 1)^2} \geq \frac{2\epsilon c_1^2\alpha^2L^2}{(L-\alpha)^2},
\end{aligned}
$$

where the last inequality holds because $\eta \leq \eta^* - \epsilon \leq 1/\alpha$ and $\sum_i^d c_i^2\lambda_i \leq L$.

If $\eta \in [\eta^* + \epsilon, \infty)$ and $\eta \leq \frac{2}{\alpha+L}$, we have

$$
\begin{aligned}
\left|\hat{F}'(\eta)\right| &\geq 2\frac{c_d^2\alpha^2\left((1-\eta\alpha+\epsilon\alpha)^{2t-1} - (1-\eta\alpha)^{2t-1}\right)}{\left(\sum_{i=1}^d c_i^2\lambda_i\right)(1-\eta\alpha)^{2t}} \\
&\geq \frac{2\epsilon c_d^2\alpha^3}{L},
\end{aligned}
$$

If $\eta \in [\eta^* + \epsilon, \infty)$ and $\eta \geq \frac{2}{\alpha+L}$, we have

$$
\begin{aligned}
\left|\hat{F}'(\eta)\right| &\geq 2\frac{c_1^2L^2\left((1-\eta L+\eta\epsilon)^{2t-1} - (1-\eta L)^{2t-1}\right)}{\left(\sum_{i=1}^d c_i^2\lambda_i\right)(1-\eta L)^{2t}} \\
&\geq \frac{2\epsilon c_1^2L^3}{\left(\sum_{i=1}^d c_i^2\lambda_i\right)(\eta L - 1)^2} \geq \frac{2\epsilon c_1^2L^2}{(\hat{M}L - 1)^2},
\end{aligned}
$$

where the last inequality uses the assumption that $\eta \leq \hat{M}$.

With the above three claims, we are ready to prove the optimization result. By Claim 1, we know $F'(\eta) < 0$ for any $\eta \in [0, \eta^*)$ and $F'(\eta) > 0$ for any $\eta \in (\eta^*, \infty)$. So the opposite gradient descent always points to the minimizer.

Since $\mu_k = 1/\sqrt{k}$, when $k \geq k_1 := \frac{D_{\max}^2}{\epsilon^2}$ we know $\mu_k \leq \frac{\epsilon}{D_{\max}}$. By Claim 2, we know $|\hat{F}'(\eta)| \leq D_{\max}$ for all $\eta \geq 0$, which implies $|\mu_k\hat{F}'(\eta)| \leq \epsilon$ for all $k \geq k_1$. That means, meta gradient descent will never overshoot the minimizer by more than $\epsilon$ when $k \geq k_1$. In other words, after $k_1$ meta iterations, once $\eta$ enters the $\epsilon$-neighborhood of $\eta^*$, it will never leave this neighborhood.

We also know that at meta iteration $k_1$, we have $\eta_{k_1} \leq \max(1/\alpha + D_{\max}, M) := \hat{M}$. Here, $1/\alpha + D_{\max}$ comes from the case that the eta starts from the left of $\eta^*$ and overshoot to the right of $\eta^*$ by $D_{\max}$. Since $\eta^* \in [1/L, 1/\alpha]$, we have $|\eta_{k_1} - \eta^*| \leq \max(1/\alpha, 1/\alpha + D_{\max} - 1/L, M - 1/L) := R$. By Claim 3, we know that $|\hat{F}'(\eta)| \geq D_{\min}(\hat{M})$ for any $\eta \in [0, \eta^* - \epsilon] \cup [\eta^* + \epsilon, \hat{M}]$. Choosing some $k_2$ satisfying $\sum_{k=k_1}^{k_2} 1/\sqrt{k} \geq \frac{R}{D_{\min}}$, we know for any $k \geq k_2$, $|\eta_k - \eta^*| \leq \epsilon$.

Plugging in all the bounds for $D_{\min}, D_{\max}$ from Claim 3 and Claim 2, we know there exists $k_1 = \frac{1}{\epsilon^2}\text{poly}(\frac{1}{c_{\min}}, L, \frac{1}{\alpha}, \frac{1}{L-\alpha}), k_2 = \frac{M^6}{\epsilon^2}\text{poly}(\frac{1}{c_{\min}}, L, \frac{1}{\alpha}, \frac{1}{L-\alpha})$ satisfying these conditions. $\qquad\square$

Next, we show although the meta-gradient is polynomaily bounded, the intermediate results can still vanish or explode if we use back-propogation to compute the meta-gradient.

**Corollary 1.** *If we choose the meta-objective as $\hat{F}(\eta) = \frac{1}{t}\log f(w_{t,\eta})$, when computing the meta-gradient using back-propagation, there are intermediate results that are exponentially large/small in number of inner-steps $t$.*

**Proof of Corollary 1.** This is done by direct calculation. If we use back-propagation to compute the derivative of $\frac{1}{t}\log(f(w_{t,\eta}))$, we need to first compute $\frac{\partial f(w_{t,\eta})}{\partial}\frac{1}{t}\log(f(w_{t,\eta}))$ that equals $\frac{1}{tf(w_{t,\eta})}$. Same as the analysis in Theorem 3, we can show $\frac{1}{tf(w_{t,\eta})}$ is exponentially large when $\eta < 2/L$ and is exponentially small when $\eta > 2/L$. $\qquad\square$

# B    PROOFS OF TRAIN-BY-TRAIN V.S. TRAIN-BY-VALIDATION (GD)

In this section, we show when the number of samples is small and when the noise level is a large constant, train-by-train overfits to the noise in training tasks while train-by-validation generalizes well. We separately prove the results for train-by-train and train-by-validation in Theorem 7 and Theorem 8, respectively. Then, Theorem 5 is simply a combination of Theorem 7 and Theorem 8.

Recall that in the train-by-train setting, each task $P$ contains a training set $S_{\text{train}}$ with $n$ samples. The inner objective is defined as $\hat{f}(w) = \frac{1}{2n}\sum_{(x,y)\in S_{\text{train}}}(\langle w, x\rangle - y)^2$. Let $\{w_{\tau,\eta}\}$ be the GD sequence running on $\hat{f}(w)$ from initialization $0$ (with truncation). The meta-loss on task $P$ is defined as the inner objective of the last point, $\Delta_{TbT(n)}(\eta, P) = \hat{f}(w_{t,\eta}) = \frac{1}{2n}\sum_{(x,y)\in S_{\text{train}}}(\langle w_{t,\eta}, x\rangle - y)^2$. The empirical meta objective $\hat{F}_{TbT(n)}(\eta)$ is the average of the meta-loss across $m$ different tasks. We show that under $\hat{F}_{TbT(n)}(\eta)$, the optimal step size is a constant and the learned weight is far from ground truth $w^*$ on new tasks. We prove Theorem 7 in Section B.2.

**Theorem 7.** *Let the meta objective $\hat{F}_{TbT(n)}(\eta)$ be as defined in Equation 1 with $n \in [d/4, 3d/4]$. Assume noise level $\sigma$ is a large constant $c_1$. Assume unroll length $t \geq c_2$, number of training tasks $m \geq c_3\log(mt)$ and dimension $d \geq c_4\log(m)$ for certain constants $c_2, c_3, c_4$. With probability at least $0.99$ in the sampling of the training tasks, we have*

$$\eta^*_{train} = \Theta(1) \text{ and } \mathbb{E}\left\|w_{t,\eta^*_{train}} - w^*\right\|^2 = \Omega(1)\sigma^2,$$

*for all $\eta^*_{train} \in \arg\min_{\eta \geq 0}\hat{F}_{TbT(n)}(\eta)$, where the expectation is taken over new tasks.*

In Theorem 7, $\Omega(1)$ is an absolute constant independent with $\sigma$. Intuitively, the reason that train-by-train performs badly in this setting is because there is a way to set the step size to a constant such that gradient descent converges very quickly to the empirical risk minimizer, therefore making the train-by-train objective very small. However, when the noise is large and the number of samples is smaller than the dimension, the empirical risk minimizer (ERM) overfits to the noise and is not the best solution.

In the train-by-validation setting, each task $P$ contains a training set $S_{\text{train}}$ with $n_1$ samples and a validation set with $n_2$ samples. The inner objective is defined as $\hat{f}(w) = \frac{1}{2n_1}\sum_{(x,y)\in S_{\text{train}}}(\langle w, x\rangle - y)^2$. Let $\{w_{\tau,\eta}\}$ be the GD sequence running on $\hat{f}(w)$ from initialization $0$ (with truncation). For each task $P$, the meta-loss $\Delta_{TbV(n_1,n_2)}(\eta, P)$ is defined as the loss of the last point $w_{t,\eta}$ evaluated on the validation set $S_{\text{valid}}$. That is, $\Delta_{TbV(n_1,n_2)}(\eta, P) = \frac{1}{2n_2}\sum_{(x,y)\in S_{\text{valid}}}(\langle w_{t,\eta}, x\rangle - y)^2$. The empirical meta objective $\hat{F}_{TbV(n_1,n_2)}(\eta)$ is the average of the meta-loss across $m$ different tasks $P_1, P_2, ..., P_m$. We show that under $\hat{F}_{TbV(n_1,n_2)}(\eta)$, the optimal step size is $\Theta(1/t)$ and the learned weight is better than initialization $0$ by a constant on new tasks. Theorem 8 is proved in Section B.3.

**Theorem 8.** *Let the meta objective $\hat{F}_{TbV(n_1,n_2)}(\eta)$ be as defined in Equation 2 with $n_1, n_2 \in [d/4, 3d/4]$. Assume noise level $\sigma$ is a large constant $c_1$. Assume unroll length $t \geq c_2$, number of*

*training tasks $m \geq c_3$ and dimension $d \geq c_4 \log(t)$ for certain constants $c_2, c_3, c_4$. With probability at least $0.99$ in the sampling of training tasks, we have*

$$\eta^*_{valid} = \Theta(1/t) \text{ and } \mathbb{E}\left\|w_{t,\eta^*_{valid}} - w^*\right\|^2 = \|w^*\|^2 - \Omega(1)$$

*for all $\eta^*_{valid} \in \arg\min_{\eta \geq 0} \hat{F}_{TbV(n_1, n_2)}(\eta)$, where the expectation is taken over new tasks.*

Intuitively, train-by-validation is optimizing the right objective. As long as the meta-training problem has good generalization performance (that is, good performance on a few tasks implies good performance on the distribution of tasks), then train-by-validation should be able to choose the optimal learning rate. The step size of $\Theta(1/t)$ here serves as regularization similar to early-stopping, which allows gradient descent algorithm to achieve better error on test data.

**Notations** We define more quantities that are useful in the analysis. In the train by train setting, given a task $P_k := (\mathcal{D}(w_k^*), S_{\text{train}}^{(k)}, \ell)$. The training set $S_{\text{train}}^{(k)}$ contains $n$ samples $\{x_i^{(k)}, y_i^{(k)}\}_{i=1}^n$ with $y_i^{(k)} = \left\langle w_k^*, x_i^{(k)} \right\rangle + \xi_i^{(k)}$.

Let $X_{\text{train}}^{(k)}$ be an $n \times d$ matrix with its $i$-th row as $(x_i^{(k)})^\top$. Let $H_{\text{train}}^{(k)} := \frac{1}{n}(X_{\text{train}}^{(k)})^\top X_{\text{train}}^{(k)}$ be the covariance matrix of the inputs in $S_{\text{train}}^{(k)}$. Let $\xi_{\text{train}}^{(k)}$ be an $n$-dimensional column vector with its $i$-th entry equal to $\xi_i^{(k)}$.

Since $n \leq d$, with probability 1, we know $X_{\text{train}}^{(k)}$ is full row rank. Therefore, $X_{\text{train}}^{(k)}$ has pseudo-inverse $(X_{\text{train}}^{(k)})^\dagger$ such that $X_{\text{train}}^{(k)}(X_{\text{train}}^{(k)})^\dagger = I_n$. It's not hard to verify that there exists $w_{\text{train}}^{(k)} = \text{Proj}_{(X_{\text{train}}^{(k)})^\top} w_k^* + (X_{\text{train}}^{(k)})^\dagger \xi_{\text{train}}^{(k)}$ such that $y_i^{(k)} = \left\langle w_{\text{train}}^{(k)}, x_i^{(k)} \right\rangle$ for every $(x_i^{(k)}, y_i^{(k)}) \in S_{\text{train}}^{(k)}$. Here, $\text{Proj}_{(X_{\text{train}}^{(k)})^\top}$ is the projection matrix onto the column span of $(X_{\text{train}}^{(k)})^\top$. We also denote $\text{Proj}_{(X_{\text{train}}^{(k)})^\top} w_k^*$ as $(w_{\text{train}}^{(k)})^*$. We use $B_{t,\eta}^{(k)}$ to denote $(I - (I - \eta H_{\text{train}}^{(k)})^t)$. Let $w_{t,\eta}^{(k)}$ be the weight obtained by running GD on $S_{\text{train}}^{(k)}$ with step size $\eta$ (with truncation).

With the above notations, it's not hard to verify that for task $P_k$, the inner objective $\hat{f}(w) = \frac{1}{2}\left\|w - w_{\text{train}}^{(k)}\right\|^2_{H_{\text{train}}^{(k)}}$. The meta-loss on task $P_k$ is just $\Delta_{TbT(n)}(\eta, P_k) = \frac{1}{2}\left\|w_{t,\eta} - w_{\text{train}}^{(k)}\right\|^2_{H_{\text{train}}^{(k)}}$.

In the train-by-validation setting, each task $P_k$ contains a training set $S_{\text{train}}^{(k)}$ with $n_1$ samples and a validation set $S_{\text{valid}}^{(k)}$ with $n_2$ samples. Similar as above, for the training set $S_{\text{train}}^{(k)}$, we can define $\xi_{\text{train}}^{(k)}, X_{\text{train}}^{(k)}, H_{\text{train}}^{(k)}, w_{\text{train}}^{(k)}, B_{t,\eta}^{(k)}, w_{t,\eta}^{(k)}$; for the validation set $S_{\text{valid}}^{(k)}$, we can define $\xi_{\text{valid}}^{(k)}, X_{\text{valid}}^{(k)}, H_{\text{valid}}^{(k)}, w_{\text{valid}}^{(k)}$. With these notations, the inner objective is $\hat{f}(w) = \frac{1}{2}\left\|w - w_{\text{train}}^{(k)}\right\|^2_{H_{\text{train}}^{(k)}}$ and the meta-loss is $\Delta_{TbV(n_1, n_2)}(\eta, P_k) = \frac{1}{2}\left\|w_{t,\eta} - w_{\text{valid}}^{(k)}\right\|^2_{H_{\text{valid}}^{(k)}}$.

We also use these notations without index $k$ to refer to the quantities defined on task $P$. In the proofs, we ignore the subscripts on $n, n_1, n_2$ and simply write $\Delta_{TbT}(\eta, P_k), \Delta_{TbV}(\eta, P_k), \hat{F}_{TbT}, \hat{F}_{TbV}, F_{TbT}, F_{TbV}$.

## B.1 OVERALL PROOF STRATEGY

In this section (and the next), we follow similar proof strategies that consists of three steps.

**Step 1:** First, we show for both train-by-train and train-by-validation, there is a good step size that achieves small empirical meta-objective (however the step sizes and the empirical meta-objective they achieve are different in the two settings). This does not necessarily mean that the actual optimal step size is exactly the good step size that we propose, but it gives an upperbound on the empirical meta-objective for the optimal step size.

**Step 2:** Second, we define a threshold step size such that for any step size larger than it, the empirical meta-objective must be higher than what was achieved at the good step size in Step 1. This immediately implies that the optimal step size cannot exceed this threshold step size.

**Step 3:** Third, we show the meta-learning problem has good generalization performance, that is, if a learning rate $\eta$ performs well on the training tasks, it must also perform well on the task distribution, and vice versa. Thanks to Step 1 and Step 2, we know the optimal step size cannot exceed certain threshold and then only need to prove generalization result within this range. The generalization result is not surprising as we only have a single trainable parameter $\eta$, however we also emphasize that this is non-trivial as we will not restrict the step size $\eta$ to be small enough that the algorithms do not diverge. Instead we use a truncation to alleviate the diverging problem (this allows us to run the algorithm on distribution of data whose largest possible learning rate is unknown).

Combing Step 1, 2, 3, we know the population meta-objective has to be small at the optimal step size. Finally, we show that as long as the population meta-objective is small, the performance of the algorithms satisfy what we stated in Theorem 5. The last step is easier for the train-by-validation setting, because its meta-objective is exactly the correct measure that we are looking at; for the train-by-train setting we instead look at the property of empirical risk minimizer (ERM), and show that anything close to the ERM is going to behave similarly.

### B.2 TRAIN-BY-TRAIN (GD)

Recall Theorem 7 as follows.

**Theorem 7.** *Let the meta objective $\hat{F}_{TbT(n)}(\eta)$ be as defined in Equation 1 with $n \in [d/4, 3d/4]$. Assume noise level $\sigma$ is a large constant $c_1$. Assume unroll length $t \geq c_2$, number of training tasks $m \geq c_3 \log(mt)$ and dimension $d \geq c_4 \log(m)$ for certain constants $c_2, c_3, c_4$. With probability at least $0.99$ in the sampling of the training tasks, we have*

$$\eta^*_{train} = \Theta(1) \text{ and } \mathbb{E} \left\| w_{t,\eta^*_{train}} - w^* \right\|^2 = \Omega(1)\sigma^2,$$

*for all $\eta^*_{train} \in \arg\min_{\eta \geq 0} \hat{F}_{TbT(n)}(\eta)$, where the expectation is taken over new tasks.*

According to the data distribution, we know $X_{\text{train}}$ is an $n \times d$ random matrix with each entry i.i.d. sampled from standard Gaussian distribution. In the following lemma, we show that the covariance matrix $H_{\text{train}}$ is approximately isotropic when $d/4 \leq n \leq 3d/4$. Specifically, we show $\frac{\sqrt{d}}{\sqrt{L}} \leq \sigma_i(X_{\text{train}}) \leq \sqrt{Ld}$ and $\frac{1}{L} \leq \lambda_i(H_{\text{train}}) \leq L$ for all $i \in [n]$ with $L = 100$. We use letter $L$ to denote the upper bound of $\|H_{\text{train}}\|$ to emphasize that this bounds the smoothness of the inner objective. Throughout this section, we use letter $L$ to denote constant $100$. The proof of Lemma 1 follows from random matrix theory. We defer its proof into Section B.2.4.

**Lemma 1.** *Let $X \in \mathbb{R}^{n \times d}$ be a random matrix with each entry i.i.d. sampled from standard Gaussian distribution. Let $H := 1/nX^\top X$. Assume $n = cd$ with $c \in [\frac{1}{4}, \frac{3}{4}]$. Then, with probability at least $1 - \exp(-\Omega(d))$, there exists constant $L = 100$ such that*

$$\frac{\sqrt{d}}{\sqrt{L}} \leq \sigma_i(X) \leq \sqrt{Ld} \text{ and } \frac{1}{L} \leq \lambda_i(H) \leq L,$$

*for all $i \in [n]$.*

In this section, we always assume the size of each training set is within $[d/4, 3d/4]$ so Lemma 1 holds. Since $\|H_{\text{train}}\|$ is upper bounded by $L$ with high probability, we know the GD sequence converges to $w_{\text{train}}$ for $\eta \in [0, 1/L]$. In Lemma 2, we prove that the empirical meta objective $\hat{F}_{TbT}$ monotonically decreases as $\eta$ increases until $1/L$. Also, we show $\hat{F}_{TbT}$ is exponentially small in $t$ at step size $1/L$. This serves as step 1 in Section B.1. The proof is deferred into Section B.2.1.

**Lemma 2.** *With probability at least $1 - m\exp(-\Omega(d))$, $\hat{F}_{TbT}(\eta)$ is monotonically decreasing in $[0, 1/L]$ and*

$$\hat{F}_{TbT}(1/L) \leq 2L^2\sigma^2 \left(1 - \frac{1}{L^2}\right)^t.$$

When the step size is larger than $1/L$, the GD sequence can diverge, which incurs a high loss in meta objective. Later in Definition 1, we define a step size $\tilde{\eta}$ such that the GD sequence gets truncated with descent probability for any step size that is larger than $\tilde{\eta}$. In Lemma 3, we show with high probability, the empirical meta objective is high for all $\eta > \tilde{\eta}$. This serves as step 2 in the proof strategy described in Section B.1. The proof is deferred into Section B.2.2.

**Lemma 3.** *With probability at least* $1 - \exp(-\Omega(m))$,

$$\hat{F}_{TbT}(\eta) \geq \frac{\sigma^2}{10L^8},$$

*for all* $\eta > \tilde{\eta}$.

By Lemma 2 and Lemma 3, we know the optimal step size must lie in $[1/L, \tilde{\eta}]$. We can also show $1/L < \tilde{\eta} < 3L$, so $\eta^*_{\text{train}}$ is a constant. To relate the empirical loss at $\eta^*_{\text{train}}$ to the population loss. We prove a generalization result for step sizes within $[1/L, \tilde{\eta}]$. This serves as step 3 in Section B.1. The proof is deferred into Section B.2.3.

**Lemma 4.** *Suppose* $\sigma$ *is a large constant* $c_1$. *Assume* $t \geq c_2, d \geq c_4$ *for certain constants* $c_2, c_4$. *With probability at least* $1 - m \exp(-\Omega(d)) - O(t + m) \exp(-\Omega(m))$,

$$|F_{TbT}(\eta) - \hat{F}_{TbT}(\eta)| \leq \frac{\sigma^2}{L^3},$$

*for all* $\eta \in [1/L, \tilde{\eta}]$,

Combining the above lemmas, we know the population meta objective $F_{TbT}$ is small at $\eta^*_{\text{train}}$, which means $w_{t,\eta^*_{\text{train}}}$ is close to the ERM solution. Since the ERM solution overfits to the noise in training tasks, we know $\left\| w_{t,\eta^*_{\text{train}}} - w^* \right\|$ has to be large. We present the proof of Theorem 7 as follows.

**Proof of Theorem 7.** We assume $\sigma$ is a large constant in this proof. According to Lemma 2, we know with probability at least $1 - m \exp(-\Omega(d))$, $\hat{F}_{TbT}(\eta)$ is monotonically decreasing in $[0, 1/L]$ and $\hat{F}_{TbT}(1/L) \leq 2L^2\sigma^2(1 - 1/L^2)^t$. This implies that the optimal step size $\eta^*_{\text{train}} \geq 1/L$ and $\hat{F}_{TbT}(\eta^*_{\text{train}}) \leq 2L^2\sigma^2(1-1/L^2)^t$. By Lemma 3, we know with probability at least $1-\exp(-\Omega(m))$, $\hat{F}_{TbT}(\eta) \geq \frac{\sigma^2}{10L^8}$ for all $\eta > \tilde{\eta}$, where $\tilde{\eta}$ is defined in Definition 1. As long as $t \geq c_2$ for certain constant $c_2$, we know $\frac{\sigma^2}{10L^8} > 2L^2\sigma^2(1-1/L^2)^t$, which then implies that the optimal step size $\eta^*_{\text{train}}$ lies in $[1/L, \tilde{\eta}]$. According to Lemma 6, we know $\tilde{\eta} \in (1/L, 3L)$. Therefore $\eta^*_{\text{train}}$ is a constant.

According to Lemma 4, we know with probability at least $1 - m \exp(-\Omega(d)) - O(t + m) \exp(-\Omega(m))$, $|F_{TbT}(\eta) - \hat{F}_{TbT}(\eta)| \leq \frac{\sigma^2}{L^3}$, for all $\eta \in [1/L, \tilde{\eta}]$. As long as $t$ is larger than some constant, we have $\hat{F}_{TbT}(\eta^*_{\text{train}}) \leq \frac{\sigma^2}{L^3}$. Combing with the generalization result, we have $F_{TbT}(\eta^*_{\text{train}}) \leq \frac{2\sigma^2}{L^3}$. Next, we show that under a small population loss, $\mathbb{E} \left\| w_{t,\eta^*_{\text{train}}} - w^* \right\|^2$ has to be large.

Let $\mathcal{E}_1$ be the event that $\sqrt{d}/\sqrt{L} \leq \sigma_i(X_{\text{train}}) \leq \sqrt{L}d$ and $1/L \leq \lambda_i(H_{\text{train}}) \leq L$ for all $i \in [n]$ and $\sqrt{d}\sigma/4 \leq \|\xi_{\text{train}}\| \leq \sqrt{d}\sigma$. We have

$$\begin{aligned}
\mathbb{E} \left\| w_{t,\eta^*_{\text{train}}} - w_{\text{train}} \right\|^2_{H_{\text{train}}} &\geq \frac{1}{L}\mathbb{E} \left\| w_{t,\eta^*_{\text{train}}} - w_{\text{train}} \right\|^2 \mathbb{1}\{\mathcal{E}_1\} \\
&\geq \frac{1}{L} \left( \mathbb{E} \left\| w_{t,\eta^*_{\text{train}}} - w^*_{\text{train}} - (X_{\text{train}})^\dagger \xi_{\text{train}} \right\| \mathbb{1}\{\mathcal{E}_1\} \right)^2 \\
&\geq \frac{1}{L} \left( \mathbb{E} \left\| (X_{\text{train}})^\dagger \xi_{\text{train}} \right\| \mathbb{1}\{\mathcal{E}_1\} - \mathbb{E} \left\| w_{t,\eta^*_{\text{train}}} - w^*_{\text{train}} \right\| \mathbb{1}\{\mathcal{E}_1\} \right)^2.
\end{aligned}$$

Since $\mathbb{E} \left\| w_{t,\eta^*_{\text{train}}} - w_{\text{train}} \right\|^2_{H_{\text{train}}} \leq \frac{4\sigma^2}{L^3}$, this then implies

$$\mathbb{E} \left\| (X_{\text{train}})^\dagger \xi_{\text{train}} \right\| \mathbb{1}\{\mathcal{E}_1\} - \mathbb{E} \left\| w_{t,\eta^*_{\text{train}}} - w^*_{\text{train}} \right\| \mathbb{1}\{\mathcal{E}_1\} \leq \sqrt{L\frac{4\sigma^2}{L^3}} = \frac{2\sigma}{L}.$$

Conditioning on $\mathcal{E}_1$, we can lower bound $\left\| (X_{\text{train}})^\dagger \xi_{\text{train}} \right\|$ by $\frac{\sigma}{4\sqrt{L}}$. According to Lemma 1 and Lemma 45, we know $\Pr[\mathcal{E}_1] \geq 1 - \exp(-\Omega(d))$. As long as $d$ is at least certain constant, we have

$\Pr[\mathcal{E}_1] \geq 0.9$. This then implies $\mathbb{E}\left\|(X_{\text{train}})^\dagger \xi_{\text{train}}\right\| \mathbb{1}\{\mathcal{E}_1\} \geq \frac{9\sigma}{40\sqrt{L}}$. Therefore, we have

$$\mathbb{E}\left\|w_{t,\eta^*_{\text{train}}} - w^*_{\text{train}}\right\| \mathbb{1}\{\mathcal{E}_1\} \geq \frac{9\sigma}{40\sqrt{L}} - \frac{2\sigma}{L} = \frac{9\sigma}{4L} - \frac{2\sigma}{L} = \frac{\sigma}{4L},$$

where the first equality uses $L = 100$. Then, we have

$$\mathbb{E}\left\|w_{t,\eta^*_{\text{train}}} - w^*\right\|^2 \geq \mathbb{E}\left\|w_{t,\eta^*_{\text{train}}} - w^*_{\text{train}}\right\|^2 \mathbb{1}\{\mathcal{E}_1\} \geq \left(\mathbb{E}\left\|w_{t,\eta^*_{\text{train}}} - w^*_{\text{train}}\right\| \mathbb{1}\{\mathcal{E}_1\}\right)^2 \geq \frac{\sigma^2}{16L^2},$$

where the first inequality holds because for any $S_{\text{train}}$, $w^*_{\text{train}}$ is the projection of $w^*$ on the subspace of $S_{\text{train}}$ and $w_{t,\eta^*_{\text{train}}}$ is also in this subspace. Taking a union bound for all the bad events, we know this result holds with probability at least 0.99 as long as $\sigma$ is a large constant $c_1$ and $t \geq c_2, m \geq c_3 \log(mt)$ and $d \geq c_4 \log(m)$ for certain constants $c_2, c_3, c_4$. $\qquad\square$

### B.2.1 Behavior of $\hat{F}_{TbT}$ for $\eta \in [0, 1/L]$

In this section, we prove the empirical meta objective $\hat{F}_{TbT}$ is monotonically decreasing in $[0, 1/L]$. Furthermore, we show $\hat{F}_{TbT}(1/L)$ is exponentially small in $t$.

**Lemma 2.** *With probability at least* $1 - m \exp(-\Omega(d))$, $\hat{F}_{TbT}(\eta)$ *is monotonically decreasing in* $[0, 1/L]$ *and*

$$\hat{F}_{TbT}(1/L) \leq 2L^2\sigma^2 \left(1 - \frac{1}{L^2}\right)^t.$$

**Proof of Lemma 2.** For each $k \in [m]$, let $\mathcal{E}_k$ be the event that $\sqrt{d}/\sqrt{L} \leq \sigma_i(X_{\text{train}}) \leq \sqrt{Ld}$ and $1/L \leq \lambda_i(H_{\text{train}}) \leq L$ for all $i \in [n]$ and $\sqrt{d}\sigma/4 \leq \|\xi_{\text{train}}\| \leq \sqrt{d}\sigma$. Here, $L$ is constant 100 from Lemma 1. According to Lemma 1 and Lemma 45, we know for each $k \in [m]$, $\mathcal{E}_k$ happens with probability at least $1 - \exp(-\Omega(d))$. Taking a union bound over all $k \in [m]$, we know $\cap_{k\in[m]}\mathcal{E}_k$ holds with probability at least $1 - m \exp(-\Omega(d))$. From now on, we assume $\cap_{k\in[m]}\mathcal{E}_k$ holds.

Let's first consider each individual loss function $\Delta_{TbT}(\eta, P_k)$. Let $\{\hat{w}^{(k)}_{\tau,\eta}\}$ be the GD sequence without truncation. We have

$$\hat{w}^{(k)}_{\tau,\eta} - w^{(k)}_{\text{train}} = \hat{w}^{(k)}_{\tau-1,\eta} - w^{(k)}_{\text{train}} - \eta H^{(k)}_{\text{train}}(\hat{w}^{(k)}_{\tau-1,\eta} - w^{(k)}_{\text{train}})$$
$$= (I - \eta H^{(k)}_{\text{train}})(\hat{w}^{(k)}_{\tau-1,\eta} - w^{(k)}_{\text{train}}) = -(I - \eta H^{(k)}_{\text{train}})^t w^{(k)}_{\text{train}}.$$

For any $\eta \in [0, 1/L]$, we have $\left\|\hat{w}^{(k)}_{\tau,\eta}\right\| \leq \left\|w^{(k)}_{\text{train}}\right\| = \left\|(w^{(k)}_{\text{train}})^* + (X^{(k)}_{\text{train}})^\dagger \xi^{(k)}_{\text{train}}\right\| \leq 2\sqrt{L}\sigma$ for any $\tau$. Therefore, $\left\|w^{(k)}_{t,\eta}\right\|$ never exceeds the norm threshold and never gets truncated.

Noticing that $\Delta_{TbT}(\eta, P_k) = \frac{1}{2}(w^{(k)}_{t,\eta} - w^{(k)}_{\text{train}})^\top H^{(k)}_{\text{train}}(w^{(k)}_{t,\eta} - w^{(k)}_{\text{train}})$, we have

$$\Delta_{TbT}(\eta, P_k) = \frac{1}{2}(w^{(k)}_{\text{train}})^\top H^{(k)}_{\text{train}}(I - \eta H^{(k)}_{\text{train}})^{2t} w^{(k)}_{\text{train}}.$$

Taking the derivative of $\Delta_{TbT}(\eta, P_k)$ in $\eta$, we have

$$\frac{\partial}{\partial\eta}\Delta_{TbT}(\eta, P_k) = -t(w^{(k)}_{\text{train}})^\top (H^{(k)}_{\text{train}})^2 (I - \eta H^{(k)}_{\text{train}})^{2t-1} w^{(k)}_{\text{train}}.$$

Conditioning on $\mathcal{E}_k$, we know $1/L \leq \lambda_i(H^{(k)}_{\text{train}}) \leq L$ for all $i \in [n]$ and $H^{(k)}_{\text{train}}$ is full rank in the row span of $X^{(k)}_{\text{train}}$. Therefore, we know $\frac{\partial}{\partial\eta}\Delta_{TbT}(\eta, P_k) < 0$ for all $\eta \in [0, 1/L]$. Here, we assume $\left\|w^{(k)}_{\text{train}}\right\| > 0$, which happens with probability 1.

Overall, we know that conditioning on $\cap_{k\in[m]}\mathcal{E}_k$, every $\Delta_{TbT}(\eta, P_k)$ is strictly decreasing for $\eta \in [0, 1/L]$. Since $\hat{F}_{TbT}(\eta) := \frac{1}{m}\sum_{k=1}^m \Delta_{TbT}(\eta, P_k)$, we know $\hat{F}_{TbT}(\eta)$ is strictly decreasing when $\eta \in [0, 1/L]$.

At step size $\eta = 1/L$, we have

$$\Delta_{TbT}(\eta, P_k) = \frac{1}{2}(w_{\text{train}}^{(k)})^\top H_{\text{train}}^{(k)}(I - \eta H_{\text{train}}^{(k)})^{2t} w_{\text{train}}^{(k)}$$

$$\leq \frac{1}{2}L\left(1 - \frac{1}{L^2}\right)^t \left\|w_{\text{train}}^{(k)}\right\|^2 \leq 2L^2\sigma^2\left(1 - \frac{1}{L^2}\right)^t,$$

where we upper bound $\left\|w_{\text{train}}^{(k)}\right\|^2$ by $4L\sigma^2$ at the last step. Therefore, we have $\hat{F}_{TbT}(1/L) \leq 2L^2\sigma^2(1 - \frac{1}{L^2})^t$. □

### B.2.2 Lower bounding $\hat{F}_{TbT}$ for $\eta \in (\tilde{\eta}, \infty)$

In this section, we prove that the empirical meta objective is lower bounded by $\Omega(\sigma^2)$ with high probability for $\eta \in (\tilde{\eta}, \infty)$. Step size $\tilde{\eta}$ is defined such that there is a descent probability of diverging for any step size larger than $\tilde{\eta}$. Then, we show the contribution from these truncated sequence will be enough to provide an $\Omega(\sigma^2)$ lower bound for $\hat{F}_{TbT}$. The proof of Lemma 3 is given at the end of this section.

**Lemma 3.** *With probability at least $1 - \exp(-\Omega(m))$,*

$$\hat{F}_{TbT}(\eta) \geq \frac{\sigma^2}{10L^8},$$

*for all $\eta > \tilde{\eta}$.*

We define $\tilde{\eta}$ as the smallest step size such that the contribution from the truncated sequence in the population meta objective exceeds certain threshold. The precise definition is as follows.

**Definition 1.** *Given a training task $P$, let $\mathcal{E}_1$ be the event that $\sqrt{d}/\sqrt{L} \leq \sigma_i(X_{train}) \leq \sqrt{Ld}$ and $1/L \leq \lambda_i(H_{train}) \leq L$ for all $i \in [n]$ and $\sqrt{d}\sigma/4 \leq \|\xi_{train}\| \leq \sqrt{d}\sigma$. Let $\bar{\mathcal{E}}_2(\eta)$ be the event that the GD sequence is truncated with step size $\eta$. Define $\tilde{\eta}$ as follows,*

$$\tilde{\eta} = \inf\left\{\eta \geq 0 \bigg| \mathbb{E}\frac{1}{2}\|w_{t,\eta} - w_{train}\|_{H_{train}}^2 \mathbb{1}\left\{\mathcal{E}_1 \cap \bar{\mathcal{E}}_2(\eta)\right\} \geq \frac{\sigma^2}{L^6}\right\}.$$

In the next lemma, we prove that for any fixed training set, $\mathbb{1}\left\{\mathcal{E}_1 \cap \bar{\mathcal{E}}_2(\eta')\right\} \geq \mathbb{1}\left\{\mathcal{E}_1 \cap \bar{\mathcal{E}}_2(\eta)\right\}$ for any $\eta' \geq \eta$. This immediately implies that $\Pr[\mathcal{E}_1 \cap \bar{\mathcal{E}}_2(\eta)]$ and $\mathbb{E}\frac{1}{2}\|w_{t,\eta} - w_{\text{train}}\|_{H_{\text{train}}}^2 \mathbb{1}\left\{\mathcal{E}_1 \cap \bar{\mathcal{E}}_2(\eta)\right\}$ is non-decreasing in $\eta$.

Basically we need to show, conditioning on $\mathcal{E}_1$, if a GD sequence gets truncated at step size $\eta$, it must be also truncated for larger step sizes. Let $\{w'_{\tau,\eta}\}$ be the GD sequence without truncation. We only need to show that for any $\tau$, if $\|w'_{\tau,\eta}\|$ exceeds the norm threshold, $\|w'_{\tau,\eta'}\|$ must also exceed the norm threshold for any $\eta' \geq \eta$. This is easy to prove if $\tau$ is odd because in this case $\|w'_{\tau,\eta}\|$ is always non-decreasing in $\eta$. The case when $\tau$ is even is trickier because there indeed exists certain range of $\eta$ such that $\|w'_{\tau,\eta}\|$ is decreasing in $\eta$. We manage to prove that this problematic case cannot happen when $\|w'_{\tau,\eta}\|$ is at least $4\sqrt{L}\sigma$. The full proof of Lemma 5 is deferred into Section B.2.4.

**Lemma 5.** *Fixing a task $P$, let $\mathcal{E}_1$ and $\bar{\mathcal{E}}_2(\eta)$ be as defined in Definition 1. We have*

$$\mathbb{1}\left\{\mathcal{E}_1 \cap \bar{\mathcal{E}}_2(\eta')\right\} \geq \mathbb{1}\left\{\mathcal{E}_1 \cap \bar{\mathcal{E}}_2(\eta)\right\},$$

*for any $\eta' \geq \eta$.*

In the next Lemma, we prove that $\tilde{\eta}$ must lie within $(1/L, 3L)$. We prove this by showing that the GD sequence never gets truncated for $\eta \in [0, 2/L]$ and almost always gets truncated for $\eta \in [2.5L, \infty)$. The proof is deferred into Section B.2.4.

**Lemma 6.** *Let $\tilde{\eta}$ be as defined in Definition 1. Suppose $\sigma$ is a large constant $c_1$. Assume $t \geq c_2, d \geq c_4$ for some constants $c_2, c_4$. We have*

$$1/L < \tilde{\eta} < 3L.$$

Now, we are ready to give the proof of Lemma 3.

**Proof of Lemma 3.** Let $\mathcal{E}_1$ and $\bar{\mathcal{E}}_2(\eta)$ be as defined in Definition 1. For the simplicity of the proof, we assume $\mathbb{E}\frac{1}{2}\|w_{t,\tilde{\eta}} - w_{\text{train}}\|_{H_{\text{train}}}^2 \mathbb{1}\left\{\mathcal{E}_1 \cap \bar{\mathcal{E}}_2(\tilde{\eta})\right\} \geq \frac{\sigma^2}{L^6}$. We will discuss the proof for the other case at the end, which is very similar.

Conditioning on $\mathcal{E}_1$, we know $\frac{1}{2}\|w_{t,\tilde{\eta}} - w_{\text{train}}\|_{H_{\text{train}}}^2 \leq 18L^2\sigma^2$. Therefore, we know $\Pr[\mathcal{E}_1 \cap \bar{\mathcal{E}}_2(\tilde{\eta})] \geq \frac{1}{18L^8}$. For each task $P_k$, define $\mathcal{E}_1^{(k)}$ and $\bar{\mathcal{E}}_2^{(k)}(\eta)$ as the corresponding events on training set $S_{\text{train}}^{(k)}$. By Hoeffding's inequality, we know with probability at least $1 - \exp(-\Omega(m))$,

$$\frac{1}{m}\sum_{k=1}^m \mathbb{1}\left\{\mathcal{E}_1^{(k)} \cap \bar{\mathcal{E}}_2^{(k)}(\tilde{\eta})\right\} \geq \frac{1}{20L^8}.$$

By Lemma 5, we know $\mathbb{1}\left\{\mathcal{E}_1^{(k)} \cap \bar{\mathcal{E}}_2^{(k)}(\eta)\right\} \geq \mathbb{1}\left\{\mathcal{E}_1^{(k)} \cap \bar{\mathcal{E}}_2^{(k)}(\tilde{\eta})\right\}$ for any $\eta \geq \tilde{\eta}$. Then, we can lower bound $\hat{F}_{TbT}$ for any $\eta > \tilde{\eta}$ as follows,

$$
\begin{aligned}
\hat{F}_{TbT}(\eta) = \frac{1}{m}\sum_{k=1}^m \frac{1}{2}\left\|w_{t,\eta}^{(k)} - w_{\text{train}}^{(k)}\right\|_{H_{\text{train}}^{(k)}}^2 &\geq \frac{1}{m}\sum_{k=1}^m \frac{1}{2}\left\|w_{t,\eta}^{(k)} - w_{\text{train}}^{(k)}\right\|_{H_{\text{train}}^{(k)}}^2 \mathbb{1}\left\{\mathcal{E}_1^{(k)} \cap \bar{\mathcal{E}}_2^{(k)}(\eta)\right\} \\
&\geq 2\sigma^2 \frac{1}{m}\sum_{k=1}^m \mathbb{1}\left\{\mathcal{E}_1^{(k)} \cap \bar{\mathcal{E}}_2^{(k)}(\eta)\right\} \\
&\geq 2\sigma^2 \frac{1}{m}\sum_{k=1}^m \mathbb{1}\left\{\mathcal{E}_1^{(k)} \cap \bar{\mathcal{E}}_2^{(k)}(\tilde{\eta})\right\} \geq \frac{\sigma^2}{10L^8},
\end{aligned}
$$

where the second inequality lower bounds the loss for one task by $2\sigma^2$ when the sequence gets truncated.

We have assumed $\mathbb{E}\frac{1}{2}\|w_{t,\tilde{\eta}} - w_{\text{train}}\|_{H_{\text{train}}}^2 \mathbb{1}\left\{\mathcal{E}_1 \cap \bar{\mathcal{E}}_2(\tilde{\eta})\right\} \geq \frac{\sigma^2}{L^6}$ in the proof. Now, we show the proof also works when $\mathbb{E}\frac{1}{2}\|w_{t,\tilde{\eta}} - w_{\text{train}}\|_{H_{\text{train}}}^2 \mathbb{1}\left\{\mathcal{E}_1 \cap \bar{\mathcal{E}}_2(\tilde{\eta})\right\} < \frac{\sigma^2}{L^6}$ with slight changes. According to the definition and Lemma 5, we know $\mathbb{E}\frac{1}{2}\|w_{t,\tilde{\eta}} - w_{\text{train}}\|_{H_{\text{train}}}^2 \mathbb{1}\left\{\mathcal{E}_1 \cap \bar{\mathcal{E}}_2(\eta)\right\} > \frac{\sigma^2}{L^6}$ for all $\eta > \tilde{\eta}$. At each training set $S_{\text{train}}$, we can define $\mathbb{1}\left\{\mathcal{E}_1 \cap \bar{\mathcal{E}}_2(\tilde{\eta}')\right\}$ as $\lim_{\eta \to \tilde{\eta}^+} \mathbb{1}\left\{\mathcal{E}_1 \cap \bar{\mathcal{E}}_2(\eta)\right\}$. We also have $\Pr[\mathcal{E}_1 \cap \bar{\mathcal{E}}_2(\tilde{\eta}')] \geq \frac{1}{18L^8}$. The remaining proof is the same as before as we substitute $\mathbb{1}\left\{\mathcal{E}_1 \cap \bar{\mathcal{E}}_2(\tilde{\eta})\right\}$ by $\mathbb{1}\left\{\mathcal{E}_1 \cap \bar{\mathcal{E}}_2(\tilde{\eta}')\right\}$. $\qed$

### B.2.3 GENERALIZATION FOR $\eta \in [1/L, \tilde{\eta}]$

In this section, we show empirical meta objective $\hat{F}_{TbT}$ is point-wise close to population meta objective $F_{TbT}$ for all $\eta \in [1/L, \tilde{\eta}]$.

**Lemma 4.** *Suppose $\sigma$ is a large constant $c_1$. Assume $t \geq c_2, d \geq c_4$ for certain constants $c_2, c_4$. With probability at least $1 - m\exp(-\Omega(d)) - O(t + m)\exp(-\Omega(m))$,*

$$|F_{TbT}(\eta) - \hat{F}_{TbT}(\eta)| \leq \frac{\sigma^2}{L^3},$$

*for all $\eta \in [1/L, \tilde{\eta}]$,*

In this section, we first show $\hat{F}_{TbT}$ concentrates on $F_{TbT}$ for any fixed $\eta$ and then construct $\epsilon$-net for $\hat{F}_{TbT}$ and $F_{TbT}$ for $\eta \in [1/L, \tilde{\eta}]$. We give the proof of Lemma 4 at the end.

We first show that for a fixed $\eta$, $\hat{F}_{TbT}(\eta)$ is close to $F_{TbT}(\eta)$ with high probability. We prove the meta-loss on each task $\Delta_{TbT}(\eta, P_k)$ is $O(1)$-subexponential. Then we apply Bernstein's inequality to get the result. The proof is deferred into Section B.2.4. We will assume $\sigma$ is a large constant and $t \geq c_2, d \geq c_4$ for some constants $c_2, c_4$ so that Lemma 6 holds and $\tilde{\eta}$ is a constant.

**Lemma 7.** *Suppose $\sigma$ is a constant. For any fixed $\eta$ and any $1 > \epsilon > 0$, with probability at least $1 - \exp(-\Omega(\epsilon^2 m))$,*

$$\left|\hat{F}_{TbT}(\eta) - F_{TbT}(\eta)\right| \leq \epsilon.$$

Next, we construct an $\epsilon$-net for $F_{TbT}$. By the definition of $\tilde{\eta}$, we know for any $\eta \leq \tilde{\eta}$, the contribution from truncated sequences in $F_{TbT}(\eta)$ is small. We can show the contribution from the un-truncated sequences is $O(t)$-lipschitz.

**Lemma 8.** *Suppose $\sigma$ is a large constant $c_1$. Assume $t \geq c_2, d \geq c_4$ for some constant $c_2, c_4$. There exists an $\frac{11\sigma^2}{L^4}$-net $N \subset [1/L, \tilde{\eta}]$ for $F_{TbT}$ with $|N| = O(t)$. That means, for any $\eta \in [1/L, \tilde{\eta}]$,*

$$|F_{TbT}(\eta) - F_{TbT}(\eta')| \leq \frac{11\sigma^2}{L^4},$$

*for $\eta' = \arg\min_{\eta'' \in N, \eta'' \leq \eta}(\eta - \eta'')$.*

**Proof of Lemma 8.** Let $\mathcal{E}_1$ and $\bar{\mathcal{E}}_2(\eta)$ be as defined in Definition 1. For the simplicity of the proof, we assume $\mathbb{E}\frac{1}{2}\|w_{t,\tilde{\eta}} - w_{\text{train}}\|^2_{H_{\text{train}}} \mathbb{1}\{\mathcal{E}_1 \cap \bar{\mathcal{E}}_2(\tilde{\eta})\} \leq \frac{\sigma^2}{L^6}$. We will discuss the proof for the other case at the end, which is very similar.

We can divide $\mathbb{E}\frac{1}{2}\|w_{t,\eta} - w_{\text{train}}\|^2_{H_{\text{train}}}$ as follows,

$$
\mathbb{E}\frac{1}{2}\|w_{t,\eta} - w_{\text{train}}\|^2_{H_{\text{train}}}
$$
$$
=\mathbb{E}\frac{1}{2}\|w_{t,\eta} - w_{\text{train}}\|^2_{H_{\text{train}}} \mathbb{1}\{\mathcal{E}_1 \cap \mathcal{E}_2(\tilde{\eta})\} + \mathbb{E}\frac{1}{2}\|w_{t,\eta} - w_{\text{train}}\|^2_{H_{\text{train}}} \mathbb{1}\{\mathcal{E}_1 \cap \bar{\mathcal{E}}_2(\tilde{\eta})\}
$$
$$
+ \mathbb{E}\frac{1}{2}\|w_{t,\eta} - w_{\text{train}}\|^2_{H_{\text{train}}} \mathbb{1}\{\bar{\mathcal{E}}_1\}.
$$

We will construct an $\epsilon$-net for the first term and show the other two terms are small. Let's first consider the third term. Since $\frac{1}{2}\|w_{t,\eta} - w_{\text{train}}\|^2_{H_{\text{train}}}$ is $O(1)$-subexponential and $\Pr[\bar{\mathcal{E}}_1] \leq \exp(-\Omega(d))$, we have $\mathbb{E}\frac{1}{2}\|w_{t,\eta} - w_{\text{train}}\|^2_{H_{\text{train}}} \mathbb{1}\{\bar{\mathcal{E}}_1\} = O(1)\exp(-\Omega(d))$. Choosing $d$ to be at least certain constant, we know $\frac{1}{2}\|w_{t,\eta} - w_{\text{train}}\|^2_{H_{\text{train}}} \mathbb{1}\{\bar{\mathcal{E}}_1\} \leq \sigma^2/L^4$.

Then we upper bound the second term. Since $\mathbb{E}\frac{1}{2}\|w_{t,\tilde{\eta}} - w_{\text{train}}\|^2_{H_{\text{train}}} \mathbb{1}\{\mathcal{E}_1 \cap \bar{\mathcal{E}}_2(\tilde{\eta})\} \leq \frac{\sigma^2}{L^6}$ and $\frac{1}{2}\|w_{t,\tilde{\eta}} - w_{\text{train}}\|^2_{H_{\text{train}}} \geq 2\sigma^2$ when $w_{t,\tilde{\eta}}$ diverges, we know $\Pr[\mathcal{E}_1 \cap \bar{\mathcal{E}}_2(\tilde{\eta})] \leq \frac{1}{2L^6}$. Then, we can upper bound the second term as follows,

$$\mathbb{E}\frac{1}{2}\|w_{t,\eta} - w_{\text{train}}\|^2_{H_{\text{train}}} \mathbb{1}\{\mathcal{E}_1 \cap \bar{\mathcal{E}}_2(\tilde{\eta})\} \leq 18L^2\sigma^2\frac{1}{2L^6} = \frac{9\sigma^2}{L^4}$$

Next, we show the first term $\frac{1}{2}\|w_{t,\eta} - w_{\text{train}}\|^2_{H_{\text{train}}} \mathbb{1}\{\mathcal{E}_1 \cap \mathcal{E}_2(\tilde{\eta})\}$ has desirable Lipschitz condition. According to Lemma 5, we know $\mathbb{1}\{\mathcal{E}_1 \cap \mathcal{E}_2(\eta)\} \geq \mathbb{1}\{\mathcal{E}_1 \cap \mathcal{E}_2(\tilde{\eta})\}$ for any $\eta \leq \tilde{\eta}$. Therefore, conditioning on $\mathcal{E}_1 \cap \mathcal{E}_2(\tilde{\eta})$, we know $w_{t,\eta}$ never gets truncated for any $\eta \leq \tilde{\eta}$. This means $w_{t,\eta} = B_{t,\eta}w_{\text{train}}$ with $B_{t,\eta} = (I - (I - \eta H_{\text{train}})^t)$. We can compute the derivative of $\frac{1}{2}\|w_{t,\eta} - w_{\text{train}}\|^2_{H_{\text{train}}}$ as follows,

$$\frac{\partial}{\partial \eta}\frac{1}{2}\|w_{t,\eta} - w_{\text{train}}\|^2_{H_{\text{train}}} = \langle tH_{\text{train}}(I - \eta H_{\text{train}})^{t-1}w_{\text{train}}, H_{\text{train}}(w_{t,\eta} - w_{\text{train}})\rangle.$$

Since $\|w_{t,\eta}\| = \|(I - (I - \eta H_{\text{train}})^t)w_{\text{train}}\| \leq 4\sqrt{L}\sigma$ and $\|w_{\text{train}}\| \leq 2\sqrt{L}\sigma$, we have $\|(I - \eta H_{\text{train}})^t w_{\text{train}}\| \leq 6\sqrt{L}\sigma$. We can bound $\|(I - \eta H_{\text{train}})^{t-1}w_{\text{train}}\|$ with $\|(I - \eta H_{\text{train}})^t w_{\text{train}}\| + \|w_{\text{train}}\|$ by bounding the expanding directions using $\|(I - \eta H_{\text{train}})^t w_{\text{train}}\|$ and bounding the shrinking directions using $\|w_{\text{train}}\|$. Therefore, we can bound the derivative as follows,

$$\left|\frac{\partial}{\partial \eta}\frac{1}{2}\|w_{t,\eta} - w_{\text{train}}\|^2_{H_{\text{train}}}\right| \leq tL \times 8\sqrt{L}\sigma \times 6L\sqrt{L}\sigma = 48L^3\sigma^2 t.$$

Suppose $\sigma$ is a constant, we know $\mathbb{E}\frac{1}{2}\|w_{t,\eta} - w_{\text{train}}\|^2_{H_{\text{train}}} \mathbb{1}\{\mathcal{E}_1 \cap \mathcal{E}_2(\tilde{\eta})\}$ is $O(t)$-lipschitz. Therefore, there exists an $\frac{\sigma^2}{L^4}$-net $N$ for $\mathbb{E}\frac{1}{2}\|w_{t,\eta} - w_{\text{train}}\|^2_{H_{\text{train}}} \mathbb{1}\{\mathcal{E}_1 \cap \mathcal{E}_2(\tilde{\eta})\}$ with size $O(t)$. That means, for any $\eta \in [1/L, \tilde{\eta}]$,

$$\left|\mathbb{E}\frac{1}{2}\|w_{t,\eta} - w_{\text{train}}\|^2_{H_{\text{train}}} \mathbb{1}\{\mathcal{E}_1 \cap \mathcal{E}_2(\tilde{\eta})\} - \mathbb{E}\frac{1}{2}\|w_{t,\eta'} - w_{\text{train}}\|^2_{H_{\text{train}}} \mathbb{1}\{\mathcal{E}_1 \cap \mathcal{E}_2(\tilde{\eta})\}\right| \leq \frac{\sigma^2}{L^4}$$

for $\eta' = \arg\min_{\eta'' \in N, \eta'' \leq \eta}(\eta - \eta'')$. Note we construct the $\epsilon$-net in a particular way such that $\eta'$ is chosen as the largest step size in $N$ that is at most $\eta$.

Combing with the upper bounds on the second term and the third term, we have for any $\eta \in [1/L, \tilde{\eta}]$,

$$|F_{TbT}(\eta) - F_{TbT}(\eta')| \leq \frac{11\sigma^2}{L^4}$$

for $\eta' = \arg\min_{\eta'' \in N, \eta'' \leq \eta}(\eta - \eta'')$.

In the above analysis, we have assumed $\mathbb{E}\frac{1}{2}\|w_{t,\tilde{\eta}} - w_{\text{train}}\|_{H_{\text{train}}}^2 \mathbb{1}\{\mathcal{E}_1 \cap \bar{\mathcal{E}}_2(\tilde{\eta})\} \leq \frac{\sigma^2}{L^6}$. The proof can be easily generalized to the other case. We can define $\mathbb{1}\{\mathcal{E}_1 \cap \bar{\mathcal{E}}_2(\tilde{\eta}')\}$ as $\lim_{\eta \to \tilde{\eta}^-} \mathbb{1}\{\mathcal{E}_1 \cap \bar{\mathcal{E}}_2(\eta)\}$. Then the proof works as long as we substitute $\mathbb{1}\{\mathcal{E}_1 \cap \bar{\mathcal{E}}_2(\tilde{\eta})\}$ by $\mathbb{1}\{\mathcal{E}_1 \cap \bar{\mathcal{E}}_2(\tilde{\eta}')\}$. We will also add $\tilde{\eta}$ into the $\epsilon$-net. $\qquad\square$

In order to prove $F_{TbT}$ is close to $\hat{F}_{TbT}$ point-wise in $[1/L, \tilde{\eta}]$, we still need to construct an $\epsilon$-net for the empirical meta objective $\hat{F}_{TbT}$.

**Lemma 9.** *Suppose $\sigma$ is a large constant $c_1$. Assume $t \geq c_2, d \geq c_4$ for certain constants $c_2, c_4$. With probability at least $1 - m\exp(-\Omega(d))$, there exists an $\frac{\sigma^2}{L^4}$-net $N' \subset [1/L, \tilde{\eta}]$ for $\hat{F}_{TbT}$ with $|N| = O(t + m)$. That means, for any $\eta \in [1/L, \tilde{\eta}]$,*

$$|\hat{F}_{TbT}(\eta) - \hat{F}_{TbT}(\eta')| \leq \frac{\sigma^2}{L^4},$$

*for $\eta' = \arg\min_{\eta'' \in N', \eta'' \leq \eta}(\eta - \eta'')$.*

**Proof of Lemma 9.** For each $k \in [m]$, let $\mathcal{E}_{1,k}$ be the event that $\sqrt{d}/\sqrt{L} \leq \sigma_i(X_{\text{train}}^{(k)}) \leq \sqrt{Ld}$ and $1/L \leq \lambda_i(H_{\text{train}}^{(k)}) \leq L$ for all $i \in [n]$ and $\sqrt{d}\sigma/4 \leq \left\|\xi_{\text{train}}^{(k)}\right\| \leq \sqrt{d}\sigma$. According to Lemma 1 and Lemma 45, we know with probability at least $1 - m\exp(-\Omega(d))$, $\mathcal{E}_{1,k}$'s hold for all $k \in [m]$. From now on, we assume all these events hold.

Recall that the empirical meta objective as follows,

$$\hat{F}_{TbT}(\eta) := \frac{1}{m} \sum_{k=1}^{m} \Delta_{TbT}(\eta, P_k).$$

For any $k \in [m]$, let $\eta_{c,k}$ be the smallest step size such that $w_{t,\eta}^{(k)}$ gets truncated. If $\eta_{c,k} > \hat{\eta}$, by similar argument as in Lemma 8, we know $\Delta_{TbT}(\eta, P_k)$ is $O(t)$-Lipschitz in $[1/L, \hat{\eta}]$ as long as $\sigma$ is a constant. If $\eta_{c,k} \leq \hat{\eta}$, by Lemma 5 we know $w_{t,\eta}^{(k)}$ gets truncated for any $\eta \geq \eta_{c,k}$. This then implies that $\Delta_{TbT}(\eta, P_k)$ is a constant function for $\eta \in [\eta_{c,k}, \hat{\eta}]$. We can also show that $\Delta_{TbT}(\eta, P_k)$ is $O(t)$-Lipschitz in $[1/L, \eta_{c,k})$. There might be a discontinuity in function value at $\eta_{c,k}$, so we need to add $\eta_{c,k}$ into the $\epsilon$-net.

Overall, we know there exists an $\frac{\sigma^2}{L^4}$-net $N'$ with $|N'| = O(t + m)$ for $\hat{F}_{TbT}$. That means, for any $\eta \in [1/L, \tilde{\eta}]$,

$$\left|\hat{F}_{TbT}(\eta) - \hat{F}_{TbT}(\eta')\right| \leq \frac{\sigma^2}{L^4}$$

for $\eta' = \arg\min_{\eta'' \in N', \eta'' \leq \eta}(\eta - \eta'')$. $\qquad\square$

Finally, we combine Lemma 7, Lemma 8 and Lemma 9 to prove that $\hat{F}_{TbT}$ is point-wise close to $F_{TbT}$ for $\eta \in [1/L, \tilde{\eta}]$.

**Proof of Lemma 4.** We assume $\sigma$ as a constant in this proof. By Lemma 7, we know with probability at least $1 - \exp(-\Omega(\epsilon^2 m))$, $\left|\hat{F}_{TbT}(\eta) - F_{TbT}(\eta)\right| \leq \epsilon$ for any fixed $\eta$. By Lemma 8, we know there exists an $\frac{11\sigma^2}{L^4}$-net $N$ for $F_{TbT}$ with size $O(t)$. By Lemma 9, we know with probability at least $1 - m\exp(-\Omega(d))$, there exists an $\frac{\sigma^2}{L^4}$-net $N'$ for $\hat{F}_{TbT}$ with size $O(t + m)$. According to the proofs

of Lemma 8 and Lemma 9, it's not hard to verify that $N \cup N'$ is still an $\frac{11\sigma^2}{L^4}$-net for $\hat{F}_{T b T}$ and $F_{T b T}$. That means, for any $\eta \in [1/L, \tilde{\eta}]$, we have

$$|F_{T b T}(\eta) - F_{T b T}(\eta')|, |\hat{F}_{T b T}(\eta) - \hat{F}_{T b T}(\eta')| \leq \frac{11\sigma^2}{L^4},$$

for $\eta' = \arg\min_{\eta'' \in N \cup N', \eta'' \leq \eta}(\eta - \eta'')$.

Taking a union bound over $N \cup N'$, we have with probability at least $1 - O(t + m)\exp(-\Omega(m))$,

$$\left| \hat{F}_{T b T}(\eta) - F_{T b T}(\eta) \right| \leq \frac{\sigma^2}{L^4}$$

for all $\eta \in N \cup N'$.

Overall, we know with probability at least $1 - m\exp(-\Omega(d)) - O(t + m)\exp(-\Omega(m))$, for all $\eta \in [1/L, \tilde{\eta}]$,

$$\begin{aligned}
&|F_{T b T}(\eta) - \hat{F}_{T b T}(\eta)| \\
\leq &|F_{T b T}(\eta) - F_{T b T}(\eta')| + |\hat{F}_{T b T}(\eta) - \hat{F}_{T b T}(\eta')| + |\hat{F}_{T b T}(\eta') - F_{T b T}(\eta')| \\
\leq &\frac{23\sigma^2}{L^4} \leq \frac{\sigma^2}{L^3},
\end{aligned}$$

where $\eta' = \arg\min_{\eta'' \in N \cup N', \eta'' \leq \eta}(\eta - \eta'')$. We use the fact that $L = 100$ in the last inequality. $\square$

### B.2.4 Proofs of Technical Lemmas

**Proof of Lemma 1.** Recall that $X_{\text{train}}$ is an $n \times d$ matix with $n = cd$ where $c \in [1/4, 3/4]$. According to Lemma 48, with probability at least $1 - 2\exp(-t^2/2)$, we have

$$\sqrt{d} - \sqrt{cd} - t \leq \sigma_i(X_{\text{train}}) \leq \sqrt{d} + \sqrt{cd} + t,$$

for all $i \in [n]$.

Since $H_{\text{train}} = 1/n X_{\text{train}}^\top X_{\text{train}}$, we know $\lambda_i(H_{\text{train}}) = 1/n\sigma_i^2(X_{\text{train}})$. Since $c \in [\frac{1}{4}, \frac{3}{4}]$, we have $\frac{1}{cd}(\sqrt{d} + \sqrt{cd})^2 \leq 100 - c'$ and $\frac{1}{cd}(\sqrt{d} - \sqrt{cd})^2 \geq \frac{1}{100} + c'$, for some constant $c'$. Therefore, we know with probability at least $1 - \exp(-\Omega(d))$,

$$\frac{1}{100} \leq \lambda_i(H_{\text{train}}) \leq 100,$$

for all $i \in [n]$.

Similarly, since there exists constant $c''$ such that $\sqrt{d} + \sqrt{cd} \leq (10 - c'')\sqrt{d}$ and $\sqrt{d} - \sqrt{cd} \geq (1/10 + c'')\sqrt{d}$, we know with probability at least $1 - \exp(-\Omega(d))$,

$$\frac{1}{10}\sqrt{d} \leq \sigma_i(X_{\text{train}}) \leq 10\sqrt{d},$$

for all $i \in [n]$. Choosing $L = 100$ finishes the proof. $\square$

**Proof of Lemma 5.** We prove that for any training set $S_{\text{train}}$, $\mathbb{1}\left\{\mathcal{E}_1 \cap \bar{\mathcal{E}}_2(\eta')\right\} \geq \mathbb{1}\left\{\mathcal{E}_1 \cap \bar{\mathcal{E}}_2(\eta')\right\}$ for any $\eta' > \eta$. This is trivially true if $\mathcal{E}_1$ is false on $S_{\text{train}}$. Therefore, we focus on the case when $\mathcal{E}_1$ holds for $S_{\text{train}}$. Suppose $\eta_c$ is the smallest step size such that the GD sequence gets truncated. Let $\{w'_{\tau, \eta_c}\}$ be the GD sequence without truncation. There must exists $\tau \leq t$ such that $\left\|w'_{\tau, \eta_c}\right\| \geq 4\sqrt{L}\sigma$. We only need to prove that $\left\|w'_{\tau, \eta}\right\| \geq 4\sqrt{L}\sigma$ for any $\eta \geq \eta_c$. We prove this by showing the derivative of $\left\|w'_{\tau, \eta}\right\|^2$ in $\eta$ is non-negative assuming $\left\|w'_{\tau, \eta}\right\|^2 \geq 4\sqrt{L}\sigma$.

Recall the recursion of $w'_{\tau, \eta}$ as $w'_{\tau, \eta} = w_{\text{train}} - (I - \eta H_{\text{train}})^\tau w_{\text{train}}$. If $\tau$ is an odd number, it's clear that $\frac{\partial}{\partial \eta}\left\|w'_{\tau, \eta}\right\|^2$ is non-negative at any $\eta \geq 0$. From now on, we assume $\tau$ is an even number. Actually in this case, $\frac{\partial}{\partial \eta}\left\|w'_{\tau, \eta}\right\|^2$ can be negative for some $\eta$. However, we can prove the derivative must be non-negative assuming $\left\|w'_{\tau, \eta}\right\|^2 \geq 4\sqrt{L}\sigma$.

Suppose the eigenvalue decomposition of $H_{\text{train}}$ is $\sum_{i=1}^{n} \lambda_i u_i u_i^\top$ with $\lambda_1 \geq \cdots \lambda_n$. Denote $c_i$ as $\langle w_{\text{train}}, u_i \rangle$. Let $\lambda_j$ be the smallest eigenvalue such that $(1 - \eta\lambda_j) \leq -1$. This implies $\lambda_i \leq 2/\eta$ for any $i \geq j+1$. We can write down $\left\|w'_{\tau,\eta}\right\|^2$ as follows

$$\left\|w'_{\tau,\eta}\right\|^2 = \sum_{i=1}^{j} \left(1 - (1 - \eta\lambda_i)^t\right)^2 c_i^2 + \sum_{i=j+1}^{n} \left(1 - (1 - \eta\lambda_i)^t\right)^2 c_i^2$$

$$\leq \sum_{i=1}^{j} \left(1 - (1 - \eta\lambda_i)^t\right)^2 c_i^2 + \left\|w_{\text{train}}\right\|^2.$$

Since $\mathcal{E}_1$ holds, we know $\left\|w_{\text{train}}\right\|^2 \leq 4L\sigma^2$. Combining with $\left\|w'_{\tau,\eta}\right\|^2 \geq 16L\sigma^2$, we have $\sum_{i=1}^{j} \left(1 - (1 - \eta\lambda_i)^t\right)^2 c_i^2 \geq 12L\sigma^2$. We can lower bound the derivative as follows,

$$\frac{\partial}{\partial \eta} \left\|w_{\tau,\eta}\right\|^2 = \sum_{i=1}^{j} 2t\lambda_i(1 - \eta\lambda_i)^{t-1} \left(1 - (1 - \eta\lambda_i)^t\right) c_i^2 + \sum_{i=j+1}^{n} 2t\lambda_i(1 - \eta\lambda_i)^{t-1} \left(1 - (1 - \eta\lambda_i)^t\right) c_i^2$$

$$\geq 2t \sum_{i=1}^{j} \lambda_i(1 - \eta\lambda_i)^{t-1} \left(1 - (1 - \eta\lambda_i)^t\right) c_i^2 - 2t\frac{2}{\eta} \sum_{i=j+1}^{n} c_i^2$$

$$\geq 2t \sum_{i=1}^{j} \lambda_i(1 - \eta\lambda_i)^{t-1} \left(1 - (1 - \eta\lambda_i)^t\right) c_i^2 - 2t \times 8L\sigma^2/\eta.$$

Then, we only need to show that $\sum_{i=1}^{j} \lambda_i(1 - \eta\lambda_i)^{t-1} \left(1 - (1 - \eta\lambda_i)^t\right) c_i^2$ is larger than $8L\sigma^2/\eta$. We have

$$\sum_{i=1}^{j} \lambda_i(1 - \eta\lambda_i)^{t-1} \left(1 - (1 - \eta\lambda_i)^t\right) c_i^2 = \sum_{i=1}^{j} \lambda_i \frac{(1 - \eta\lambda_i)^{t-1}}{1 - (1 - \eta\lambda_i)^t} \left(1 - (1 - \eta\lambda_i)^t\right)^2 c_i^2$$

$$= \sum_{i=1}^{j} \lambda_i \frac{(\eta\lambda_i - 1)^{t-1}}{(\eta\lambda_i - 1)^t - 1} \left(1 - (1 - \eta\lambda_i)^t\right)^2 c_i^2$$

$$= \sum_{i=1}^{j} \lambda_i \frac{(\eta\lambda_i - 1)^t}{(\eta\lambda_i - 1)^t - 1} \frac{1}{\eta\lambda_i - 1} \left(1 - (1 - \eta\lambda_i)^t\right)^2 c_i^2$$

$$\geq \sum_{i=1}^{j} \frac{1}{\eta} \left(1 - (1 - \eta\lambda_i)^t\right)^2 c_i^2 \geq 12L\sigma^2/\eta > 8L\sigma^2/\eta.$$

$\square$

**Proof of Lemma 6.** Similar as the analysis in Lemma 2, conditioning on $\mathcal{E}_1$, we know the GD sequence never exceeds the norm threshold for any $\eta \in [0, 2/L]$. This then implies

$$\mathbb{E}\frac{1}{2} \left\|w_{t,\eta} - w_{\text{train}}\right\|^2_{H_{\text{train}}} \mathbb{1}\left\{\mathcal{E}_1 \cap \bar{\mathcal{E}}_2(\eta)\right\} = 0,$$

for all $\eta \in [0, 2/L]$.

Let $\{w'_{\tau,\eta}\}$ be the GD sequence without truncation. For any step size $\eta \in [2.5L, \infty]$, conditioning on $\mathcal{E}_1$, we have

$$\left\|w'_{t,\eta}\right\| \geq \left((\eta/L - 1)^t - 1\right) \left\|w_{\text{train}}\right\| \geq \left(1.5^t - 1\right) \left(\frac{\sigma}{4\sqrt{L}} - 1\right) \geq 4\sqrt{L}\sigma,$$

where the last inequality holds as long as $\sigma \geq 5\sqrt{L}, t \geq c_2$ for some constant $c_2$. Therefore, we know when $\eta \in [2.5L, \infty), \mathbb{1}\left\{\mathcal{E}_1 \cap \bar{\mathcal{E}}_2(\eta)\right\} = \mathbb{1}\left\{\mathcal{E}_1\right\}$. Then, we have for any $\eta \geq 2.5L$,

$$\mathbb{E}\frac{1}{2} \left\|w_{t,\eta} - w_{\text{train}}\right\|^2_{H_{\text{train}}} \mathbb{1}\left\{\mathcal{E}_1 \cap \bar{\mathcal{E}}_2(\eta)\right\} \geq \frac{1}{2L} \left(4\sqrt{L}\sigma - 2\sqrt{L}\sigma\right)^2 \Pr[\mathcal{E}_1] \geq 2\sigma^2 \Pr[\mathcal{E}_1] \geq \frac{\sigma^2}{L^3},$$

where the last inequality uses $\Pr[\mathcal{E}_1] \geq 1 - \exp(-\Omega(d))$ and assume $d \geq c_4$ for some constant $c_4$.

Overall, we know $\mathbb{E}\frac{1}{2}\|w_{t,\eta} - w_{\text{train}}\|^2_{H_{\text{train}}} \mathbb{1}\left\{\mathcal{E}_1 \cap \bar{\mathcal{E}}_2(\eta)\right\}$ equals zero for all $\eta \in [0, 2/L]$ and is at least $\frac{\sigma^2}{L^3}$ for all $\eta \in [2.5L, \infty)$. By definition, we know $\tilde{\eta} \in (1/L, 3L)$. $\qquad\square$

**Proof of Lemma 7.** Recall that $\hat{F}_{TbT}(\eta) := \frac{1}{m}\sum_{k=1}^m \Delta_{TbT}(\eta, P_k)$. We prove that each $\Delta_{TbT}(\eta, P_k)$ is $O(1)$-subexponential. We can further write $\Delta_{TbT}(\eta, P_k)$ as follows,

$$
\begin{aligned}
\Delta_{TbT}(\eta, P_k) =& \frac{1}{2}\left\|w_{t,\eta}^{(k)} - w_k^* - (X_{\text{train}}^{(k)})^\dagger \xi_{\text{train}}^{(k)}\right\|^2_{H_{\text{train}}^{(k)}} \\
\leq& \frac{1}{2}\left\|w_{t,\eta}^{(k)} - w_k^*\right\|^2 \left\|H_{\text{train}}^{(k)}\right\| + \frac{1}{2n}\left\|\xi_{\text{train}}^{(k)}\right\|^2 + \left\|w_{t,\eta}^{(k)} - w_k^*\right\|\left(\frac{1}{\sqrt{n}}\left\|\xi_{\text{train}}^{(k)}\right\|\right)\left(\frac{1}{\sqrt{n}}\left\|X_{\text{train}}^{(k)}\right\|\right).
\end{aligned}
$$

We can write $\left\|H_{\text{train}}^{(k)}\right\|$ as $\sigma^2_{\max}(\frac{1}{\sqrt{n}}X_{\text{train}}^{(k)})$. According to Lemma 47, we know $\sigma_{\max}(X_{\text{train}}^{(k)}) - \mathbb{E}\sigma_{\max}(X_{\text{train}}^{(k)})$ is $O(1)$-subgaussian, which implies that $\sigma_{\max}(\frac{1}{\sqrt{n}}X_{\text{train}}^{(k)}) - \mathbb{E}\sigma_{\max}(\frac{1}{\sqrt{n}}X_{\text{train}}^{(k)})$ is $O(1/\sqrt{d})$-subgaussian. Since $\mathbb{E}\sigma_{\max}(\frac{1}{\sqrt{n}}X_{\text{train}}^{(k)})$ is a constant, we know $\sigma_{\max}(\frac{1}{\sqrt{n}}X_{\text{train}}^{(k)})$ is $O(1)$-subgaussian and $\sigma^2_{\max}(\frac{1}{\sqrt{n}}X_{\text{train}}^{(k)})$ is $O(1)$-subexponential. Similarly, we know both $\frac{1}{2n}\left\|\xi_{\text{train}}^{(k)}\right\|^2$ and $\left(\frac{1}{\sqrt{n}}\left\|X_{\text{train}}^{(k)}\right\|\right)\left(\frac{1}{\sqrt{n}}\left\|\xi_{\text{train}}^{(k)}\right\|\right)$ are $O(1)$-subexponential.

Suppose $\sigma$ is a constant, we know $\left\|w_{t,\eta}^{(k)} - w_k^*\right\|$ is upper bounded by a constant. Then, we know $\Delta_{TbT}(\eta, P_k)$ is $O(1)$-subexponential. Therefore, $\hat{F}_{TbT}(\eta)$ is the average of $m$ i.i.d. $O(1)$-subexponential random variables. By standard concentration inequality, we know for any $1 > \epsilon > 0$, with probability at least $1 - \exp(-\Omega(\epsilon^2 m))$,

$$
\left|\hat{F}_{TbT}(\eta) - F_{TbT}(\eta)\right| \leq \epsilon.
$$

$\qquad\square$

### B.3 TRAIN-BY-VALIDATION (GD)

In this section, we show that the optimal step size under $\hat{F}_{TbV}$ is $\Theta(1/t)$. Furthermore, we show under this optimal step size, GD sequence makes constant progress towards the ground truth. Precisely, we prove the following theorem.

**Theorem 8.** *Let the meta objective $\hat{F}_{TbV(n_1,n_2)}(\eta)$ be as defined in Equation 2 with $n_1, n_2 \in [d/4, 3d/4]$. Assume noise level $\sigma$ is a large constant $c_1$. Assume unroll length $t \geq c_2$, number of training tasks $m \geq c_3$ and dimension $d \geq c_4 \log(t)$ for certain constants $c_2, c_3, c_4$. With probability at least $0.99$ in the sampling of training tasks, we have*

$$
\eta_{valid}^* = \Theta(1/t) \text{ and } \mathbb{E}\left\|w_{t,\eta_{valid}^*} - w^*\right\|^2 = \|w^*\|^2 - \Omega(1)
$$

*for all $\eta_{valid}^* \in \arg\min_{\eta \geq 0} \hat{F}_{TbV(n_1,n_2)}(\eta)$, where the expectation is taken over new tasks.*

In this section, we still use $L$ to denote constant $100$. We start from analyzing the behavior of the population meta-objective $F_{TbV}$ for step sizes within $[0, 1/L]$. We show the optimal step size within this range is $\Theta(1/t)$ and GD sequence moves towards $w^*$ under the optimal step size. This serves as step 1 in Section B.1 We defer the proof of Lemma 10 into Section B.3.1.

**Lemma 10.** *Suppose noise level $\sigma$ is a large enough constant $c_1$. Assume unroll length $t \geq c_2$ and dimension $d \geq c_4$ for some constants $c_2, c_4$. There exist $\eta_1, \eta_2, \eta_3 = \Theta(1/t)$ with $\eta_1 < \eta_2 < \eta_3$ such that*

$$
F_{TbV}(\eta_2) \leq \frac{1}{2}\|w^*\|^2 - \frac{9}{10}C + \frac{\sigma^2}{2}
$$

$$
F_{TbV}(\eta) \geq \frac{1}{2}\|w^*\|^2 - \frac{6}{10}C + \frac{\sigma^2}{2}, \forall \eta \in [0, \eta_1] \cup [\eta_3, 1/L]
$$

*where $C$ is a positive constant.*

To relate the behavior of $F_{TbV}$ to the behavior of $\hat{F}_{TbV}$, we prove the following generalization result for step sizes in $[0, 1/L]$. This serves as step 3 in Section B.1. The proof is deferred into Section B.3.2.

**Lemma 11.** *For any $1 > \epsilon > 0$, assume $d \geq c_4 \log(1/\epsilon)$ for some constant $c_4$. With probability at least $1 - O(1/\epsilon) \exp(-\Omega(\epsilon^2 m))$,*

$$|\hat{F}_{TbV}(\eta) - F_{TbV}(\eta)| \leq \epsilon,$$

*for all $\eta \in [0, 1/L]$.*

In Lemma 12, we show the empirical meta objective $\hat{F}_{TbV}$ is high for all step size larger than $1/L$, which then implies $\eta^*_{\text{valid}} \in [0, 1/L]$. This serves as step 2 in Section B.1. We prove this lemma in Section B.3.3.

**Lemma 12.** *Suppose $\sigma$ is a large constant. Assume $t \geq c_2, d \geq c_4 \log(t)$ for some constants $c_2, c_4$. With probability at least $1 - \exp(-\Omega(m))$,*

$$\hat{F}_{TbV}(\eta) \geq C'\sigma^2 + \frac{1}{2}\sigma^2,$$

*for all $\eta \geq 1/L$, where $C'$ is a positive constant independent with $\sigma$.*

Combining Lemma 10, Lemma 11 and Lemma 12, we give the proof of Theorem 8.

**Proof of Theorem 8.** According to Lemma 10, we know as long as $d$ and $t$ are larger than certain constants, there exists $\eta_1, \eta_2, \eta_3 = \Theta(1/t)$ with $\eta_1 < \eta_2 < \eta_3$ such that

$$F_{TbV}(\eta_2) \leq \frac{1}{2}\|w^*\|^2 - \frac{9}{10}C + \sigma^2/2$$

$$F_{TbV}(\eta) \geq \frac{1}{2}\|w^*\|^2 - \frac{6}{10}C + \sigma^2/2, \forall \eta \in [0, \eta_1] \cup [\eta_3, 1/L],$$

for some positive constant $C$.

Choosing $\epsilon = \min(1, C/10)$ in Lemma 11, we know as long as $d$ is larger than certain constant, with probability at least $1 - \exp(-\Omega(m))$,

$$|\hat{F}_{TbV}(\eta) - F_{TbV}(\eta)| \leq C/10,$$

for all $\eta \in [0, 1/L]$.

Therefore,

$$\hat{F}_{TbV}(\eta_2) \leq \frac{1}{2}\|w^*\|^2 - \frac{8}{10}C + \sigma^2/2$$

$$\hat{F}_{TbV}(\eta) \geq \frac{1}{2}\|w^*\|^2 - \frac{7}{10}C + \sigma^2/2, \forall \eta \in [0, \eta_1] \cup [\eta_3, 1/L].$$

By Lemma 12, we know as long as $t \geq c_2, d \geq c_4 \log(t)$ for some constants $c_2, c_4$, with probability at least $1 - \exp(-\Omega(m))$,

$$\hat{F}_{TbV}(\eta) \geq C'\sigma^2 + \frac{1}{2}\sigma^2,$$

for all $\eta \geq 1/L$. As long as $\sigma \geq 1/\sqrt{C'}$, we have $\hat{F}_{TbV}(\eta) \geq 1 + \frac{1}{2}\sigma^2$ for all $\eta \geq 1/L$. Combining with $\hat{F}_{TbV}(\eta_2) \leq \frac{1}{2}\|w^*\|^2 - \frac{8}{10}C + \sigma^2/2$, we know $\eta^*_{\text{valid}} \in [0, 1/L]$. Furthermore, since $\hat{F}_{TbV}(\eta) \geq \frac{1}{2}\|w^*\|^2 - \frac{7}{10}C + \sigma^2/2, \forall \eta \in [0, \eta_1] \cup [\eta_3, 1/L]$, we have $\eta_1 \leq \eta^*_{\text{valid}} \leq \eta_3$.

Recall that $\eta_1, \eta_3 = \Theta(1/t)$, we know $\eta^*_{\text{valid}} = \Theta(1/t)$. At the optimal step size, we have

$$F_{TbV}(\eta^*_{\text{valid}}) \leq \hat{F}_{TbV}(\eta^*_{\text{valid}}) + C/10 \leq \hat{F}_{TbV}(\eta_2) + C/10 \leq \frac{1}{2}\|w^*\|^2 - \frac{7}{10}C + \sigma^2/2.$$

Since $F_{TbV}(\eta^*_{\text{valid}}) = \mathbb{E}\frac{1}{2}\left\|w_{t,\eta^*_{\text{valid}}} - w^*\right\|^2 + \sigma^2/2$, we have

$$\mathbb{E}\left\|w_{t,\eta^*_{\text{valid}}} - w^*\right\|^2 \leq \|w^*\|^2 - \frac{7}{5}C.$$

Choosing $m$ to be at least certain constant, this holds with probability at least 0.99. $\qquad\square$

### B.3.1 BEHAVIOR OF $F_{TbV}$ FOR $\eta \in [0, 1/L]$

In this section, we study the behavior of $F_{TbV}$ when $\eta \in [0, 1/L]$. We prove the following Lemma.

**Lemma 10.** *Suppose noise level $\sigma$ is a large enough constant $c_1$. Assume unroll length $t \geq c_2$ and dimension $d \geq c_4$ for some constants $c_2, c_4$. There exist $\eta_1, \eta_2, \eta_3 = \Theta(1/t)$ with $\eta_1 < \eta_2 < \eta_3$ such that*

$$F_{TbV}(\eta_2) \leq \frac{1}{2} \|w^*\|^2 - \frac{9}{10}C + \frac{\sigma^2}{2}$$

$$F_{TbV}(\eta) \geq \frac{1}{2} \|w^*\|^2 - \frac{6}{10}C + \frac{\sigma^2}{2}, \forall \eta \in [0, \eta_1] \cup [\eta_3, 1/L]$$

*where $C$ is a positive constant.*

It's not hard to verify that $F_{TbV}(\eta) = \mathbb{E}1/2 \|w_{t,\eta} - w^*\|^2 + \sigma^2/2$. For convenience, denote $Q(\eta) := 1/2 \|w_{t,\eta} - w^*\|^2$. In order to prove Lemma 10, we only need to show that $\mathbb{E}Q(\eta_2) \leq \frac{1}{2} \|w^*\|^2 - \frac{9}{10}C$ and $\mathbb{E}Q(\eta) \geq \frac{1}{2} \|w^*\|^2 - \frac{6}{10}C$ for all $\eta \in [0, \eta_1] \cup [\eta_3, 1/L]$. In Lemma 13, we first show that this happens with high probability over the sampling of tasks.

**Lemma 13.** *Suppose noise level $\sigma$ is a large enough constant $c_1$. Assume unroll length $t \geq c_2$ for certain constant $c_2$. Then, with probability at least $1 - \exp(-\Omega(d))$ over the sampling of tasks, there exists $\eta_1, \eta_2, \eta_3 = \Theta(1/t)$ with $\eta_1 < \eta_2 < \eta_3$ such that*

$$Q(\eta_2) := \frac{1}{2} \|w_{t,\eta_2} - w^*\|^2 \leq \frac{1}{2} \|w^*\|^2 - C$$

$$Q(\eta) := \frac{1}{2} \|w_{t,\eta} - w^*\|^2 \geq \frac{1}{2} \|w^*\|^2 - \frac{C}{2}, \forall \eta \in [0, \eta_1] \cup [\eta_3, 1/L]$$

*where $C$ is a positive constant.*

Since we are in the small step size regime, we know the GD sequence converges with high probability and will not be truncated. For now, let's assume $w_{t,\eta} = B_{t,\eta} w^*_{\text{train}} + B_{t,\eta}(X_{\text{train}})^\dagger \xi_{\text{train}}$, where $B_{t,\eta} = I - (I - \eta H_{\text{train}})^t$. We have

$$Q(\eta) = \frac{1}{2} \left\| B_{t,\eta} w^*_{\text{train}} + B_{t,\eta}(X_{\text{train}})^\dagger \xi_{\text{train}} - w^* \right\|^2$$

$$= \frac{1}{2} \|B_{t,\eta} w^*_{\text{train}} - w^*\|^2 + \frac{1}{2} \left\| B_{t,\eta}(X_{\text{train}})^\dagger \xi_{\text{train}} \right\|^2$$

$$+ \left\langle B_{t,\eta} w^*_{\text{train}} - w^*, B_{t,\eta}(X_{\text{train}})^\dagger \xi_{\text{train}} \right\rangle$$

$$= \frac{1}{2} \|w^*\|^2 + \frac{1}{2} \|B_{t,\eta} w^*_{\text{train}}\|^2 + \frac{1}{2} \left\| B_{t,\eta}(X_{\text{train}})^\dagger \xi_{\text{train}} \right\|^2 - \left\langle B_{t,\eta} w^*_{\text{train}}, w^* \right\rangle$$

$$+ \left\langle B_{t,\eta} w^*_{\text{train}} - w^*, B_{t,\eta}(X_{\text{train}})^\dagger \xi_{\text{train}} \right\rangle.$$

In Lemma 14, we show that with high probability the crossing term $\left\langle B_{t,\eta} w^*_{\text{train}} - w^*, B_{t,\eta}(X_{\text{train}})^\dagger \xi_{\text{train}} \right\rangle$ is negligible for all $\eta \in [0, 1/L]$. By Hoeffding's inequality, we know the crossing term is small for any fixed $\eta$. Constructing an $\epsilon$-net for the crossing term in $\eta$, we can take a union bound and show it's small for all $\eta \in [0, 1/L]$. We defer the proof of Lemma 14 to Section B.3.4.

**Lemma 14.** *Assume $\sigma$ is a constant. For any $1 > \epsilon > 0$, we know with probability at least $1 - O(1/\epsilon) \exp(-\Omega(\epsilon^2 d))$,*

$$\left| \left\langle B_{t,\eta} w^*_{train} - w^*, B_{t,\eta}(X_{train})^\dagger \xi_{train} \right\rangle \right| \leq \epsilon,$$

*for all $\eta \in [0, 1/L]$.*

Denote

$$G(\eta) := \frac{1}{2} \|w^*\|^2 + \frac{1}{2} \|B_{t,\eta} w^*_{\text{train}}\|^2 + \frac{1}{2} \left\| B_{t,\eta}(X_{\text{train}})^\dagger \xi_{\text{train}} \right\|^2 - \left\langle B_{t,\eta} w^*_{\text{train}}, w^* \right\rangle.$$

Choosing $\epsilon = C/4$ in Lemma 14, we only need to show $G(\eta_2) \leq \|w^*\|^2 - 5C/4$ and $G(\eta) \geq \|w^*\|^2 - C/4$ for all $\eta \in [0, \eta_1] \cup [\eta_3, 1/L]$.

We first show that there exists $\eta_2 = \Theta(1/t)$ such that $G(\eta_2) \leq \frac{1}{2} \|w^*\|^2 - 5C/4$ for some constant $C$. It's not hard to show that $\frac{1}{2} \|B_{t,\eta} w^*_{\text{train}}\|^2 + \frac{1}{2} \|B_{t,\eta}(X_{\text{train}})^\dagger \xi_{\text{train}}\|^2 = O(\eta^2 t^2 \sigma^2)$. In Lemma 15, we show that the improvement $\langle B_{t,\eta} w^*_{\text{train}}, w^* \rangle = \Omega(\eta t)$ is linear in $\eta$. Therefore there exists $\eta_2 = \Theta(1/t)$ such that $G(\eta_2) \leq \frac{1}{2} \|w^*\|^2 - 5C/4$ for some constant $C$. We defer the proof of Lemma 15 to Section B.3.4.

**Lemma 15.** *For any fixed $\eta \in [0, L/t]$ with probability at least $1 - \exp(-\Omega(d))$,*

$$\langle B_{t,\eta} w^*_{train}, w^* \rangle \geq \frac{\eta t}{16L}.$$

To lower bound $G(\eta)$ for small $\eta$, we notice

$$G(\eta) \geq \frac{1}{2} \|w^*\|^2 - \langle B_{t,\eta} w^*_{\text{train}}, w^* \rangle.$$

We can show that $\langle B_{t,\eta} w^*_{\text{train}}, w^* \rangle = O(\eta t)$. Therefore, there exists $\eta_1 = \Theta(1/t)$ such that $\langle B_{t,\eta} w^*_{\text{train}}, w^* \rangle \leq C/4$ for all $\eta \in [0, \eta_1]$.

To lower bound $G(\eta)$ for large $\eta$, we lower bound $G(\eta)$ using the noise square term,

$$G(\eta) \geq \frac{1}{2} \|B_{t,\eta}(X_{\text{train}})^\dagger \xi_{\text{train}}\|^2.$$

We show that with high probability $\|B_{t,\eta}(X_{\text{train}})^\dagger \xi_{\text{train}}\|^2 = \Omega(\sigma^2)$ for all $\eta \in [\log(2)L/t, 1/L]$. Therefore, as long as $\sigma$ is larger than some constant, there exists $\eta_3 = \Theta(1/t)$ such that $G(\eta) \geq \frac{1}{2} \|w^*\|^2$ for all $\eta \in [\eta_3, 1/L]$.

Combing Lemma 14 and Lemma 15, we give a complete proof for Lemma 13.

**Proof of Lemma 13.** Recall that

$$Q(\eta) = \frac{1}{2} \|B_{t,\eta} w^*_{\text{train}} - w^*\|^2 + \frac{1}{2} \|B_{t,\eta}(X_{\text{train}})^\dagger \xi_{\text{train}}\|^2$$
$$+ \langle B_{t,\eta} w^*_{\text{train}} - w^*, B_{t,\eta}(X_{\text{train}})^\dagger \xi_{\text{train}} \rangle$$
$$= G(\eta) + \langle B_{t,\eta} w^*_{\text{train}} - w^*, B_{t,\eta}(X_{\text{train}})^\dagger \xi_{\text{train}} \rangle$$

We first show that with probability at least $1 - \exp(-\Omega(d))$, there exist $\eta_1, \eta_2, \eta_3 = \Theta(1/t)$ with $\eta_1 < \eta_2 < \eta_3$ such that $G(\eta_2) \leq 1/2 \|w^*\|^2 - 5C/4$ and $G(\eta) \geq 1/2 \|w^*\|^2 - C/4$ for all $\eta \in [0, \eta_1] \cup [\eta_3, 1/L]$.

According to Lemma 1, we know with probability at least $1 - \exp(-\Omega(d))$, $\sqrt{d}/\sqrt{L} \leq \sigma_i(X_{\text{train}}) \leq \sqrt{Ld}$ and $1/L \leq \lambda_i(H_{\text{train}}) \leq L$ for all $i \in [n]$ with $L = 100$.

**Upper bounding $G(\eta_2)$:** We can expand $G(\eta)$ as follows:

$$G(\eta) := \frac{1}{2} \|B_{t,\eta} w^*_{\text{train}} - w^*\|^2 + \frac{1}{2} \|B_{t,\eta}(X_{\text{train}})^\dagger \xi_{\text{train}}\|^2$$
$$= \frac{1}{2} \|w^*\|^2 + \frac{1}{2} \|B_{t,\eta} w^*_{\text{train}}\|^2 + \frac{1}{2} \|B_{t,\eta}(X_{\text{train}})^\dagger \xi_{\text{train}}\|^2 - \langle B_{t,\eta} w^*_{\text{train}}, w^* \rangle.$$

Recall that $B_{t,\eta} = I - (I - \eta H_{\text{train}})^t$, for any vector $w$ in the span of $H_{\text{train}}$,

$$\|B_{t,\eta} w\| = \|(I - (I - \eta H_{\text{train}})^t) w\| \leq L\eta t \|w\|.$$

According to Lemma 45, we know with probability at least $1 - \exp(-\Omega(d))$, $\|\xi_{\text{train}}\| \leq \sqrt{d}\sigma$. Therefore, we have

$$\frac{1}{2} \|B_{t,\eta} w^*_{\text{train}}\|^2 + \frac{1}{2} \|B_{t,\eta}(X_{\text{train}})^\dagger \xi_{\text{train}}\|^2 \leq L^2 \eta^2 t^2/2 + L^3 \eta^2 t^2 \sigma^2/2 \leq L^3 \eta^2 t^2 \sigma^2,$$

where the second inequality uses $\sigma, L \geq 1$. According to Lemma 15, for any fixed $\eta \in [0, L/t]$, with probability at least $1 - \exp(-\Omega(d))$, $\langle B_{t,\eta} w^*_{\text{train}}, w^* \rangle \geq \frac{\eta t}{16L}$. Therefore,

$$G(\eta) \leq \frac{1}{2} \|w^*\|^2 + L^3 \eta^2 t^2 \sigma^2 - \frac{\eta t}{16L} \leq \frac{1}{2} \|w^*\|^2 - \frac{\eta t}{32L},$$

where the second inequality holds as long as $\eta \leq \frac{1}{32L^4\sigma^2 t}$. Choosing $\eta_2 := \frac{1}{32L^4\sigma^2 t}$, we have

$$G(\eta_2) \leq \frac{1}{2}\left\|w^*\right\|^2 - \frac{1}{1024L^5\sigma^2} = \frac{1}{2}\left\|w^*\right\|^2 - \frac{5C}{4},$$

where $C = \frac{1}{819.2L^5\sigma^2}$. Note $C$ is a constant as $\sigma, L$ are constants.

**Lower bounding $G(\eta)$ for $\eta \in [0, \eta_1]$ :** Now, we prove that there exists $\eta_1 = \Theta(1/t)$ with $\eta_1 < \eta_2$ such that for any $\eta \in [0, \eta_1]$, $G(\eta) \geq \frac{1}{2}\left\|w^*\right\|^2 - \frac{C}{4}$. Recall that

$$G(\eta) = \frac{1}{2}\left\|w^*\right\|^2 + \frac{1}{2}\left\|B_{t,\eta}w^*_{\text{train}}\right\|^2 + \frac{1}{2}\left\|B_{t,\eta}(X_{\text{train}})^\dagger\xi_{\text{train}}\right\|^2 - \left\langle B_{t,\eta}w^*_{\text{train}}, w^*\right\rangle.$$

$$\geq \frac{1}{2}\left\|w^*\right\|^2 - \left\langle B_{t,\eta}w^*_{\text{train}}, w^*\right\rangle.$$

Since $\left|\left\langle B_{t,\eta}w^*_{\text{train}}, w^*\right\rangle\right| \leq L\eta t$, we know for any $\eta \in [0, \eta_1]$,

$$G(\eta) \geq \frac{1}{2}\left\|w^*\right\|^2 - L\eta_1 t.$$

Choosing $\eta_1 = \frac{C}{4Lt}$, we have for any $\eta \in [0, \eta_1]$,

$$G(\eta) \geq \frac{1}{2}\left\|w^*\right\|^2 - \frac{C}{4}.$$

**Lower bounding $G(\eta)$ for $\eta \in [\eta_3, 1/L]$:** Now, we prove that there exists $\eta_3 = \Theta(1/t)$ with $\eta_3 > \eta_2$ such that for all $\eta \in [\eta_3, 1/L]$,

$$G(\eta) \geq \frac{1}{2}\left\|w^*\right\|^2 - \frac{C}{4}.$$

Recall that

$$G(\eta) = \frac{1}{2}\left\|B_{t,\eta}w^*_{\text{train}} - w^*\right\|^2 + \frac{1}{2}\left\|B_{t,\eta}(X_{\text{train}})^\dagger\xi_{\text{train}}\right\|^2 \geq \frac{1}{2}\left\|B_{t,\eta}(X_{\text{train}})^\dagger\xi_{\text{train}}\right\|^2.$$

According to Lemma 45, we know with probability at least $1 - \exp(-\Omega(d))$, $\frac{\sqrt{d}\sigma}{2\sqrt{2}} \leq \left\|\xi_{\text{train}}\right\|$. Therefore,

$$\left\|B_{t,\eta}(X_{\text{train}})^\dagger\xi_{\text{train}}\right\|^2 \geq \left(1 - e^{-\eta t/L}\right)^2 \frac{\sigma^2}{8L} \geq \frac{\sigma^2}{32L},$$

where the last inequality assumes $\eta \geq \log(2)L/t$. As long as $t \geq \log(2)L^2$, we have $\log(2)L/t \leq 1/L$. Choosing $\eta_3 = \log(2)L/t$, we know for all $\eta \in [\eta_3, 1/L]$,

$$G(\eta) \geq \frac{1}{2}\left\|B_{t,\eta}(X_{\text{train}})^\dagger\xi_{\text{train}}\right\|^2 \geq \frac{\sigma^2}{64L}.$$

Note that $\frac{1}{2}\left\|w^*\right\|^2 = 1/2$. Therefore, as long as $\sigma \geq 8\sqrt{L}$, we have

$$G(\eta) \geq \frac{1}{2}\left\|w^*\right\|^2$$

for all $\eta \in [\eta_3, 1/L]$.

Overall, we have shown that there exist $\eta_1, \eta_2, \eta_3 = \Theta(1/t)$ with $\eta_1 < \eta_2 < \eta_3$ such that $G(\eta_2) \leq 1/2\left\|w^*\right\|^2 - 5C/4$ and $G(\eta) \geq 1/2\left\|w^*\right\|^2 - C/4$ for all $\eta \in [0, \eta_1] \cup [\eta_3, 1/L]$. Recall that $Q(\eta) = G(\eta) + \left\langle B_{t,\eta}w^*_{\text{train}} - w^*, B_{t,\eta}(X_{\text{train}})^\dagger\xi_{\text{train}}\right\rangle$. Choosing $\epsilon = C/4$ in Lemma 14, we know with probability at least $1 - \exp(-\Omega(d))$, $\left|\left\langle B_{t,\eta}w^*_{\text{train}} - w^*, B_{t,\eta}(X_{\text{train}})^\dagger\xi_{\text{train}}\right\rangle\right| \leq C/4$ for all $\eta \in [0, 1/L]$. Therefore, we know $Q(\eta_2) \leq 1/2\left\|w^*\right\|^2 - C$ and $Q(\eta) \geq 1/2\left\|w^*\right\|^2 - C/2$ for all $\eta \in [0, \eta_1] \cup [\eta_3, 1/L]$. $\qquad\square$

Next, we give the proof of Lemma 10.

**Proof of Lemma 10.** Recall that $F_{TbV}(\eta) = \mathbb{E}1/2\|w_{t,\eta} - w^*\|^2 + \frac{\sigma^2}{2}$. For convenience, denote $Q(\eta) := 1/2\|w_{t,\eta} - w^*\|^2$. In order to prove Lemma 10, we only need to show that $\mathbb{E}Q(\eta_2) \leq \frac{1}{2}\|w^*\|^2 - \frac{9}{10}C$ and $\mathbb{E}Q(\eta) \geq \frac{1}{2}\|w^*\|^2 - \frac{6}{10}C$ for all $\eta \in [0, \eta_1] \cup [\eta_3, 1/L]$.

According to Lemma 13, as long as $\sigma$ is a large enough constant $c_1$ and $t$ is at least certain constant $c_2$, with probability at least $1 - \exp(-\Omega(d))$ over the sampling of $S_{\text{train}}$, there exists $\eta_1, \eta_2, \eta_3 = \Theta(1/t)$ with $\eta_1 < \eta_2 < \eta_3$ such that

$$Q(\eta_2) := 1/2\|w_{t,\eta_2} - w^*\|^2 \leq \frac{1}{2}\|w^*\|^2 - C$$

$$Q(\eta) := 1/2\|w_{t,\eta} - w^*\|^2 \geq \frac{1}{2}\|w^*\|^2 - \frac{C}{2}, \forall \eta \in [0, \eta_1] \cup [\eta_3, 1/L]$$

where $C$ is a positive constant. Call this event $\mathcal{E}$. Suppose the probability that $\mathcal{E}$ happens is $1 - \delta$. We can write $\mathbb{E}Q(\eta)$ as follows,

$$\mathbb{E}Q(\eta) = \mathbb{E}[Q(\eta)|\mathcal{E}]\Pr[\mathcal{E}] + \mathbb{E}[Q(\eta)|\bar{\mathcal{E}}]\Pr[\bar{\mathcal{E}}].$$

According to the algorithm, we know $\|w_{t,\eta}\|$ is always bounded by $4\sqrt{L}\sigma$. Therefore, $Q(\eta) := 1/2\|w_{t,\eta} - w^*\|^2 \leq 13L\sigma^2$. When $\eta = \eta_2$, we have

$$\mathbb{E}Q(\eta_2) \leq \left(\frac{1}{2}\|w^*\|^2 - C\right)(1 - \delta) + 13L\sigma^2\delta$$

$$= \frac{1}{2}\|w^*\|^2 - \frac{\delta}{2} - C + (C + 13L\sigma^2)\delta$$

$$\leq \frac{1}{2}\|w^*\|^2 - \frac{9C}{10},$$

where the last inequality assumes $\delta \leq \frac{C}{10C + 130L\sigma^2}$.

When $\eta \in [0, \eta_1] \cup [\eta_3, 1/L]$, we have

$$\mathbb{E}Q(\eta_2) \geq \left(\frac{1}{2}\|w^*\|^2 - \frac{C}{2}\right)(1 - \delta) - 13L\sigma^2\delta$$

$$= \frac{1}{2}\|w^*\|^2 - \frac{\delta}{2} - (1 - \delta)\frac{C}{2} - 13L\sigma^2\delta$$

$$\geq \frac{1}{2}\|w^*\|^2 - \frac{C}{2} - (1/2 + 13L\sigma^2)\delta$$

$$\geq \frac{1}{2}\|w^*\|^2 - \frac{6C}{10},$$

where the last inequality holds as long as $\delta \leq \frac{C}{5C + 130L\sigma^2}$.

According to Lemma 13, we know $\delta \leq \exp(-\Omega(d))$. Therefore, the conditions for $\delta$ can be satisfied as long as $d$ is larger than certain constant. $\square$

### B.3.2 GENERALIZATION FOR $\eta \in [0, 1/L]$

In this section, we show $\hat{F}_{TbV}$ is point-wise close to $F_{TbV}$ for all $\eta \in [0, 1/L]$. Recall Lemma 11 as follows.

**Lemma 11.** *For any $1 > \epsilon > 0$, assume $d \geq c_4 \log(1/\epsilon)$ for some constant $c_4$. With probability at least $1 - O(1/\epsilon)\exp(-\Omega(\epsilon^2 m))$,*

$$|\hat{F}_{TbV}(\eta) - F_{TbV}(\eta)| \leq \epsilon,$$

*for all $\eta \in [0, 1/L]$.*

In order to prove Lemma 11, let's first show that for a fixed $\eta$ with high probability $\hat{F}_{TbV}(\eta)$ is close to $F_{TbV}(\eta)$. Similar as in Lemma 7, we show each $\Delta_{TbV}(\eta, P_k)$ is $O(1)$-subexponential. We defer its proof to Section B.3.4.

**Lemma 16.** *Suppose $\sigma$ is a constant. For any fixed $\eta \in [0, 1/L]$ and any $1 > \epsilon > 0$, with probability at least $1 - \exp(-\Omega(\epsilon^2 m))$,*

$$\left| \hat{F}_{TbV}(\eta) - F_{TbV}(\eta) \right| \leq \epsilon.$$

Next, we show that there exists an $\epsilon$-net for $F_{TbV}$ with size $O(1/\epsilon)$. By $\epsilon$-net, we mean there exists a finite set $N_\epsilon$ of step size such that $|F_{TbV}(\eta) - F_{TbV}(\eta')| \leq \epsilon$ for any $\eta \in [0, 1/L]$ and $\eta' \in \arg\min_{\eta \in N_\epsilon} |\eta - \eta'|$. We defer the proof of Lemma 17 to Section B.3.4.

**Lemma 17.** *Suppose $\sigma$ is a constant. For any $1 > \epsilon > 0$, assume $d \geq c_4 \log(1/\epsilon)$ for constant $c_4$. There exists an $\epsilon$-net $N_\epsilon$ for $F_{TbV}$ with $|N_\epsilon| = O(1/\epsilon)$. That means, for any $\eta \in [0, 1/L]$,*

$$|F_{TbV}(\eta) - F_{TbV}(\eta')| \leq \epsilon,$$

*for $\eta' \in \arg\min_{\eta \in N_\epsilon} |\eta - \eta'|$.*

Next, we show that with high probability, there also exists an $\epsilon$-net for $\hat{F}_{TbV}$ with size $O(1/\epsilon)$.

**Lemma 18.** *Suppose $\sigma$ is a constant. For any $1 > \epsilon > 0$, assume $d \geq c_4 \log(1/\epsilon)$ for constant $c_4$. With probability at least $1 - \exp(-\Omega(\epsilon^2 m))$, there exists an $\epsilon$-net $N'_\epsilon$ for $\hat{F}_{TbV}$ with $|N_\epsilon| = O(1/\epsilon)$. That means, for any $\eta \in [0, 1/L]$,*

$$|\hat{F}_{TbV}(\eta) - \hat{F}_{TbV}(\eta')| \leq \epsilon,$$

*for $\eta' \in \arg\min_{\eta \in N_\epsilon} |\eta - \eta'|$.*

Combing Lemma 16, Lemma 17 and Lemma 18, now we give the proof of Lemma 11.

**Proof of Lemma 11.** The proof is very similar as in Lemma 4. By Lemma 16, we know with probability at least $1 - \exp(-\Omega(\epsilon^2 m))$, $\left| \hat{F}_{TbV}(\eta) - F_{TbV}(\eta) \right| \leq \epsilon$ for any fixed $\eta$. By Lemma 17 and Lemma 18, we know as long as $d = \Omega(\log(1/\epsilon))$, with probability at least $1 - \exp(-\Omega(\epsilon^2 m))$, there exists $\epsilon$-net $N_\epsilon$ and $N'_\epsilon$ for $F_{TbV}$ and $\hat{F}_{TbV}$ respectively. Here, both of $N_\epsilon$ and $N'_\epsilon$ have size $O(1/\epsilon)$. According to the proofs of Lemma 17 and Lemma 18, it's not hard to verify that $N_\epsilon \cup N'_\epsilon$ is still an $\epsilon$-net for $\hat{F}_{TbV}$ and $F_{TbV}$. That means, for any $\eta \in [0, 1/L]$, we have

$$|F_{TbV}(\eta) - F_{TbV}(\eta')|, |\hat{F}_{TbV}(\eta) - \hat{F}_{TbV}(\eta')| \leq \epsilon,$$

for $\eta' \in \arg\min_{\eta \in N_\epsilon \cup N'_\epsilon} |\eta - \eta'|$.

Taking a union bound over $N_\epsilon \cup N'_\epsilon$, we have with probability at least $1 - O(1/\epsilon) \exp(-\Omega(\epsilon^2 m))$,

$$\left| \hat{F}_{TbV}(\eta) - F_{TbV}(\eta) \right| \leq \epsilon$$

for any $\eta \in N_\epsilon \cup N'_\epsilon$.

Overall, we know with probability at least $1 - O(1/\epsilon) \exp(-\Omega(\epsilon^2 m))$, for all $\eta \in [0, 1/L]$,

$$
\begin{aligned}
&|F_{TbV}(\eta) - \hat{F}_{TbV}(\eta)| \\
\leq &|F_{TbV}(\eta) - F_{TbV}(\eta')| + |\hat{F}_{TbV}(\eta) - \hat{F}_{TbV}(\eta')| + |\hat{F}_{TbV}(\eta') - F_{TbV}(\eta')| \\
\leq &3\epsilon,
\end{aligned}
$$

where $\eta' \in \arg\min_{\eta \in N_\epsilon \cup N'_\epsilon} |\eta - \eta'|$. Changing $\epsilon$ to $\epsilon'/3$ finishes the proof. $\qquad \square$

### B.3.3 LOWER BOUNDING $\hat{F}_{TbV}$ FOR $\eta \in [1/L, \infty)$

In this section, we prove $\hat{F}_{TbV}$ is large for any step size $\eta \geq 1/L$. Therefore, the optimal step size $\eta^*_{\text{valid}}$ must be smaller than $\hat{F}_{TbV}$.

**Lemma 12.** *Suppose $\sigma$ is a large constant. Assume $t \geq c_2, d \geq c_4 \log(t)$ for some constants $c_2, c_4$. With probability at least $1 - \exp(-\Omega(m))$,*

$$\hat{F}_{TbV}(\eta) \geq C'\sigma^2 + \frac{1}{2}\sigma^2,$$

*for all $\eta \geq 1/L$, where $C'$ is a positive constant independent with $\sigma$.*

When the step size is very large (larger than $3L$), we know the GD sequence gets truncated with high probability, which immediately implies the loss is high. The proof of Lemma 19 is deferred into Section B.3.4.

**Lemma 19.** *Assume $t \geq c_2, d \geq c_4$ for some constants $c_2, c_4$. With probability at least $1 - \exp(-\Omega(m))$,*

$$\hat{F}_{TbV}(\eta) \geq \sigma^2,$$

*for all $\eta \in [3L, \infty)$*

The case for step size within $[1/L, 3L]$ requires more efforts. We give the proof of Lemma 20 in this section later.

**Lemma 20.** *Suppose $\sigma$ is a large constant. Assume $t \geq c_2, d \geq c_4 \log(t)$ for some constants $c_2, c_4$. With probability at least $1 - \exp(-\Omega(m))$,*

$$\hat{F}_{TbV}(\eta) \geq C_4 \sigma^2 + \frac{1}{2}\sigma^2,$$

*for all $\eta \in [1/L, 3L]$, where $C_4$ is a positive constant independent with $\sigma$.*

With the above two lemmas, Lemma 12 is just a combination of them.

**Proof of Lemma 12.** The result follows by taking a union bound and choosing $C' = \min(C_4, 1/2)$.
□

In the remaining of this section, we give the proof of Lemma 20. When the step size is between $1/L$ and $3L$, if the GD sequence has a reasonable probability of diverging, we can still show the loss is high similar as before. If not, we need to show the GD sequence overfits the noise in the training set, which incurs a high loss.

Recall that the noise term is roughly $\frac{1}{2} \left\| (I - (I - \eta H_{\text{train}})^t)(X_{\text{train}})^\dagger \xi_{\text{train}} \right\|^2$. When $\eta \in [1/L, 3L]$, the eigenvalues of $I - \eta H_{\text{train}}$ in $S_{\text{train}}$ subspace can be negative. If all the non-zero $n$ eigenvalues of $H_{\text{train}}$ have the same value, there exists a step size such that the eigenvalues of $I - \eta H_{\text{train}}$ in subspace $S_{\text{train}}$ is $-1$. If $t$ is even, the eigenvalues of $I - (I - \eta H_{\text{train}})^t$ in $S_{\text{train}}$ subspace are zero, which means GD sequence does not catch any noise in $S_{\text{train}}$.

Notice that the above problematic case cannot happen when the eigenvalues of $H_{\text{train}}$ are spread out. Basically, when there are two different eigenvalues, there won't exist any large $\eta$ that can cancel both directions at the same time. In Lemma 21, we show with constant probability, the eigenvalues of $H_{\text{train}}$ are indeed spread out. The proof is deferred into Section B.3.4.

**Lemma 21.** *Let the top $n$ eigenvalues of $H_{train}$ be $\lambda_1 \geq \cdots \geq \lambda_n$. Assume dimension $d \geq c_4$ for certain constant $c_4$. There exist positive constants $\mu, \mu', \mu''$ such that with probability at least $\mu$,*

$$\lambda_{\mu'n} - \lambda_{n-\mu'n+1} \geq \mu''.$$

Next, we utilize this variance in eigenvalues to prove that the GD sequence has to learn a constant fraction of the noise in training set.

**Lemma 22.** *Suppose noise level $\sigma$ is a large enough constant $c_1$. Assume unroll length $t \geq c_2$ and dimension $d \geq c_4$ for some constants $c_2, c_4$. Then, with probability at least $C_1$*

$$\|B_{t,\eta} w_{train} - w^*\|_{H_{train}}^2 \geq C_2 \sigma^2,$$

*for all $\eta \in [1/L, 3L]$, where $C_1, C_2$ are positive constants.*

**Proof of Lemma 22.** Let $\mathcal{E}_1$ be the event that $\sqrt{d}/\sqrt{L} \leq \sigma_i(X_{\text{train}}) \leq \sqrt{Ld}$ and $1/L \leq \lambda_i(H_{\text{train}}) \leq L$ for all $i \in [n]$ and $\sqrt{d}\sigma/4 \leq \|\xi_{\text{train}}\| \leq \sqrt{d}\sigma$. Let $\mathcal{E}_3$ be the event that $\sqrt{d}/\sqrt{L} \leq \sigma_i(X_{\text{valid}}) \leq \sqrt{Ld}$ and $1/L \leq \lambda_i(H_{\text{valid}}) \leq L$ for all $i \in [n]$ and $\sqrt{d}\sigma/4 \leq \|\xi_{\text{valid}}\| \leq \sqrt{d}\sigma$. According to Lemma 1 and Lemma 45, we know both $\mathcal{E}_1$ and $\mathcal{E}_3$ hold with probability at least $1 - \exp(-\Omega(d))$.

Let the top $n$ eigenvalues of $H_{\text{train}}$ be $\lambda_1 \geq \cdots \geq \lambda_n$. According to Lemma 21, assuming $d$ is larger than certain constant, we know there exist positive constants $\mu_1, \mu_2, \mu_3$ such that with probability at least $\mu_1, \lambda_{\mu_2 n} - \lambda_{n-\mu_2 n+1} \geq \mu_3$. Call this event $\mathcal{E}_2$.

Let $S_1$ and $S_2$ be the span of the bottom and top $\mu_2 n$ eigenvectors of $H_{\text{train}}$ respectively. According to Lemma 45, we know $\|\xi_{\text{train}}\| \geq \frac{\sqrt{d}}{4}\sigma$ with probability at least $1 - \exp(-\Omega(d))$. Let $P_1 \in \mathbb{R}^{n \times n}$ be a rank-$\mu_2 n$ projection matrix such that the column span of $(X_{\text{train}})^\dagger P_1$ is $S_1$. By Johnson-Lindenstrauss Lemma, we know with probability at least $1 - \exp(-\Omega(d))$, $\left\|\text{Proj}_{P_1}\xi_{\text{train}}\right\| \geq \frac{\sqrt{\mu_2}}{2}\|\xi_{\text{train}}\|$. Taking a union bound, with probability at least $1 - \exp(-\Omega(d))$, $\left\|\text{Proj}_{P_1}\xi_{\text{train}}\right\| \geq \frac{\sqrt{\mu_2 d}\sigma}{8}$. Similarly, we can define $P_2$ for the $S_2$ subspace and show with probability at least $1 - \exp(-\Omega(d))$, $\left\|\text{Proj}_{P_2}\xi_{\text{train}}\right\| \geq \frac{\sqrt{\mu_2 d}\sigma}{8}$. Call the intersection of both events as $\mathcal{E}_4$, which happens with with probability at least $1 - \exp(-\Omega(d))$.

Taking a union bound, we know $\mathcal{E}_1 \cap \mathcal{E}_2 \cap \mathcal{E}_3 \cap \mathcal{E}_4$ holds with probability at least $\mu_1/2$ as long as $d$ is larger than certain constant. Through the proof, we assume $\mathcal{E}_1 \cap \mathcal{E}_2 \cap \mathcal{E}_3 \cap \mathcal{E}_4$ holds.

Let's first lower bound $\|B_{t,\eta}w_{\text{train}} - w_{\text{train}}^*\|$ as follows,

$$\|B_{t,\eta}w_{\text{train}} - w_{\text{train}}^*\| = \left\|B_{t,\eta}\left(w_{\text{train}}^* + (X_{\text{train}})^\dagger\xi_{\text{train}}\right) - w_{\text{train}}^*\right\|$$
$$\geq \left(\left\|B_{t,\eta}\left(w_{\text{train}}^* + (X_{\text{train}})^\dagger\xi_{\text{train}}\right)\right\| - 1\right)$$

Recall that we define $S_1$ and $S_2$ as the span of the bottom and top $\mu_2 n$ eigenvectors of $H_{\text{train}}$ respectively. We rely on $S_1$ to lower bound $\|w_{t,\eta} - w^*\|$ when $\eta$ is small and rely on $S_2$ when $\eta$ is large.

**Case 1:** Let $\sigma_{\min}^{S_1}(B_{t,\eta})$ be the smallest singular value of $B_{t,\eta}$ within $S_1$ subspace. If $\eta\lambda_{n-\mu_2 n+1} \leq 2 - \mu_3/(2L)$, we have

$$\sigma_{\min}^{S_1}(B_{t,\eta}) \geq \min\left(1 - \left(1 - \frac{1}{L^2}\right)^t, 1 - \left(1 - \frac{\mu_3}{2L}\right)^t\right) \geq \frac{1}{2},$$

where the second inequality assumes $t \geq \max(L^2, 2L/\mu_3)\log 2$. Then, we have

$$\|w_{t,\eta} - w^*\| \geq \left(\sigma_{\min}^{S_1}(B_{t,\eta})\left(\left\|\text{Proj}_{S_1}(X_{\text{train}})^\dagger\xi_{\text{train}}\right\| - 1\right) - 1\right)$$
$$\geq \left(\frac{1}{2}\left(\frac{\sqrt{\mu_2}\sigma}{8\sqrt{L}} - 1\right) - 1\right) \geq \frac{\sqrt{\mu_2}\sigma}{32\sqrt{L}},$$

where the second inequality uses $\left\|\text{Proj}_{P_1}\xi_{\text{train}}\right\| \geq \frac{\sqrt{\mu_2 d}\sigma}{8}$ and the last inequality assumes $\sigma \geq \frac{48\sqrt{L}}{\sqrt{\mu_2}}$.

**Case 2:** If $\eta\lambda_{n-\mu_2 n+1} > 2 - \mu_3/(2L)$, we have $\eta\lambda_{\mu_2 n} \geq 2 + \mu_3/(2L)$ since $\lambda_{\mu_2 n} - \lambda_{n-\mu_2 n+1} \geq \mu_3$ and $\eta \geq 1/L$. Let $\sigma_{\min}^{S_2}(B_{t,\eta})$ be the smallest singular value of $B_{t,\eta}$ within $S_2$ subspace. We have

$$\sigma_{\min}^{S_2}(B_{t,\eta}) \geq \left(\left(1 + \frac{\mu_3}{2L}\right)^t - 1\right) \geq \frac{1}{2},$$

where the last inequality assumes $t \geq 4L/\mu_3$. Then, similar as in Case 1, we can also prove $\|w_{t,\eta} - w^*\| \geq \frac{\sqrt{\mu_2}\sigma}{32\sqrt{L}}$.

Therefore, we have

$$\|B_{t,\eta}w_{\text{train}} - w^*\|_{H_{\text{train}}}^2 = \|B_{t,\eta}w_{\text{train}} - w_{\text{train}}^*\|_{H_{\text{train}}}^2 \geq \frac{1}{L}\|B_{t,\eta}w_{\text{train}} - w_{\text{train}}^*\|^2 \geq \frac{\mu_2\sigma^2}{1024L^2},$$

for all $\eta \in [1/L, 3L]$. We denote $C_1 := \mu_1/2$ and $C_2 = \frac{\mu_2}{1024L^2}$. $\qquad\square$

Before we present the proof of Lemma 20, we still need a technical lemma that shows the noise in $S_{\text{valid}}$ concentrates at its mean. The proof of Lemma 23 is deferred into Section B.3.4.

**Lemma 23.** *Suppose $\sigma$ is constant. For any $1 > \epsilon > 0$, with probability at least $1 - O(t/\epsilon)\exp(-\Omega(\epsilon^2 d))$, $\lambda_n(H_{valid}) \geq 1/L$ and*

$$\|w_{t,\eta} - w_{valid}\|_{H_{valid}}^2 \geq \|w_{t,\eta} - w^*\|_{H_{valid}}^2 + (1 - \epsilon)\sigma^2,$$

*for all $\eta \in [1/L, 3L]$.*

Combing the above lemmas, we give the proof of Lemma 20.

**Proof of Lemma 20.** According to Lemma 23, we know given $1 > \epsilon > 0$, with probability at least $1 - O(t/\epsilon)\exp(-\Omega(\epsilon^2 d))$, $\lambda_n(H_{\text{valid}}) \geq 1/L$ and $\|w_{t,\eta} - w_{\text{valid}}\|^2_{H_{\text{valid}}} \geq \|w_{t,\eta} - w^*\|^2_{H_{\text{valid}}} + (1 - \epsilon)\sigma^2$ for all $\eta \in [1/L, 3L]$. Call this event $\mathcal{E}_1$. Suppose $\Pr[\mathcal{E}_1] \geq 1 - \delta/2$, where $\delta$ will be specifies later. For each training set $S_{\text{train}}^{(k)}$, we also define $\mathcal{E}_1^{(k)}$. By concentration, we know with probability at least $1 - \exp(-\Omega(\delta^2 m))$, $1/m \sum_{k=1}^{m} \mathbb{1}\left\{\mathcal{E}_1^{(k)}\right\} \geq 1 - \delta$.

According to Lemma 22, we know there exist constants $C_1, C_2$ such that with probability at least $C_1$,
$\|B_{t,\eta} w_{\text{train}} - w^*\|^2_{H_{\text{train}}} \geq C_2 \sigma^2$ for all $\eta \in [1/L, 3L]$. Call this event $\mathcal{E}_2$. For each training set $S_{\text{train}}^{(k)}$, we also define $\mathcal{E}_2^{(k)}$. By concentration, we know with probability at least $1 - \exp(-\Omega(m))$, $1/m \sum_{k=1}^{m} \mathbb{1}\left\{\mathcal{E}_2^{(k)}\right\} \geq C_1/2$.

For any step size $\eta \in [1/L, 3L]$, we can lower bound $\hat{F}_{TbV}(\eta)$ as follows,

$$
\begin{aligned}
\hat{F}_{TbV}(\eta) =& \frac{1}{m} \sum_{k=1}^{m} \frac{1}{2} \left\| w_{t,\eta}^{(k)} - w_{\text{valid}}^{(k)} \right\|^2_{H_{\text{valid}}^{(k)}} \\
\geq & \frac{1}{m} \sum_{k=1}^{m} \frac{1}{2} \left\| w_{t,\eta}^{(k)} - w_{\text{valid}}^{(k)} \right\|^2_{H_{\text{valid}}^{(k)}} \mathbb{1}\left\{\mathcal{E}_1^{(k)}\right\} \\
\geq & \frac{1}{m} \sum_{k=1}^{m} \frac{1}{2} \left\| w_{t,\eta}^{(k)} - w_k^* \right\|^2_{H_{\text{valid}}} \mathbb{1}\left\{\mathcal{E}_1^{(k)}\right\} + \frac{1}{2}(1 - \epsilon)(1 - \delta)\sigma^2 \\
\geq & \frac{1}{m} \sum_{k=1}^{m} \frac{1}{2} \left\| w_{t,\eta}^{(k)} - w_k^* \right\|^2_{H_{\text{valid}}} \mathbb{1}\left\{\mathcal{E}_1^{(k)} \cap \mathcal{E}_2^{(k)}\right\} + \frac{1}{2}(1 - \epsilon)(1 - \delta)\sigma^2.
\end{aligned}
$$

As long as $\delta \leq C_1/4$, we know $\frac{1}{m}\sum_{k=1}^{m} \mathbb{1}\left\{\mathcal{E}_1^{(k)} \cap \mathcal{E}_2^{(k)}\right\} \geq C_1/4$. Let $\bar{\mathcal{E}}_3(\eta)$ be the event that $w_{t,\eta}^{(k)}$ gets truncated with step size $\eta$. We have

$$
\begin{aligned}
& \frac{1}{m} \sum_{k=1}^{m} \frac{1}{2} \left\| w_{t,\eta}^{(k)} - w_k^* \right\|^2_{H_{\text{valid}}} \mathbb{1}\left\{\mathcal{E}_1^{(k)} \cap \mathcal{E}_2^{(k)}\right\} \\
=& \frac{1}{m} \sum_{k=1}^{m} \frac{1}{2} \left\| w_{t,\eta}^{(k)} - w_k^* \right\|^2_{H_{\text{valid}}} \mathbb{1}\left\{\mathcal{E}_1^{(k)} \cap \mathcal{E}_2^{(k)} \cap \mathcal{E}_3^{(k)}\right\} \\
& + \frac{1}{m} \sum_{k=1}^{m} \frac{1}{2} \left\| w_{t,\eta}^{(k)} - w_k^* \right\|^2_{H_{\text{valid}}} \mathbb{1}\left\{\mathcal{E}_1^{(k)} \cap \mathcal{E}_2^{(k)} \cap \bar{\mathcal{E}}_3^{(k)}\right\}.
\end{aligned}
$$

If $\frac{1}{m}\sum_{k=1}^{m} \mathbb{1}\left\{\mathcal{E}_1^{(k)} \cap \mathcal{E}_2^{(k)} \cap \bar{\mathcal{E}}_3^{(k)}\right\} \geq C_1/8$, we have

$$
\begin{aligned}
\frac{1}{m} \sum_{k=1}^{m} \frac{1}{2} \left\| w_{t,\eta}^{(k)} - w_k^* \right\|^2_{H_{\text{valid}}} \mathbb{1}\left\{\mathcal{E}_1^{(k)} \cap \mathcal{E}_2^{(k)}\right\} \geq & \frac{1}{m} \sum_{k=1}^{m} \frac{1}{2} \left\| w_{t,\eta}^{(k)} - w_k^* \right\|^2_{H_{\text{valid}}} \mathbb{1}\left\{\mathcal{E}_1^{(k)} \cap \mathcal{E}_2^{(k)} \cap \bar{\mathcal{E}}_3^{(k)}\right\} \\
\geq & \frac{C_1}{8} \times \frac{9\sigma^2}{2} = \frac{9 C_1 \sigma^2}{16}.
\end{aligned}
$$

Here, we lower bound $\left\| w_{t,\eta}^{(k)} - w_k^* \right\|^2_{H_{\text{valid}}}$ by $9\sigma^2$ when the sequence gets truncated.

If $\frac{1}{m}\sum_{k=1}^{m} \mathbb{1}\left\{\mathcal{E}_1^{(k)} \cap \mathcal{E}_2^{(k)} \cap \bar{\mathcal{E}}_3^{(k)}\right\} < C_1/8$, we know $\frac{1}{m}\sum_{k=1}^{m} \mathbb{1}\left\{\mathcal{E}_1^{(k)} \cap \mathcal{E}_2^{(k)} \cap \mathcal{E}_3^{(k)}\right\} \geq C_1/8$. Then, we have

$$
\begin{aligned}
\frac{1}{m} \sum_{k=1}^{m} \frac{1}{2} \left\| w_{t,\eta}^{(k)} - w_k^* \right\|^2_{H_{\text{valid}}} \mathbb{1}\left\{\mathcal{E}_1^{(k)} \cap \mathcal{E}_2^{(k)}\right\} \geq & \frac{1}{m} \sum_{k=1}^{m} \frac{1}{2} \left\| B_{t,\eta}^{(k)} w_{\text{train}} - w_k^* \right\|^2_{H_{\text{valid}}} \mathbb{1}\left\{\mathcal{E}_1^{(k)} \cap \mathcal{E}_2^{(k)} \cap \mathcal{E}_3^{(k)}\right\} \\
\geq & \frac{C_1}{8} \times \frac{C_2 \sigma^2}{2} = \frac{C_1 C_2 \sigma^2}{16}
\end{aligned}
$$

Letting $C_3 = \min(\frac{9C_1}{16}, \frac{C_1 C_2}{16})$, we then have

$$\hat{F}_{TbV}(\eta) \geq C_3 \sigma^2 + \frac{1}{2}(1 - \epsilon)(1 - \delta)\sigma^2 \geq \frac{C_3 \sigma^2}{2} + \frac{1}{2}\sigma^2,$$

where the last inequality chooses $\delta = \epsilon = C_3/2$. In order for $\Pr[\mathcal{E}_1] \geq 1 - \delta/2$, we only need $d \geq c_4 \log(t)$ for some constant $c_4$. Replacing $C_3/2$ by $C_4$ finishes the proof. $\qquad\square$

### B.3.4 Proofs of Technical Lemmas

**Proof of Lemma 14.** We first show that for a fixed $\eta \in [0, 1/L]$, the crossing term $\left|\left\langle B_{t,\eta} w^*_{\text{train}} - w^*, B_{t,\eta}(X_{\text{train}})^\dagger \xi_{\text{train}} \right\rangle\right|$ is small with high probability. We can write down the crossing term as follows:

$$\left\langle B_{t,\eta} w^*_{\text{train}} - w^*, B_{t,\eta}(X_{\text{train}})^\dagger \xi_{\text{train}} \right\rangle = \left\langle [(X_{\text{train}})^\dagger]^\top B_{t,\eta}(B_{t,\eta} w^*_{\text{train}} - w^*), \xi_{\text{train}} \right\rangle.$$

Noticing that $\xi_{\text{train}}$ is independent with $[(X_{\text{train}})^\dagger]^\top B_{t,\eta}(B_{t,\eta} w^*_{\text{train}} - w^*)$, we will use Hoeffding's inequality to bound $\left|\left\langle B_{t,\eta} w^*_{\text{train}} - w^*, B_{t,\eta}(X_{\text{train}})^\dagger \xi_{\text{train}} \right\rangle\right|$. According to Lemma 1, we know with probability at least $1 - \exp(-\Omega(d))$, $\sqrt{d}/\sqrt{L} \leq \sigma_i(X_{\text{train}}) \leq \sqrt{Ld}$ and $1/L \leq \lambda_i(H_{\text{train}}) \leq L$ for all $i \in [n]$ with $L = 100$. Since $\eta \leq 1/L$, we know $\|B_{t,\eta}\| = \|I - (I - \eta H_{\text{train}})^t\| \leq 1$. Therefore, we have

$$\left\|[(X_{\text{train}})^\dagger]^\top B_{t,\eta}(B_{t,\eta} w^*_{\text{train}} - w^*)\right\| \leq \frac{2\sqrt{L}}{\sqrt{d}},$$

for any $\eta \in [0, 1/L]$. Then, for any $\epsilon > 0$, by Hoeffding's inequality, with probability at least $1 - \exp(-\Omega(\epsilon^2 d))$,

$$\left|\left\langle B_{t,\eta} w^*_{\text{train}} - w^*, B_{t,\eta}(X_{\text{train}})^\dagger \xi_{\text{train}} \right\rangle\right| \leq \epsilon.$$

Next, we construct an $\epsilon$-net on $\eta$ and show the crossing term is small for all $\eta \in [0, 1/L]$. Let

$$g(\eta) := \left\langle B_{t,\eta} w^*_{\text{train}} - w^*, B_{t,\eta}(X_{\text{train}})^\dagger \xi_{\text{train}} \right\rangle.$$

We compute the derivative of $g(\eta)$ as follows:

$$g'(\eta) = \left\langle t H_{\text{train}}(I - \eta H_{\text{train}})^{t-1} w^*_{\text{train}}, B_{t,\eta}(X_{\text{train}})^\dagger \xi_{\text{train}} \right\rangle$$
$$+ \left\langle B_{t,\eta} w^*_{\text{train}} - w^*, t H_{\text{train}}(I - \eta H_{\text{train}})^{t-1}(X_{\text{train}})^\dagger \xi_{\text{train}} \right\rangle$$

By Lemma 45, we know with probability at least $1 - \exp(-\Omega(d))$, $\|\xi_{\text{train}}\| \leq \sqrt{d}\sigma$. Therefore,

$$|g'(\eta)| \leq L^{1.5} t \left(1 - \frac{\eta}{L}\right)^{t-1} \sigma + 2L^{1.5} t \left(1 - \frac{\eta}{L}\right)^{t-1} \sigma = 3L^{1.5} t \left(1 - \frac{\eta}{L}\right)^{t-1} \sigma.$$

We can control $|g'(\eta)|$ in different regimes:

- For $\eta \in [0, \frac{L}{t-1}]$, we have $|g'(\eta)| \leq 3L^{1.5} t \sigma$.
- Given any $1 \leq i \leq \log t - 1$, for any $\eta \in (\frac{iL}{t-1}, \frac{(i+1)L}{t-1}]$, we have $|g'(\eta)| \leq \frac{3L^{1.5} t \sigma}{e^i}$.
- For any $\eta \in (\frac{L \log t}{t-1}, 1/L]$, we have $|g'(\eta)| \leq 3L^{1.5} \sigma$.

Fix any $\epsilon > 0$, we know there exists an $\epsilon$-net $N_\epsilon$ with size

$$|N_\epsilon| = \frac{1}{\epsilon}\left(\frac{L}{t-1}\sum_{i=0}^{\log t - 1}\frac{3L^{1.5} t \sigma}{e^i} + \left(\frac{1}{L} - \frac{L \log t}{t-1}\right)3L^{1.5}\sigma\right)$$
$$\leq \frac{1}{\epsilon}\left(\frac{3eL^{2.5} t \sigma}{t-1} + 3\sqrt{L}\sigma\right) = O(\frac{1}{\epsilon})$$

such that for any $\eta \in [0, 1/L]$, there exists $\eta' \in N_\epsilon$ with $|g(\eta) - g(\eta')| \leq \epsilon$. Note that $L = 100$ and $\sigma$ is a constant. Taking a union bound over $N_\epsilon$ and all the other bad events, we have with probability at least $1 - \exp(-\Omega(d)) - O(1/\epsilon)\exp(-\Omega(\epsilon^2 d))$, for all $\eta \in [0, 1/L]$,

$$\left|\left\langle B_{t,\eta} w^*_{\text{train}} - w^*, B_{t,\eta}(X_{\text{train}})^\dagger \xi_{\text{train}} \right\rangle\right| \leq \epsilon + \epsilon = 2\epsilon.$$

As long as $1 > \epsilon > 0$, this happens with probability at least $1 - O(1/\epsilon)\exp(-\Omega(\epsilon^2 d))$. Replacing $\epsilon$ by $\epsilon'/2$ finishes the proof. $\qquad\square$

**Proof of Lemma 15.** According to Lemma 1, we know with probability at least $1 - \exp(-\Omega(d))$, $1/L \leq \lambda_i(H_{\text{train}}) \leq L$ for all $i \in [n]$ with $L = 100$. We can lower bound $\langle B_{t,\eta} w^*_{\text{train}}, w^* \rangle$ as follows,

$$
\begin{aligned}
\langle B_{t,\eta} w^*_{\text{train}}, w^* \rangle &= \left\langle \left( I - (I - \eta H_{\text{train}})^t \right) w^*_{\text{train}}, w^*_{\text{train}} \right\rangle \\
&\geq \lambda_{\min}\left( I - (I - \eta H_{\text{train}})^t \right) \| w^*_{\text{train}} \|^2 \\
&\geq \left( 1 - \exp\left( -\frac{\eta t}{L} \right) \right) \| w^*_{\text{train}} \|^2 .
\end{aligned}
$$

By Johnson-Lindenstrauss lemma (Lemma 49), we know with probability at least $1 - 2\exp(-c\epsilon^2 d/4)$,

$$
\| w^*_{\text{train}} \| \geq \frac{1}{2}(1 - \epsilon) \| w^* \| = \frac{1}{2}(1 - \epsilon).
$$

Then, we know with probability at least $1 - 2\exp(-c\epsilon^2 d/4) - \exp(-\Omega(d))$,

$$
\begin{aligned}
\langle B_{t,\eta} w^*_{\text{train}}, w^* \rangle &\geq \left( 1 - \exp\left( -\frac{\eta t}{L} \right) \right) \| w^*_{\text{train}} \|^2 \\
&\geq \left( 1 - \exp\left( -\frac{\eta t}{L} \right) \right) \frac{1}{4}(1 - \epsilon)^2 \\
&\geq \frac{1 - 2\epsilon}{4} \left( 1 - \exp\left( -\frac{\eta t}{L} \right) \right)
\end{aligned}
$$

Since $e^x \leq 1 - x + x^2/2$ for any $x \leq 0$, we know $\exp(-\eta t/L) \leq 1 - \eta t/L + \eta^2 t^2/(2L^2)$. For any $\eta \leq L/t$, we have $\exp(-\eta t/L) \leq 1 - \eta t/(2L)$. Then with probability at least $1 - 2\exp(-c\epsilon^2 d/4) - \exp(-\Omega(d))$,

$$
\begin{aligned}
\langle B_{t,\eta} w^*_{\text{train}}, w^* \rangle &\geq \frac{1 - 2\epsilon}{4} \frac{\eta t}{2L} \\
&\geq \frac{\eta t}{16L},
\end{aligned}
$$

where the second inequality holds by choosing $\epsilon = 1/4$. $\qquad\square$

**Proof of Lemma 16.** Recall that

$$
\hat{F}_{TbV}(\eta) := \frac{1}{m} \sum_{k=1}^{m} \Delta_{TbV}(\eta, P_k)
$$

For each individual loss function $\Delta_{TbV}(\eta, P_k)$, we have

$$
\begin{aligned}
\Delta_{TbV}(\eta, P_k) &= \frac{1}{2} \left\| w^{(k)}_{t,\eta} - w^* - (X^{(k)}_{\text{valid}})^\dagger \xi^{(k)}_{\text{valid}} \right\|^2_{H^{(k)}_{\text{valid}}} \\
&= \frac{1}{2} \left\| w^{(k)}_{t,\eta} - w^* \right\|^2_{H^{(k)}_{\text{valid}}} + \frac{1}{2n} \left\| \xi^{(k)}_{\text{valid}} \right\|^2 + \left\langle w^{(k)}_{t,\eta} - w^*, \frac{1}{n}(X^{(k)}_{\text{valid}})^\top \xi^{(k)}_{\text{valid}} \right\rangle \\
&\leq \frac{25 L \sigma^2}{2} \left\| H^{(k)}_{\text{valid}} \right\| + \frac{1}{2n} \left\| \xi^{(k)}_{\text{valid}} \right\|^2 + 5\sqrt{L}\sigma \left( \frac{1}{\sqrt{n}} \left\| X^{(k)}_{\text{valid}} \right\| \right) \left( \frac{1}{\sqrt{n}} \left\| \xi^{(k)}_{\text{valid}} \right\| \right)
\end{aligned}
$$

We can write $\left\| H^{(k)}_{\text{valid}} \right\|$ as $\sigma^2_{\max}(\frac{1}{\sqrt{n}} X^{(k)}_{\text{valid}})$. According to Lemma 47, we know $\sigma_{\max}(X^{(k)}_{\text{valid}}) - \mathbb{E}\sigma_{\max}(X^{(k)}_{\text{valid}})$ is $O(1)$-subgaussian, which implies that $\sigma_{\max}(\frac{1}{\sqrt{n}} X^{(k)}_{\text{valid}}) - \mathbb{E}\sigma_{\max}(\frac{1}{\sqrt{n}} X^{(k)}_{\text{valid}})$ is $O(1/\sqrt{d})$-subgaussian. Since $\mathbb{E}\sigma_{\max}(\frac{1}{\sqrt{n}} X^{(k)}_{\text{valid}})$ is a constant, we know $\sigma_{\max}(\frac{1}{\sqrt{n}} X^{(k)}_{\text{valid}})$ is $O(1)$-subgaussian and $\sigma^2_{\max}(\frac{1}{\sqrt{n}} X^{(k)}_{\text{valid}})$ is $O(1)$-subexponential. Similarly, we know both $\frac{1}{2n} \left\| \xi^{(k)}_{\text{valid}} \right\|^2$ and $\left( \frac{1}{\sqrt{n}} \left\| X^{(k)}_{\text{valid}} \right\| \right) \left( \frac{1}{\sqrt{n}} \left\| \xi^{(k)}_{\text{valid}} \right\| \right)$ are $O(1)$-subexponential. This further implies that $\Delta_{TbV}(\eta, P_k)$ is

$O(1)$-subexponential. Therefore, $\hat{F}_{TbV}$ is the average of $m$ i.i.d. $O(1)$-subexponential random variables. By standard concentration inequality, we know for any $1 > \epsilon > 0$, with probability at least $1 - \exp(-\Omega(\epsilon^2 m))$,

$$\left| \hat{F}_{TbV}(\eta) - F_{TbV}(\eta) \right| \le \epsilon.$$

$\square$

**Proof of Lemma 17.** Recall that

$$F_{TbV}(\eta) = \mathbb{E} \frac{1}{2} \|w_{t,\eta} - w^*\|^2 + \sigma^2/2.$$

We only need to construct an $\epsilon$-net for $\mathbb{E}\frac{1}{2} \|w_{t,\eta} - w^*\|^2$. Let $\mathcal{E}$ be the event that $\sqrt{d}/\sqrt{L} \le \sigma_i(X_{\text{train}}) \le \sqrt{Ld}$ and $1/L \le \lambda_i(H_{\text{train}}) \le L$ for all $i \in [n]$ and $\|\xi_{\text{train}}\| \le \sqrt{d}\sigma$. We have

$$\mathbb{E}\frac{1}{2} \|w_{t,\eta} - w^*\|^2 = \mathbb{E}\left[ \frac{1}{2} \|w_{t,\eta} - w^*\|^2 \,|\mathcal{E}\right] \Pr[\mathcal{E}] + \mathbb{E}\left[ \frac{1}{2} \|w_{t,\eta} - w^*\|^2 \,|\bar{\mathcal{E}}\right] \Pr[\bar{\mathcal{E}}]$$

We first construct an $\epsilon$-net for $\mathbb{E}\left[ \frac{1}{2} \|w_{t,\eta} - w^*\|^2 \,|\mathcal{E}\right] \Pr[\mathcal{E}]$. Let $Q(\eta) := \frac{1}{2} \|w_{t,\eta} - w^*\|^2$. Fix a training set $S_{\text{train}}$ under which event $\mathcal{E}$ holds. We show that $Q(\eta)$ has desirable lipschitz property. The derivative of $Q(\eta)$ can be computed as follows,

$$Q'(\eta) = \left\langle tH_{\text{train}}(I - \eta H_{\text{train}})^{t-1} w_{\text{train}}, w_{t,\eta} - w^* \right\rangle.$$

Conditioning on $\mathcal{E}$, we have

$$|Q'(\eta)| = O(1)t(1 - \frac{\eta}{L})^{t-1}.$$

Therefore, we have

$$\left| \frac{\partial}{\partial \eta} \mathbb{E}\left[ \frac{1}{2} \|w_{t,\eta} - w^*\|^2 \,|\mathcal{E}\right] \Pr[\mathcal{E}] \right| = O(1)t(1 - \frac{\eta}{L})^{t-1}.$$

Similar as in Lemma 14, for any $\epsilon > 0$, we know there exists an $\epsilon$-net $N_\epsilon$ with size $O(1/\epsilon)$ such that for any $\eta \in [0, 1/L]$,

$$\left| \mathbb{E}\left[ \frac{1}{2} \|w_{t,\eta} - w^*\|^2 \,|\mathcal{E}\right] \Pr[\mathcal{E}] - \mathbb{E}\left[ \frac{1}{2} \|w_{t,\eta'} - w^*\|^2 \,|\mathcal{E}\right] \Pr[\mathcal{E}] \right| \le \epsilon$$

for $\eta' \in \arg\min_{\eta \in N_\epsilon} |\eta - \eta'|$.

Suppose the probability of $\bar{\mathcal{E}}$ is $\delta$. We have

$$\mathbb{E}\left[ \frac{1}{2} \|w_{t,\eta} - w^*\|^2 \,|\bar{\mathcal{E}}\right] \Pr[\bar{\mathcal{E}}] \le \frac{25L\sigma^2}{2}\delta \le \epsilon,$$

where the last inequality assumes $\delta \le \frac{2\epsilon}{25L\sigma^2}$. According to Lemma 1 and Lemma 45, we know $\delta := \Pr[\bar{\mathcal{E}}] \le \exp(-\Omega(d))$. Therefore, given any $\epsilon > 0$, there exists constant $c_4$ such that $\delta \le \frac{2\epsilon}{25L\sigma^2}$ as long as $d \ge c_4 \log(1/\epsilon)$.

Overall, for any $\epsilon > 0$, as long as $d = \Omega(\log(1/\epsilon))$, there exists $N_\epsilon$ with size $O(1/\epsilon)$ such that for any $\eta \in [0, 1/L]$, $|F_{TbV}(\eta) - F_{TbV}(\eta')| \le 3\epsilon$ for $\eta' \in \arg\min_{\eta \in N_\epsilon} |\eta - \eta'|$. Changing $\epsilon$ to $\epsilon'/3$ finishes the proof. $\square$

**Proof of Lemma 18.** For each $k \in [m]$, let $\mathcal{E}_k$ be the event that $\sqrt{d}/\sqrt{L} \le \sigma_i(X_{\text{train}}^{(k)}) \le \sqrt{Ld}$ for any $i \in [n]$ and $\left\| \xi_{\text{train}}^{(k)} \right\| \le \sqrt{d}\sigma$. Then, we can write the empirical meta objective as follows,

$$\hat{F}_{TbV}(\eta) := \frac{1}{m} \sum_{k=1}^{m} \Delta_{TbT}(\eta, P_k) \mathbb{1}_{\mathcal{E}_k} + \frac{1}{m} \sum_{k=1}^{m} \Delta_{TbT}(\eta, P_k) \mathbb{1}_{\bar{\mathcal{E}}_k}.$$

Similar as Lemma 17, we will show that the first term has desirable Lipschitz property and the second term is small. Now, let's focus on the first term $\frac{1}{m}\sum_{k=1}^{m}\Delta_{TbT}(\eta,P_k)\mathbb{1}_{\mathcal{E}_k}$. Recall that

$$\Delta_{TbT}(\eta,P_k) = \frac{1}{2}\left\|w_{t,\eta}^{(k)} - w_{\text{valid}}^{(k)}\right\|_{H_{\text{valid}}^{(k)}}^{2}$$
$$= \frac{1}{2}\left\|B_{t,\eta}^{(k)}w_{\text{train}}^{(k)} - w^* - (X_{\text{valid}}^{(k)})^{\dagger}\xi_{\text{valid}}^{(k)}\right\|_{H_{\text{valid}}^{(k)}}^{2}.$$

Computing the derivative of $\Delta_{TbT}(\eta,P_k)$ in terms of $\eta$, we have

$$\frac{\partial}{\partial\eta}\Delta_{TbT}(\eta,P_k) = \left\langle tH_{\text{train}}^{(k)}(I - \eta H_{\text{train}}^{(k)})^{t-1}w_{\text{train}}^{(k)}, H_{\text{valid}}^{(k)}\left(w_{t,\eta}^{(k)} - w^* - (X_{\text{valid}}^{(k)})^{\dagger}\xi_{\text{valid}}^{(k)}\right)\right\rangle$$

Conditioning on $\mathcal{E}_k$, we can bound the derivative,

$$\left|\frac{\partial}{\partial\eta}\Delta_{TbT}(\eta,P_k)\right| = O(1)t\left(1 - \frac{\eta}{L}\right)^{t-1}\left(\left\|H_{\text{valid}}^{(k)}\right\| + \left(\frac{1}{\sqrt{d}}\left\|X_{\text{valid}}^{(k)}\right\|\right)\left(\frac{1}{\sqrt{d}}\left\|\xi_{\text{valid}}^{(k)}\right\|\right)\right).$$

Therefore, we have

$$\left|\frac{1}{m}\sum_{k=1}^{m}\frac{\partial}{\partial\eta}\Delta_{TbT}(\eta,P_k)\mathbb{1}_{\mathcal{E}_k}\right| = O(1)t\left(1 - \frac{\eta}{L}\right)^{t-1}\frac{1}{m}\sum_{k=1}^{m}\left(\left\|H_{\text{valid}}^{(k)}\right\| + \left(\frac{1}{\sqrt{d}}\left\|X_{\text{valid}}^{(k)}\right\|\right)\left(\frac{1}{\sqrt{d}}\left\|\xi_{\text{valid}}^{(k)}\right\|\right)\right).$$

Similar as in Lemma 16, we know both $\left\|H_{\text{valid}}^{(k)}\right\|$ and $\left(\frac{1}{\sqrt{d}}\left\|X_{\text{valid}}^{(k)}\right\|\right)\left(\frac{1}{\sqrt{d}}\left\|\xi_{\text{valid}}^{(k)}\right\|\right)$ are $O(1)$-subexponential. Therefore, we know with probability at least $1 - \exp(-\Omega(m))$, $\frac{1}{m}\sum_{k=1}^{m}\left(\left\|H_{\text{valid}}^{(k)}\right\| + \left(\frac{1}{\sqrt{d}}\left\|X_{\text{valid}}^{(k)}\right\|\right)\left(\frac{1}{\sqrt{d}}\left\|\xi_{\text{valid}}^{(k)}\right\|\right)\right) = O(1)$. This further shows that with probability at least $1 - \exp(-\Omega(m))$,

$$\left|\frac{1}{m}\sum_{k=1}^{m}\frac{\partial}{\partial\eta}\Delta_{TbT}(\eta,P_k)\mathbb{1}_{\mathcal{E}_k}\right| = O(1)t\left(1 - \frac{\eta}{L}\right)^{t-1}.$$

Similar as in Lemma 14, we can show that for any $\epsilon > 0$, there exists an $\epsilon$-net with size $O(1/\epsilon)$ for $\frac{1}{m}\sum_{k=1}^{m}\Delta_{TbT}(\eta,P_k)\mathbb{1}_{\mathcal{E}_k}$.

Next, we show that the second term $\frac{1}{m}\sum_{k=1}^{m}\Delta_{TbT}(\eta,P_k)\mathbb{1}_{\bar{\mathcal{E}}_k}$ is small with high probability. According to the proof in Lemma 16, we know

$$\Delta_{TbT}(\eta,P_k) = O(1)\left(\left\|H_{\text{valid}}^{(k)}\right\| + \frac{1}{d}\left\|\xi_{\text{valid}}^{(k)}\right\|^2 + \left(\frac{1}{\sqrt{d}}\left\|X_{\text{valid}}^{(k)}\right\|\right)\left(\frac{1}{\sqrt{d}}\left\|\xi_{\text{valid}}^{(k)}\right\|\right)\right)$$

Therefore, there exists constant $C$ such that

$$\frac{1}{m}\sum_{k=1}^{m}\Delta_{TbT}(\eta,P_k)\mathbb{1}_{\bar{\mathcal{E}}_k} \leq C\frac{1}{m}\sum_{k=1}^{m}\left(\left\|H_{\text{valid}}^{(k)}\right\| + \frac{1}{d}\left\|\xi_{\text{valid}}^{(k)}\right\|^2 + \left(\frac{1}{\sqrt{d}}\left\|X_{\text{valid}}^{(k)}\right\|\right)\left(\frac{1}{\sqrt{d}}\left\|\xi_{\text{valid}}^{(k)}\right\|\right)\right)\mathbb{1}_{\bar{\mathcal{E}}_k}.$$

It's not hard to verify that $\left(\left\|H_{\text{valid}}^{(k)}\right\| + \frac{1}{d}\left\|\xi_{\text{valid}}^{(k)}\right\|^2 + \left(\frac{1}{\sqrt{d}}\left\|X_{\text{valid}}^{(k)}\right\|\right)\left(\frac{1}{\sqrt{d}}\left\|\xi_{\text{valid}}^{(k)}\right\|\right)\right)\mathbb{1}_{\bar{\mathcal{E}}_k}$ is $O(1)$-subexponential. Suppose the expectation of $\left(\left\|H_{\text{valid}}^{(k)}\right\| + \frac{1}{d}\left\|\xi_{\text{valid}}^{(k)}\right\|^2 + \left(\frac{1}{\sqrt{d}}\left\|X_{\text{valid}}^{(k)}\right\|\right)\left(\frac{1}{\sqrt{d}}\left\|\xi_{\text{valid}}^{(k)}\right\|\right)\right)$ is $\mu$, which is a constant. Suppose the probability of $\bar{\mathcal{E}}_k$ be $\delta$. We know the expectation of $\left(\left\|H_{\text{valid}}^{(k)}\right\| + \frac{1}{d}\left\|\xi_{\text{valid}}^{(k)}\right\|^2 + \left(\frac{1}{\sqrt{d}}\left\|X_{\text{valid}}^{(k)}\right\|\right)\left(\frac{1}{\sqrt{d}}\left\|\xi_{\text{valid}}^{(k)}\right\|\right)\right)\mathbb{1}_{\bar{\mathcal{E}}_k}$ is $\mu\delta$ due to independence. By standard concentration inequality, for any $1 > \epsilon > 0$, with probability at least $1 - \exp(-\Omega(\epsilon^2 m))$,

$$C\frac{1}{m}\sum_{k=1}^{m}\left(\left\|H_{\text{valid}}^{(k)}\right\| + \frac{1}{d}\left\|\xi_{\text{valid}}^{(k)}\right\|^2 + \left(\frac{1}{\sqrt{d}}\left\|X_{\text{valid}}^{(k)}\right\|\right)\left(\frac{1}{\sqrt{d}}\left\|\xi_{\text{valid}}^{(k)}\right\|\right)\right)\mathbb{1}_{\bar{\mathcal{E}}_k} \leq C\mu\delta + C\epsilon \leq (C+1)\epsilon,$$

where the second inequality assumes $\delta \leq \epsilon/(C\mu)$. By Lemma 1 and Lemma 45, we know $\delta \leq \exp(-\Omega(d))$. Therefore, as long as $d \geq c_4\log(1/\epsilon)$ for some constant $c_4$, we have $\delta \leq \epsilon/(C\mu)$.

Overall, we know that as long as $d \geq c_4 \log(1/\epsilon)$, with probability at least $1 - \exp(-\Omega(\epsilon^2 m))$, there exists $N'_\epsilon$ with $|N'_\epsilon| = O(1/\epsilon)$ such that for any $\eta \in [0, 1/L]$,

$$|\hat{F}_{TbV}(\eta) - \hat{F}_{TbV}(\eta')| \leq (2C + 3)\epsilon,$$

for $\eta' \in \arg\min_{\eta \in N_\epsilon} |\eta - \eta'|$. Changing $\epsilon$ to $\epsilon'/(2C + 3)$ finishes the proof. $\qquad\square$

**Proof of Lemma 19.** Let $\mathcal{E}_1$ be the event that $\sqrt{d}/\sqrt{L} \leq \sigma_i(X_{\text{train}}) \leq \sqrt{Ld}$ and $1/L \leq \lambda_i(H_{\text{train}}) \leq L$ for all $i \in [n]$ and $\sqrt{d}\sigma/4 \leq \|\xi_{\text{train}}\| \leq \sqrt{d}\sigma$. Let $\mathcal{E}_2$ be the event that $\sqrt{d}/\sqrt{L} \leq \sigma_i(X_{\text{valid}}) \leq \sqrt{Ld}$ and $1/L \leq \lambda_i(H_{\text{valid}}) \leq L$ for all $i \in [n]$ and $\sqrt{d}\sigma/4 \leq \|\xi_{\text{valid}}\| \leq \sqrt{d}\sigma$. According to Lemma 1 and Lemma 45, we know both $\mathcal{E}_1$ and $\mathcal{E}_2$ hold with probability at least $1 - \exp(-\Omega(d))$. Assuming $d \geq c_4$ for certain constant $c_4$, we know $\Pr[\mathcal{E}_1 \cap \mathcal{E}_2] \geq 2/3$. Also define $\mathcal{E}_1^{(k)}$ and $\mathcal{E}_2^{(k)}$ on each training set $S_{\text{train}}^{(k)}$. By concentration, we know with probability at least $1 - \exp(-\Omega(m))$,

$$\frac{1}{m} \sum_{k=1}^{m} \mathbb{1}\left\{\mathcal{E}_1^{(k)} \cap \mathcal{E}_2^{(k)}\right\} \geq \frac{1}{2}.$$

It's easy to verify that conditioning on $\mathcal{E}_1$, the GD sequence always exceeds the norm threshold and gets truncated for $\eta \geq 3L$ as long as $t$ is larger than certain constant. We can lower bound $\hat{F}_{TbV}$ for any $\eta \geq 3L$ as follows,

$$\hat{F}_{TbV}(\eta) = \frac{1}{m} \sum_{k=1}^{m} \frac{1}{2} \left\|w_{t,\eta}^{(k)} - w_{\text{valid}}^{(k)}\right\|_{H_{\text{valid}}^{(k)}}^2$$

$$\geq \frac{1}{m} \sum_{k=1}^{m} \frac{1}{2} \left\|w_{t,\eta}^{(k)} - w_{\text{valid}}^{(k)}\right\|_{H_{\text{valid}}^{(k)}}^2 \mathbb{1}\left\{\mathcal{E}_1 \cap \mathcal{E}_2\right\} \geq 2\sigma^2 \frac{1}{2} = \sigma^2,$$

where the last inequality lower bounds $\left\|w_{t,\eta}^{(k)} - w_{\text{valid}}^{(k)}\right\|_{H_{\text{valid}}^{(k)}}^2$ by $2\sigma^2$ when $w_{t,\eta}^{(k)}$ gets truncated. $\qquad\square$

**Proof of Lemma 21.** We first show that with constant probability in $X_{\text{train}}$, the variance of the eigenvalues of $H_{\text{train}}$ is lower bounded by a constant. Let $\bar{\lambda}$ be $1/n \sum_{i=1}^{n} \lambda_i$. Specifically, we show $1/n \sum_{i=1}^{n} \lambda_i^2 - \bar{\lambda}^2$ is lower bounded by a constant.

Let's first compute the variance of the eigenvalues in expectation. Let the $i$-th row of $X_{\text{train}}$ be $x_i^\top$. We have,

$$\mathbb{E}\left[\bar{\lambda}^2\right] = \frac{1}{n^2} \mathbb{E}\left[\left(\text{tr}\left(\frac{1}{n} X_{\text{train}}^\top X_{\text{train}}\right)\right)^2\right] = \frac{1}{n^4} \mathbb{E}\left[\left(\sum_{i=1}^{n} \|x_i\|^2\right)^2\right]$$

$$= \frac{1}{n^4} \sum_{i=1}^{n} \mathbb{E}\|x_i\|^4 + \frac{1}{n^4} \sum_{1 \leq i \neq j \leq n} \mathbb{E}\|x_i\|^2 \|x_j\|^2$$

$$= \frac{1}{n^4}\left(nd(d+2) + n(n-1)d^2\right) = \frac{d^2}{n^2} + \frac{2d}{n^3}.$$

Similarly, we compute $\mathbb{E}\left[1/n \sum_{i=1}^{n} \lambda_i^2\right]$ as follows,

$$\mathbb{E}\left[\frac{1}{n} \sum_{i=1}^{n} \lambda_i^2\right] = \frac{1}{n^3} \mathbb{E}\left[\text{tr}\left(X_{\text{train}}^\top X_{\text{train}} X_{\text{train}}^\top X_{\text{train}}\right)\right]$$

$$= \frac{1}{n^3} \sum_{i=1}^{n} \mathbb{E}\|x_i\|^4 + \frac{1}{n^3} \sum_{1 \leq i \neq j \leq n} \mathbb{E}\langle x_i, x_j\rangle^2$$

$$= \frac{1}{n^3}\left(nd(d+2) + n(n-1)d\right) = \frac{d^2}{n^2} + \frac{d}{n} + \frac{d}{n^2}$$

Therefore, we have

$$\mathbb{E}\left[\frac{1}{n} \sum_{i=1}^{n} \lambda_i^2 - \bar{\lambda}^2\right] = \frac{d}{n} + \frac{d}{n^2} - \frac{2d}{n^3} \geq \frac{d}{n} \geq \frac{4}{3},$$

where the first inequality assumes $n \geq 2$ and the last inequality uses $n \leq \frac{3d}{4}$. Since $n \geq \frac{1}{4}d$, we know $n \geq 2$ as long as $d \geq 8$.

Let $\mathcal{E}$ be the event that $\sqrt{d}/\sqrt{L} \leq \sigma_i(X_{\text{train}}) \leq \sqrt{Ld}$ and $1/L \leq \lambda_i(H_{\text{train}}) \leq L$ for $i \in [n]$ with $L = 100$. According to Lemma 1, we know $\mathcal{E}$ happens with probability at least $1 - \exp(-\Omega(d))$. Let $\mathbb{1}\{\mathcal{E}\}$ be the indicator function for event $\mathcal{E}$. Next we show that $\mathbb{E}[1/n \sum_{i=1}^{n}(\lambda_i - \bar{\lambda})^2 \mathbb{1}\{\mathcal{E}\}]$ is also lower bounded.

It's clear that $\mathbb{E}\left[\bar{\lambda}^2 \mathbb{1}\{\mathcal{E}\}\right]$ is upper bounded by $\mathbb{E}\left[\bar{\lambda}^2\right]$. In order to lower bound $\mathbb{E}\left[\frac{1}{n} \sum_{i=1}^{n} \lambda_i^2 \mathbb{1}\{\mathcal{E}\}\right]$, we first show that $\mathbb{E}\left[\frac{1}{n} \sum_{i=1}^{n} \lambda_i^2 \mathbb{1}\{\bar{\mathcal{E}}\}\right]$ is small. We can decompose $\mathbb{E}\left[\frac{1}{n} \sum_{i=1}^{n} \lambda_i^2 \mathbb{1}\{\bar{\mathcal{E}}\}\right]$ into two parts,

$$\mathbb{E}\left[\frac{1}{n} \sum_{i=1}^{n} \lambda_i^2 \mathbb{1}\{\bar{\mathcal{E}}\}\right] = \mathbb{E}\left[\frac{1}{n} \sum_{i=1}^{n} \lambda_i^2 \mathbb{1}\{\bar{\mathcal{E}} \text{ and } \lambda_1 \leq L\}\right] + \mathbb{E}\left[\frac{1}{n} \sum_{i=1}^{n} \lambda_i^2 \mathbb{1}\{\lambda_1 > L\}\right].$$

The first term can be bounded by $L^2 \Pr[\bar{\mathcal{E}}]$. Since $\Pr[\bar{\mathcal{E}}] \leq \exp(-\Omega(d))$, we know the first term is at most $1/6$ as long as $d$ is larger than certain constant. The second term can be bounded by $\mathbb{E}\left[\lambda_1^2 \mathbb{1}\{\lambda_1 > L\}\right]$. According to Lemma 48, we know $\Pr[\lambda_1 \geq L + t] \leq \exp(-\Omega(dt))$. Then, it's not hard to verify that $\mathbb{E}\left[\lambda_1^2 \mathbb{1}\{\lambda_1 > L\}\right] = O(1/d)$ that is bounded by $1/6$ as long as $d$ is larger than certain constant. Overall, we know $\mathbb{E}\left[\frac{1}{n} \sum_{i=1}^{n} \lambda_i^2 \mathbb{1}\{\mathcal{E}\}\right] \geq \mathbb{E}\left[\frac{1}{n} \sum_{i=1}^{n} \lambda_i^2\right] - 1/3$. Combing with the upper bounds on $\mathbb{E}\left[\bar{\lambda}^2 \mathbb{1}\{\mathcal{E}\}\right]$, we have $\mathbb{E}\left[\frac{1}{n} \sum_{i=1}^{n}(\lambda_i - \bar{\lambda})^2 \mathbb{1}\{\mathcal{E}\}\right] \geq 1$.

Since conditioning on $\mathcal{E}$, $\lambda_i$ is bounded by $L$ for all $i \in [n]$. In order to make $\mathbb{E}\left[\frac{1}{n} \sum_{i=1}^{n}(\lambda_i - \bar{\lambda})^2 \mathbb{1}\{\mathcal{E}\}\right]$ lower bounded by one, there must exist positive constants $\mu_1, \mu_2$ such that with probability at least $\mu_1$, $\mathcal{E}$ holds and $\frac{1}{n} \sum_{i=1}^{n}(\lambda_i - \bar{\lambda})^2 \geq \mu_2$.

Since $\frac{1}{n} \sum_{i=1}^{n}(\lambda_i - \bar{\lambda})^2 \geq \mu_2$ and $\lambda_i \leq L$ for all $i \in [n]$, we know there exists a subset of eigenvalues $S \subset \{\lambda_i\}_1^n$ with size $\mu_3 n$ such that $|\lambda_i - \bar{\lambda}| \geq \mu_4$ for all $\lambda_i \in S$, where $\mu_3, \mu_4$ are both positive constants.

If at least half of eigenvalues in $S$ are larger than $\bar{\lambda}$, we know at least $\frac{\mu_3 \mu_4 n}{2L}$ number of eigenvalues are smaller than $\bar{\lambda}$. Otherwise, the expectation of the eigenvalues will be larger than $\bar{\lambda}$, which contradicts the definition of $\bar{\lambda}$. Similarly, if at least half of eigenvalues in $S$ are smaller than $\bar{\lambda}$, we know at least $\frac{\mu_3 \mu_4 n}{2L}$ number of eigenvalues are larger than $\bar{\lambda}$. Denote $\mu_5 := \frac{\mu_3 \mu_4}{2L}$. We know $\lambda_{\mu_5 n} - \lambda_{n - \mu_5 n + 1} \geq \mu_4$. $\qquad\square$

**Proof of Lemma 23.** Let $\mathcal{E}_1$ be the event that $\sqrt{d}/\sqrt{L} \leq \sigma_i(X_{\text{train}}) \leq \sqrt{Ld}$ and $1/L \leq \lambda_i(H_{\text{train}}) \leq L$ for all $i \in [n]$ and $\sqrt{d}\sigma/4 \leq \|\xi_{\text{train}}\| \leq \sqrt{d}\sigma$. Let $\mathcal{E}_3$ be the event that $\sqrt{d}/\sqrt{L} \leq \sigma_i(X_{\text{valid}}) \leq \sqrt{Ld}$ and $1/L \leq \lambda_i(H_{\text{valid}}) \leq L$ for all $i \in [n]$ and $\sqrt{d}\sigma/4 \leq \|\xi_{\text{valid}}\| \leq \sqrt{d}\sigma$. According to Lemma 1 and Lemma 45, we know both $\mathcal{E}_1$ and $\mathcal{E}_3$ hold with probability at least $1 - \exp(-\Omega(d))$. In this proof, we assume both properties hold and take a union bound at the end.

We can lower bound $\|w_{t,\eta} - w_{\text{valid}}\|_{H_{\text{valid}}}^2$ as follows,

$$\|w_{t,\eta} - w_{\text{valid}}\|_{H_{\text{valid}}}^2 = \|w_{t,\eta} - w^* - (X_{\text{valid}})^{\dagger} \xi_{\text{valid}}\|_{H_{\text{valid}}}^2$$

$$\geq \|w_{t,\eta} - w^*\|_{H_{\text{valid}}}^2 + \frac{1}{n} \|\xi_{\text{valid}}\|^2 - 2 \left|\langle w_{t,\eta} - w^*, H_{\text{valid}}(X_{\text{valid}})^{\dagger} \xi_{\text{valid}}\rangle\right|.$$

For the second term, by Lemma 45, we know for any $1 > \epsilon > 0$, with probability at least $1 - \exp(-\Omega(\epsilon^2 d))$,

$$\frac{1}{n} \|\xi_{\text{valid}}\|^2 \geq (1 - \epsilon)\sigma^2.$$

We can write down the third term as $\left\langle [(X_{\text{valid}})^{\dagger}]^{\top} H_{\text{valid}}(w_{t,\eta} - w^*), \xi_{\text{valid}}\right\rangle$. Suppose $\sigma$ is a constant, we know $\|[(X_{\text{valid}})^{\dagger}]^{\top} H_{\text{valid}}(w_{t,\eta} - w^*)\| = O(1/\sqrt{d})$. Therefore, for a fixed $\eta \in [1/L, 3L]$, we have with probability at least $1 - \exp(-\Omega(\epsilon^2 d))$,

$$\left|\langle w_{t,\eta} - w^*, H_{\text{valid}}(X_{\text{valid}})^{\dagger} \xi_{\text{valid}}\rangle\right| \leq \epsilon.$$

To prove this crossing term is small for all $\eta \in [1/L, 3L]$, we need to construct an $\epsilon$-net for the crossing term. Similar as in Lemma 9, we can show there exists an $\epsilon$-net for the crossing term with size $O(t/\epsilon)$. Taking a union bound over this $\epsilon$-net, we are able to show with probability at least $1 - O(t/\epsilon) \exp(-\Omega(\epsilon^2 d))$,

$$\left| \langle w_{t,\eta} - w^*, H_{\text{valid}}(X_{\text{valid}})^\dagger \xi_{\text{valid}} \rangle \right| \leq \epsilon,$$

for all $\eta \in [1/L, 3L]$.

Overall, we have with probability at least $1 - O(t/\epsilon) \exp(-\Omega(\epsilon^2 d))$,

$$\|w_{t,\eta} - w_{\text{valid}}\|_{H_{\text{valid}}}^2 \geq \|w_{t,\eta} - w^*\|_{H_{\text{valid}}}^2 + \frac{1}{n} \|\xi_{\text{valid}}\|^2 - 2 \left| \langle w_{t,\eta} - w^*, H_{\text{valid}}(X_{\text{valid}})^\dagger \xi_{\text{valid}} \rangle \right|$$

$$\geq \|w_{t,\eta} - w^*\|_{H_{\text{valid}}}^2 + (1 - \epsilon)\sigma^2 - 2\epsilon \geq (1 - 3\epsilon)\sigma^2,$$

for all $\eta \in [1/L, 3L]$, where the last inequality uses $\sigma \geq 1$. The proof finishes as we change $3\epsilon$ to $\epsilon'$. $\qquad\square$

## C   PROOFS OF TRAIN-BY-TRAIN WITH LARGE NUMBER OF SAMPLES (GD)

In this section, we give the proof of Theorem 6. We show when the size of each training set $n$ and the the number of training tasks $m$ are large enough, train-by-train also performs well. Recall Theorem 6 as follows.

**Theorem 6.** *Let $\hat{F}_{TbT(n)}(\eta)$ be as defined in Equation 1. Assume noise level is a constant $c_1$. Given any $1 > \epsilon > 0$, assume training set size $n \geq \frac{cd}{\epsilon^2} \log(\frac{nm}{\epsilon d})$, unroll length $t \geq c_2 \log(\frac{n}{\epsilon d})$, number of training tasks $m \geq \frac{c_3 n^2}{\epsilon^4 d^2} \log(\frac{tnm}{\epsilon d})$ and dimension $d \geq c_4$ for certain constants $c, c_2, c_3, c_4$. With high probability in the sampling of training tasks, we have*

$$\mathbb{E} \left\| w_{t,\eta_{train}^*} - w^* \right\|^2 \leq (1 + \epsilon) \frac{d\sigma^2}{n},$$

*for all $\eta_{train}^* \in \arg\min_{\eta \geq 0} \hat{F}_{TbT(n)}(\eta)$, where the expectation is taken over new tasks.*

In the proof, we use the same notations defined in Section B. On each training task $P$, in Lemma 24 we show the meta-loss can be decomposed into two terms:

$$\Delta_{TbT}(\eta, P) = \frac{1}{2} \|w_{t,\eta} - w_{\text{train}}\|_{H_{\text{train}}}^2 + \frac{1}{2n} \left\| (I_n - \text{Proj}_{X_{\text{train}}})\xi_{\text{train}} \right\|^2,$$

where $w_{\text{train}} = w^* + (X_{\text{train}})^\dagger \xi_{\text{train}}$. Recall that $X_{\text{train}}$ is a $n \times d$ matrix with its $i$-th row as $x_i^\top$. The pseudo-inverse $(X_{\text{train}})^\dagger$ has dimension $d \times n$ satisfying $X_{\text{train}}^\dagger X_{\text{train}} = I_d$. Here, $\text{Proj}_{X_{\text{train}}} \in \mathbb{R}^{n \times n}$ is a projection matrix onto the column span of $X_{\text{train}}$.

In Lemma 24, we show with a constant step size, the first term in $\Delta_{TbT}(\eta, P)$ is exponentially small. The second term is basically the projection of the noise on the orthogonal subspace of the data span. We show this term concentrates well on its mean. This lemma servers as step 1 in Section B.1. The proof of Lemma 24 is deferred into Section C.1.

**Lemma 24.** *Assume $n \geq 40d$. Given any $1 > \epsilon > 0$, with probability at least $1 - m\exp(-\Omega(n)) - \exp(-\Omega(\epsilon^4 md/n))$,*

$$\hat{F}_{TbT}(2/3) \leq 20(1 - \frac{1}{3})^{2t}\sigma^2 + \frac{n-d}{2n}\sigma^2 + \frac{\epsilon^2 d\sigma^2}{20n}.$$

In the next lemma, we show the empirical meta objective is large when $\eta$ exceeds certain threshold. We define this threshold $\hat{\eta}$ such that for any step size larger than $\hat{\eta}$ the GD sequence has reasonable probability being truncated. In the proof, we rely on the truncated sequences to argue the meta-objective must be high. The precise definition of $\hat{\eta}$ is in Definition 2. This lemma serves as step 2 in Section B.1. We leave the proof of Lemma 25 into Section C.2.

**Lemma 25.** *Let $\hat{\eta}$ be as defined in Definition 2 with $1 > \epsilon > 0$. Assume $n \geq cd, t \geq c_2, d \geq c_4$ for some constants $c, c_2, c_4$. With probability at least $1 - \exp(-\Omega(\epsilon^4 md^2/n^2))$,*

$$\hat{F}_{TbT}(\eta) \geq \frac{\epsilon^2 d\sigma^2}{8n} + \frac{n-d}{2n}\sigma^2 - \frac{\epsilon^2 d\sigma^2}{20n},$$

*for all $\eta > \hat{\eta}$.*

By Lemma 24 and Lemma 25, we know when $t$ is reasonably large, $\hat{F}_{TbT}(\eta)$ is larger than $\hat{F}_{TbT}(2/3)$ for all step sizes $\eta > \hat{\eta}$. This means the optimal step size $\hat{\eta}$ must lie in $[0, \hat{\eta}]$. In Lemma 26, we show a generalization result for $\eta \in [0, \hat{\eta}]$. This serves as step 3 in Section B.1. We prove this lemma in Section C.3.

**Lemma 26.** *Let $\hat{\eta}$ be as defined in Definition 2 with $1 > \epsilon > 0$. Suppose $\sigma$ is a constant. Assume $n \geq c\log(\frac{n}{\epsilon d})d, t \geq c_2, d \geq c_4$ for some constants $c, c_2, c_4$. With probability at least $1 - m\exp(-\Omega(n)) - O(\frac{tn}{\epsilon^2 d} + m)\exp(-\Omega(m\epsilon^4 d^2/n^2))$,*

$$|F_{TbT}(\eta) - \hat{F}_{TbT}(\eta)| \leq \frac{17\epsilon^2 d\sigma^2}{n},$$

*for all $\eta \in [0, \hat{\eta}]$,*

Combining Lemma 24, Lemma 25 and Lemma 26, we present the proof of Theorem 6 as follows.

**Proof of Theorem 6.** According to Lemma 24, assuming $n \geq 40d$, given any $1/2 > \epsilon > 0$, with probability at least $1 - m\exp(-\Omega(n)) - \exp(-\Omega(\epsilon^4 md/n))$, $\hat{F}_{TbT}(2/3) \leq 20(1 - \frac{1}{3})^{2t}\sigma^2 + \frac{n-d}{2n}\sigma^2 + \frac{\epsilon^2 d\sigma^2}{20n}$. As long as $t \geq c_2 \log(\frac{n}{\epsilon d})$ for certain constant $c_2$, we have

$$\hat{F}_{TbT}(2/3) \leq \frac{n-d}{2n}\sigma^2 + \frac{7\epsilon^2 d\sigma^2}{100n}.$$

Let $\hat{\eta}$ be as defined in Definition 2 with the same $\epsilon$. According to Lemma 25, as long as $n \geq cd, t \geq c_2, d \geq c_4$ with probability at least $1 - \exp(-\Omega(\epsilon^4 md^2/n^2))$,

$$\hat{F}_{TbT}(\eta) \geq \frac{\epsilon^2 d\sigma^2}{8n} + \frac{n-d}{2n}\sigma^2 - \frac{\epsilon^2 d\sigma^2}{20n} = \frac{n-d}{2n}\sigma^2 + \frac{7.5\epsilon^2 d\sigma^2}{100n}$$

for all $\eta > \hat{\eta}$. We have $\hat{F}_{TbT}(\eta) > \hat{F}_{TbT}(2/3)$ for all $\eta \geq \hat{\eta}$. This implies that $\eta^*_{\text{train}}$ is within $[0, \hat{\eta}]$ and $\hat{F}_{TbT}(\eta^*_{\text{train}}) \leq \hat{F}_{TbT}(2/3) \leq \frac{n-d}{2n}\sigma^2 + \frac{7\epsilon^2 d\sigma^2}{100n}$.

By Lemma 26, assuming $\sigma$ is a constant and assuming $n \geq c\log(\frac{n}{\epsilon d})d$ for some constant $c$, we have with probability at least $1 - m\exp(-\Omega(n)) - O(\frac{tn}{\epsilon^2 d} + m)\exp(-\Omega(m\epsilon^4 d^2/n^2))$,

$$|F_{TbT}(\eta) - \hat{F}_{TbT}(\eta)| \leq \frac{17\epsilon^2 d\sigma^2}{n},$$

for all $\eta \in [0, \hat{\eta}]$. This then implies

$$F_{TbT}(\eta^*_{\text{train}}) \leq \hat{F}_{TbT}(\eta^*_{\text{train}}) + \frac{17\epsilon^2 d\sigma^2}{n} \leq \frac{n-d}{2n}\sigma^2 + \frac{24\epsilon^2 d\sigma^2}{n}.$$

By the analysis in Lemma 24, we have

$$F_{TbT}(\eta^*_{\text{train}}) = \mathbb{E}\frac{1}{2}\left\|w_{t,\eta^*_{\text{train}}} - w_{\text{train}}\right\|^2_{H_{\text{train}}} + \mathbb{E}\frac{1}{2n}\left\|(I_n - \text{Proj}_{X_{\text{train}}})\xi_{\text{train}}\right\|^2$$

$$= \mathbb{E}\frac{1}{2}\left\|w_{t,\eta^*_{\text{train}}} - w_{\text{train}}\right\|^2_{H_{\text{train}}} + \frac{n-d}{2n}\sigma^2.$$

Therefore, we know $\mathbb{E}\frac{1}{2}\left\|w_{t,\eta^*_{\text{train}}} - w_{\text{train}}\right\|^2_{H_{\text{train}}} \leq \frac{24\epsilon^2 d\sigma^2}{n}$. Next, we show this implies $\mathbb{E}\left\|w_{t,\eta^*_{\text{train}}} - w^*\right\|^2$ is small.

Let $\mathcal{E}$ be the event that $1 - \epsilon \leq \lambda_i(H_{\text{train}}) \leq 1 + \epsilon$ for all $i \in [d]$. According to Lemma 27, we know $\Pr[\mathcal{E}] \geq 1 - \exp(-\Omega(\epsilon^2 n))$ as long as $n \geq 10d/\epsilon^2$. Then, we can decompose $\mathbb{E}\left\|w_{t,\eta^*_{\text{train}}} - w^*\right\|^2$ as follows,

$$\mathbb{E}\left\|w_{t,\eta^*_{\text{train}}} - w^*\right\|^2 = \mathbb{E}\left\|w_{t,\eta^*_{\text{train}}} - w^*\right\|^2 \mathbb{1}\{\mathcal{E}\} + \mathbb{E}\left\|w_{t,\eta^*_{\text{train}}} - w^*\right\|^2 \mathbb{1}\{\bar{\mathcal{E}}\}.$$

Let's first show the second term is small. Due to the truncation in our algorithm, we know $\left\|w_{t,\eta^*_{\text{train}}} - w^*\right\|^2 \leq 41^2\sigma^2$, which then implies $\mathbb{E}\left\|w_{t,\eta^*_{\text{train}}} - w^*\right\|^2 \mathbb{1}\{\bar{\mathcal{E}}\} \leq 41^2\sigma^2 \exp(-\Omega(\epsilon^2 n))$. As long as $n \geq \frac{c}{\epsilon^2}\log(\frac{n}{\epsilon d})$ for some constant $c$, we have $\mathbb{E}\left\|w_{t,\eta^*_{\text{train}}} - w^*\right\|^2 \mathbb{1}\{\bar{\mathcal{E}}\} \leq \frac{\epsilon d\sigma^2}{n}$.

We can upper bound the first term by Young's inequality,

$$\mathbb{E} \left\| w_{t,\eta^*_{\text{train}}} - w^* \right\|^2 \mathbb{1}\{\mathcal{E}\} \le (1 + \frac{1}{\epsilon})\mathbb{E} \left\| w_{t,\eta^*_{\text{train}}} - w_{\text{train}} \right\|^2 \mathbb{1}\{\mathcal{E}\} + (1+\epsilon)\mathbb{E} \left\| w_{\text{train}} - w^* \right\|^2 \mathbb{1}\{\mathcal{E}\}.$$

Conditioning on $\mathcal{E}$, we have $\left\| w_{t,\eta^*_{\text{train}}} - w_{\text{train}} \right\|^2_{H_{\text{train}}} \ge (1-\epsilon) \left\| w_{t,\eta^*_{\text{train}}} - w_{\text{train}} \right\|^2$ which implies $\left\| w_{t,\eta^*_{\text{train}}} - w_{\text{train}} \right\|^2 \le (1+2\epsilon) \left\| w_{t,\eta^*_{\text{train}}} - w_{\text{train}} \right\|^2_{H_{\text{train}}}$ as long as $\epsilon \le 1/2$. Similarly, we also have $\left\| w_{\text{train}} - w^* \right\|^2 \le (1+2\epsilon) \left\| w_{\text{train}} - w^* \right\|^2_{H_{\text{train}}}$. Then, we have

$$\begin{aligned} &\mathbb{E} \left\| w_{t,\eta^*_{\text{train}}} - w^* \right\|^2 \mathbb{1}\{\mathcal{E}\} \\ \le &(1 + \frac{1}{\epsilon})(1+2\epsilon)\mathbb{E} \left\| w_{t,\eta^*_{\text{train}}} - w_{\text{train}} \right\|^2_{H_{\text{train}}} \mathbb{1}\{\mathcal{E}\} + (1+\epsilon)(1+2\epsilon)\mathbb{E} \left\| w_{\text{train}} - w^* \right\|^2_{H_{\text{train}}} \mathbb{1}\{\mathcal{E}\} \\ \le &(5 + \frac{1}{\epsilon})\mathbb{E} \left\| w_{t,\eta^*_{\text{train}}} - w_{\text{train}} \right\|^2_{H_{\text{train}}} + (1+5\epsilon)\mathbb{E} \left\| w_{\text{train}} - w^* \right\|^2_{H_{\text{train}}} \\ \le &(5 + \frac{1}{\epsilon})\frac{48\epsilon^2 d\sigma^2}{n} + (1+5\epsilon)\frac{d\sigma^2}{n} \le (1 + 293\epsilon)\frac{d\sigma^2}{n}. \end{aligned}$$

Overall, we have $\mathbb{E} \left\| w_{t,\eta^*_{\text{train}}} - w^* \right\|^2 \le (1+293\epsilon)\frac{d\sigma^2}{n} + \frac{\epsilon d\sigma^2}{n} = (1+294\epsilon)\frac{d\sigma^2}{n}$. Combining all the conditions, we know this holds with probability at least $0.99$ as long as $\sigma$ is a constant $c_1$, $n \ge \frac{cd}{\epsilon^2}\log(\frac{nm}{\epsilon d}), t \ge c_2 \log(\frac{n}{\epsilon d}), m \ge \frac{c_3 n^2}{\epsilon^4 d^2}\log(\frac{tnm}{\epsilon d}), d \ge c_4$ for some constants $c, c_2, c_3, c_4$. We finish the proof by choosing $\epsilon = \epsilon'/294$. $\qquad\square$

## C.1 UPPER BOUNDING $\hat{F}_{TbT}(2/3)$

In this section, we show there exists a step size that achieves small empirical meta objective. On each training task $P$, we show the meta-loss can be decomposed into two terms:

$$\begin{aligned} \Delta_{TbT}(\eta, P) =& \frac{1}{2n}\sum_{i=1}^{n} \left( \langle w_{t,\eta} - w_{\text{train}}, x_i \rangle - \left( \xi_i - x_i^\top X_{\text{train}}^\dagger \xi_{\text{train}} \right) \right)^2 \\ =& \frac{1}{2} \left\| w_{t,\eta} - w_{\text{train}} \right\|^2_{H_{\text{train}}} + \frac{1}{2n} \left\| (I_n - \text{Proj}_{X_{\text{train}}})\xi_{\text{train}} \right\|^2, \end{aligned}$$

where $w_{\text{train}} = w^* + (X_{\text{train}})^\dagger \xi_{\text{train}}$. In Lemma 24, we show with a constant step size, the first term is exponentially small and the second term concentrates on its mean.

**Lemma 24.** *Assume $n \ge 40d$. Given any $1 > \epsilon > 0$, with probability at least $1 - m\exp(-\Omega(n)) - \exp(-\Omega(\epsilon^4 md/n))$,*

$$\hat{F}_{TbT}(2/3) \le 20(1 - \frac{1}{3})^{2t}\sigma^2 + \frac{n-d}{2n}\sigma^2 + \frac{\epsilon^2 d\sigma^2}{20n}.$$

Before we go to the proof of Lemma 24, let's first show the covariance matrix $H_{\text{train}}$ is very close to identity when $n$ is much larger than $d$. The proof follows from the concentration of singular values of random Gaussian matrix (Lemma 48). We leave the proof into Section C.4.

**Lemma 27.** *Given $1 > \epsilon > 0$, assume $n \ge 10d/\epsilon^2$. With probability at least $1 - \exp(-\Omega(\epsilon^2 n))$,*

$$(1-\epsilon)\sqrt{n} \le \sigma_i(X_{train}) \le (1+\epsilon)\sqrt{n} \text{ and } 1 - \epsilon \le \lambda_i(H_{train}) \le 1 + \epsilon,$$

*for all $i \in [d]$.*

Now, we are ready to present the proof of Lemma 24.

**Proof of Lemma 24.** Let's first look at one training set $S_{\text{train}}$, in which $y_i = \langle w^*, x_i \rangle + \xi_i$ for each sample. Recall the meta-loss as

$$\Delta_{TbT}(\eta, P) = \frac{1}{2n}\sum_{i=1}^{n} (\langle w_{t,\eta}, x_i \rangle - \langle w^*, x_i \rangle - \xi_i)^2.$$

Recall that $X_{\text{train}}$ is an $n \times d$ matrix with its $i$-th row as $x_i^\top$. With probability 1, we know $X_{\text{train}}$ is full column rank. Denote the pseudo-inverse of $X_{\text{train}}$ as $X_{\text{train}}^\dagger \in \mathbb{R}^{d \times n}$ that satisfies $X_{\text{train}}^\dagger X_{\text{train}} = I_d$ and $X_{\text{train}} X_{\text{train}}^\dagger = \text{Proj}_{X_{\text{train}}}$, where $\text{Proj}_{X_{\text{train}}} \in \mathbb{R}^{n \times n}$ is a projection matrix onto the column span of $X_{\text{train}}$.

Let $w_{\text{train}}$ be $w^* + X_{\text{train}}^\dagger \xi_{\text{train}}$, where $\xi_{\text{train}}$ is an $n$-dimensional vector with its $i$-th entry as $\xi_i$. We have,

$$\Delta_{TbT}(\eta, P)$$
$$= \frac{1}{2n} \sum_{i=1}^n \left( \langle w_{t,\eta} - w_{\text{train}}, x_i \rangle - \left( \xi_i - x_i^\top X_{\text{train}}^\dagger \xi_{\text{train}} \right) \right)^2$$
$$= \frac{1}{2} \left\| w_{t,\eta} - w_{\text{train}} \right\|_{H_{\text{train}}}^2 + \frac{1}{2n} \left\| (I_n - \text{Proj}_{X_{\text{train}}}) \xi_{\text{train}} \right\|^2 - \frac{1}{n} \sum_{i=1}^n \left\langle w_{t,\eta} - w_{\text{train}}, x_i \xi_i - x_i x_i^\top X_{\text{train}}^\dagger \xi_{\text{train}} \right\rangle.$$

We first show the crossing term is actually zero. We have,

$$\frac{1}{n} \sum_{i=1}^n \left\langle w_{t,\eta} - w_{\text{train}}, x_i \xi_i - x_i x_i^\top X_{\text{train}}^\dagger \xi_{\text{train}} \right\rangle = \frac{1}{n} \left\langle w_{t,\eta} - w_{\text{train}}, \sum_{i=1}^n x_i \xi_i - \sum_{i=1}^n x_i x_i^\top X_{\text{train}}^\dagger \xi_{\text{train}} \right\rangle$$
$$= \frac{1}{n} \left\langle w_{t,\eta} - w_{\text{train}}, X_{\text{train}}^\top \xi_{\text{train}} - X_{\text{train}}^\top X_{\text{train}} X_{\text{train}}^\dagger \xi_{\text{train}} \right\rangle$$
$$= \frac{1}{n} \left\langle w_{t,\eta} - w_{\text{train}}, X_{\text{train}}^\top \xi_{\text{train}} - X_{\text{train}}^\top \xi_{\text{train}} \right\rangle = 0,$$

where the second last equality holds because $X_{\text{train}} X_{\text{train}}^\dagger = \text{Proj}_{X_{\text{train}}}$.

We can define $w_{\text{train}}^{(k)}$ as $w_k^* + (X_{\text{train}}^{(k)})^\dagger \xi_{\text{train}}^{(k)}$ for every training set $S_{\text{train}}^{(k)}$. Then, we have

$$\hat{F}_{TbT}(\eta) = \frac{1}{m} \sum_{k=1}^m \frac{1}{2} \left\| w_{t,\eta}^{(k)} - w_{\text{train}}^{(k)} \right\|_{H_{\text{train}}^{(k)}}^2 + \frac{1}{m} \sum_{k=1}^m \frac{1}{2n} \left\| (I_n - \text{Proj}_{X_{\text{train}}^{(k)}}) \xi_{\text{train}}^{(k)} \right\|^2$$

We first prove that the second term concentrates on its mean. We can concatenate $m$ noise vectors $\xi_{\text{train}}^{(k)}$ into a single noise vector $\bar{\xi}_{\text{train}}$ with dimension $nm$. We can also construct a data matrix $\bar{X}_{\text{train}} \in \mathbb{R}^{nm \times dm}$ that consists of $X_{\text{train}}^{(k)}$ as diagonal blocks. Then the second term can be written as

$$\frac{1}{2} \left\| \frac{1}{\sqrt{nm}} (I_{nm} - \text{Proj}_{\bar{X}_{\text{train}}}) \bar{\xi}_{\text{train}} \right\|^2.$$

According to Lemma 45, with probability at least $1 - \exp(-\Omega(\epsilon^4 m d^2 / n))$,

$$\left( 1 - \frac{\epsilon^2 d}{n} \right) \sigma \leq \frac{1}{\sqrt{nm}} \left\| \bar{\xi}_{\text{train}} \right\| \leq \left( 1 + \frac{\epsilon^2 d}{n} \right) \sigma.$$

By Johnson-Lindenstrauss Lemma (Lemma 49), we know with probability at least $1 - \exp(-\Omega(\epsilon^4 m d))$,

$$\frac{1}{\sqrt{nm}} \left\| \text{Proj}_{\bar{X}_{\text{train}}} \bar{\xi}_{\text{train}} \right\| \geq (1 - \epsilon^2) \frac{\sqrt{md}}{\sqrt{mn}} \frac{1}{\sqrt{nm}} \left\| \bar{\xi}_{\text{train}} \right\| \geq (1 - \epsilon^2) \sqrt{\frac{d}{n}} (1 - \frac{\epsilon^2 d}{n}) \sigma.$$

Therefore, we have $\left\| \frac{1}{\sqrt{nm}} \bar{\xi}_{\text{train}} \right\|^2 \leq (1 + \frac{3\epsilon^2 d}{n}) \sigma^2$ and $\left\| \frac{1}{\sqrt{nm}} \text{Proj}_{\bar{X}_{\text{train}}} \bar{\xi}_{\text{train}} \right\|^2 \geq (1 - 2\epsilon^2) \frac{d}{n} \sigma^2$. Overall, we know with probability at least $1 - \exp(-\Omega(\epsilon^4 m d / n))$,

$$\frac{1}{2} \left\| \frac{1}{\sqrt{nm}} (I_{nm} - \text{Proj}_{\bar{X}_{\text{train}}}) \bar{\xi}_{\text{train}} \right\|^2 \leq \frac{n - d}{2n} \sigma^2 + \frac{5 \epsilon^2 d \sigma^2}{2n}.$$

Now, we show the first term in meta objective is small when we choose a right step size. According to Lemma 27, we know as long as $n \geq 40d$, with probability at least $1 - \exp(-\Omega(n))$, $\sqrt{n}/2 \leq$

$\sigma_i(X_{\text{train}}^{(k)}) \leq 3\sqrt{n}/2$ and $1/2 \leq \lambda_i(H_{\text{train}}^{(k)}) \leq 3/2$, for all $i \in [d]$. According to Lemma 45, we know with probability at least $1 - \exp(-\Omega(n))$, $\left\|\xi_{\text{train}}^{(k)}\right\| \leq 2\sqrt{n}\sigma$. Taking a union bound on $m$ tasks, we know all these events hold with probability at least $1 - m\exp(-\Omega(n))$.

For each $k \in [m]$, we have $\left\|w_{\text{train}}^{(k)}\right\| \leq 1 + \frac{2}{\sqrt{n}} 2\sqrt{n}\sigma \leq 5\sigma$. It's easy to verify that for any step size at most $2/3$, the GD sequence will not be truncated since we choose the threshold norm as $40\sigma$. Then, for any step size $\eta \leq 2/3$, we have

$$\frac{1}{m}\sum_{k=1}^m \frac{1}{2}\left\|w_{t,\eta}^{(k)} - w_{\text{train}}^{(k)}\right\|_{H_{\text{train}}^{(k)}}^2 = \frac{1}{m}\sum_{k=1}^m \frac{1}{2}\left\|(I - \eta H_{\text{train}}^{(k)})^t w_{\text{train}}^{(k)}\right\|_{H_{\text{train}}^{(k)}}^2$$

$$\leq \frac{3}{4}(1 - \frac{\eta}{2})^{2t} 25\sigma^2 \leq 20(1 - \frac{1}{3})^{2t}\sigma^2,$$

where the last inequality chooses $\eta$ as $2/3$.

Overall, we know with probability at least $1 - m\exp(-\Omega(n)) - \exp(-\Omega(\epsilon^4 md/n))$,

$$\hat{F}_{TbT}(2/3) \leq 20(1 - \frac{1}{3})^{2t}\sigma^2 + \frac{n-d}{2n}\sigma^2 + \frac{5\epsilon^2 d\sigma^2}{2n}.$$

We finish the proof by changing $\frac{5\epsilon^2}{2}$ by $(\epsilon')^2/20$. $\qquad\square$

## C.2 LOWER BOUNDING $\hat{F}_{TbT}$ FOR $\eta \in (\hat{\eta}, \infty)$

In this section, we show the empirical meta objective is large when the step size exceeds certain threshold. Recall Lemma 25 as follows.

**Lemma 25.** *Let $\hat{\eta}$ be as defined in Definition 2 with $1 > \epsilon > 0$. Assume $n \geq cd, t \geq c_2, d \geq c_4$ for some constants $c, c_2, c_4$. With probability at least $1 - \exp(-\Omega(\epsilon^4 md^2/n^2))$,*

$$\hat{F}_{TbT}(\eta) \geq \frac{\epsilon^2 d\sigma^2}{8n} + \frac{n-d}{2n}\sigma^2 - \frac{\epsilon^2 d\sigma^2}{20n},$$

*for all $\eta > \hat{\eta}$.*

Roughly speaking, we define $\hat{\eta}$ such that for any step size larger than $\hat{\eta}$ the GD sequence has a reasonable probability being truncated. The definition is very similar as $\tilde{\eta}$ in Definition 1.

**Definition 2.** *Given a training task $P$, let $\mathcal{E}_1$ be the event that $\sqrt{n}/2 \leq \sigma_i(X_{train}) \leq 3\sqrt{n}/2$ and $1/2 \leq \lambda_i(H_{train}) \leq 3/2$ for all $i \in [d]$ and $\sqrt{n}\sigma/2 \leq \|\xi_{train}\| \leq 2\sqrt{n}\sigma$. Let $\bar{\mathcal{E}}_2(\eta)$ be the event that the GD sequence is truncated with step size $\eta$. Given $1 > \epsilon > 0$, define $\hat{\eta}$ as follows,*

$$\hat{\eta} = \inf\left\{\eta \geq 0 \middle| \mathbb{E}\frac{1}{2}\|w_{t,\eta} - w_{train}\|_{H_{train}}^2 \mathbb{1}\left\{\mathcal{E}_1 \cap \bar{\mathcal{E}}_2(\eta)\right\} \geq \frac{\epsilon^2 d\sigma^2}{n}\right\}.$$

Similar as in Lemma 5, we show $\mathbb{1}\left\{\mathcal{E}_1 \cap \bar{\mathcal{E}}_2(\eta')\right\} \geq \mathbb{1}\left\{\mathcal{E}_1 \cap \bar{\mathcal{E}}_2(\eta)\right\}$ for any $\eta' \geq \eta$. This means conditioning on $\mathcal{E}_1$, if a GD sequence gets truncated with step size $\eta$, it has to be truncated with any step size $\eta' \geq \eta$. The proof is deferred into Section C.4.

**Lemma 28.** *Fixing a training set $S_{train}$, let $\mathcal{E}_1$ and $\bar{\mathcal{E}}_2(\eta)$ be as defined in Definition 2. We have*

$$\mathbb{1}\left\{\mathcal{E}_1 \cap \bar{\mathcal{E}}_2(\eta')\right\} \geq \mathbb{1}\left\{\mathcal{E}_1 \cap \bar{\mathcal{E}}_2(\eta)\right\},$$

*for any $\eta' \geq \eta$.*

Next, we show $\hat{\eta}$ does exist and is a constant. Similar as in Lemma 6, we show that the GD sequence almost never diverges when $\eta$ is small and diverges with high probability when $\eta$ is large. The proof is left in Section C.4.

**Lemma 29.** *Let $\hat{\eta}$ be as defined in Definition 2. Suppose $\sigma$ is a constant. Assume $n \geq cd, t \geq c_2, d \geq c_4$ for some constants $c, c_2, c_4$. We have*

$$\frac{4}{3} < \tilde{\eta} < 6.$$

Next, we show the empirical loss is large for any $\eta$ larger than $\tilde{\eta}$. The proof is very similar as the proof of Lemma 3.

**Proof of Lemma 25.** By Lemma 29, we know $\hat{\eta}$ is a constant as long as $n \geq cd, t \geq c_2, d \geq c_4$ for some constants $c, c_2, c_4$. Let $\mathcal{E}_1$ and $\bar{\mathcal{E}}_2(\eta)$ be as defined in Definition 2. For the simplicity of the proof, we assume $\mathbb{E} \frac{1}{2} \|w_{t,\hat{\eta}} - w_{\text{train}}\|_{H_{\text{train}}}^2 \mathbb{1}\left\{\mathcal{E}_1 \cap \bar{\mathcal{E}}_2(\hat{\eta})\right\} \geq \frac{\epsilon^2 d\sigma^2}{n}$. The other case can be resolved using same techniques in Lemma 3

Conditioning on $\mathcal{E}_1$, we know $\frac{1}{2}\|w_{t,\hat{\eta}} - w_{\text{train}}\|_{H_{\text{train}}}^2 \leq \frac{3}{4}45^2\sigma^2$. Therefore, we know $\Pr[\mathcal{E}_1 \cap \bar{\mathcal{E}}_2(\hat{\eta})] \geq \frac{4\epsilon^2 d}{3 \times 45^2 n}$. For each task $k$, define $\mathcal{E}_1^{(k)}$ and $\bar{\mathcal{E}}_2^{(k)}(\eta)$ as the corresponding events on training set $S_{\text{train}}^{(k)}$. By Hoeffding's inequality, we know with probability at least $1 - \exp(-\Omega(\epsilon^4 md^2/n^2))$,

$$\frac{1}{m}\sum_{k=1}^m \mathbb{1}\left\{\mathcal{E}_1^{(k)} \cap \bar{\mathcal{E}}_2^{(k)}(\hat{\eta})\right\} \geq \frac{\epsilon^2 d}{45^2 n}.$$

By Lemma 28, we know $\mathbb{1}\left\{\mathcal{E}_1^{(k)} \cap \bar{\mathcal{E}}_2^{(k)}(\eta)\right\} \geq \mathbb{1}\left\{\mathcal{E}_1^{(k)} \cap \bar{\mathcal{E}}_2^{(k)}(\hat{\eta})\right\}$ for any $\eta \geq \hat{\eta}$.

Recall that

$$\hat{F}_{TbT}(\eta) = \frac{1}{m}\sum_{k=1}^m \frac{1}{2}\left\|w_{t,\eta}^{(k)} - w_{\text{train}}^{(k)}\right\|_{H_{\text{train}}^{(k)}}^2 + \frac{1}{m}\sum_{k=1}^m \frac{1}{2n}\left\|(I_n - \text{Proj}_{X_{\text{train}}^{(k)}})\xi_{\text{train}}^{(k)}\right\|^2.$$

We can lower bound the first term for any $\eta > \hat{\eta}$ as follows,

$$\begin{aligned}
\hat{F}_{TbT}(\eta) = \frac{1}{m}\sum_{k=1}^m \frac{1}{2}\left\|w_{t,\eta}^{(k)} - w_{\text{train}}^{(k)}\right\|_{H_{\text{train}}^{(k)}}^2 &\geq \frac{1}{m}\sum_{k=1}^m \frac{1}{2}\left\|w_{t,\eta}^{(k)} - w_{\text{train}}^{(k)}\right\|_{H_{\text{train}}^{(k)}}^2 \mathbb{1}\left\{\mathcal{E}_1^{(k)} \cap \bar{\mathcal{E}}_2^{(k)}(\eta)\right\} \\
&\geq \frac{35^2\sigma^2}{4}\frac{1}{m}\sum_{k=1}^m \mathbb{1}\left\{\mathcal{E}_1^{(k)} \cap \bar{\mathcal{E}}_2^{(k)}(\eta)\right\} \\
&\geq \frac{35^2\sigma^2}{4}\frac{1}{m}\sum_{k=1}^m \mathbb{1}\left\{\mathcal{E}_1^{(k)} \cap \bar{\mathcal{E}}_2^{(k)}(\hat{\eta})\right\} \geq \frac{\epsilon^2 d\sigma^2}{8n},
\end{aligned}$$

where the second inequality lower bounds the loss for one task by $35^2\sigma^2$ when the sequence gets truncated.

For the second term, according to the analysis in Lemma 24, with probability at least $1 - \exp(-\Omega(\epsilon^4 md/n))$,

$$\frac{1}{m}\sum_{k=1}^m \frac{1}{2n}\left\|(I_n - \text{Proj}_{X_{\text{train}}^{(k)}})\xi_{\text{train}}^{(k)}\right\|^2 \geq \frac{n-d}{2n}\sigma^2 - \frac{\epsilon^2 d\sigma^2}{20n}.$$

Overall, with probability at least $1 - \exp(-\Omega(\epsilon^4 md^2/n^2))$,

$$\hat{F}_{TbT}(\eta) \geq \frac{\epsilon^2 d\sigma^2}{8n} + \frac{n-d}{2n}\sigma^2 - \frac{\epsilon^2 d\sigma^2}{20n},$$

for all $\eta > \hat{\eta}$. $\qquad\square$

### C.3 GENERALIZATION FOR $\eta \in [0, \hat{\eta}]$

Combing Lemma 24 and Lemma 25, it's not hard to see that the optimal step size $\eta_{\text{train}}^*$ lies in $[0, \hat{\eta}]$. In this section, we show a generalization result for step sizes in $[0, \hat{\eta}]$. The proof of Lemma 26 is given at the end of this section.

**Lemma 26.** *Let $\hat{\eta}$ be as defined in Definition 2 with $1 > \epsilon > 0$. Suppose $\sigma$ is a constant. Assume $n \geq c\log(\frac{n}{\epsilon d})d, t \geq c_2, d \geq c_4$ for some constants $c, c_2, c_4$. With probability at least $1 - m\exp(-\Omega(n)) - O(\frac{tn}{\epsilon^2 d} + m)\exp(-\Omega(m\epsilon^4 d^2/n^2))$,*

$$|F_{TbT}(\eta) - \hat{F}_{TbT}(\eta)| \leq \frac{17\epsilon^2 d\sigma^2}{n},$$

*for all $\eta \in [0, \hat{\eta}]$,*

In Lemma 30, we show $\hat{F}_{TbT}$ concentrates on $F_{TbT}$ at any fixed step size. The proof is almost the same as Lemma 7. We omit its proof.

**Lemma 30.** *Suppose $\sigma$ is a constant. For any fixed $\eta$ and any $1 > \epsilon > 0$, with probability at least $1 - \exp(-\Omega(\epsilon^2 m))$,*

$$\left|\hat{F}_{TbT}(\eta) - F_{TbT}(\eta)\right| \leq \epsilon.$$

Next, we construct an $\epsilon$-net for $F_{TbT}$ in $[0, \hat{\eta}]$. The proof is very similar as in Lemma 8. We defer its proof into Section C.4.

**Lemma 31.** *Let $\hat{\eta}$ be as defined in Definition 2 with $1 > \epsilon > 0$. Assume the conditions in Lemma 29 hold. Assume $n \geq c \log(\frac{n}{\epsilon d})d$ for some constant $c$. There exists an $\frac{8\epsilon^2 d\sigma^2}{n}$-net $N \subset [0, \hat{\eta}]$ for $F_{TbT}$ with $|N| = O(\frac{tn}{\epsilon^2 d})$. That means, for any $\eta \in [0, \hat{\eta}]$,*

$$|F_{TbT}(\eta) - F_{TbT}(\eta')| \leq \frac{8\epsilon^2 d\sigma^2}{n},$$

*for $\eta' = \arg\min_{\eta'' \in N, \eta'' \leq \eta}(\eta - \eta'')$.*

We also construct an $\epsilon$-net for the empirical meta objective. The proof is very similar as in Lemma 9. We leave its proof into Section C.4.

**Lemma 32.** *Let $\hat{\eta}$ be as defined in Definition 2 with $1 > \epsilon > 0$. Assume the conditions in Lemma 29 hold. Assume $n \geq 40d$. With probability at least $1 - m\exp(-\Omega(n))$, there exists an $\frac{\epsilon^2 d\sigma^2}{n}$-net $N' \subset [0, \hat{\eta}]$ for $\hat{F}_{TbT}$ with $|N'| = O(\frac{tn}{\epsilon^2 d} + m)$. That means, for any $\eta \in [0, \hat{\eta}]$,*

$$|\hat{F}_{TbT}(\eta) - \hat{F}_{TbT}(\eta')| \leq \frac{\epsilon^2 d\sigma^2}{n},$$

*for $\eta' = \arg\min_{\eta'' \in N', \eta'' \leq \eta}(\eta - \eta'')$.*

Combing the above three lemmas, we give the proof of Lemma 26.

**Proof of Lemma 26.** We assume $\sigma$ as a constant in this proof. By Lemma 30, we know with probability at least $1 - \exp(-\Omega(m\epsilon^4 d^2/n^2))$, $\left|\hat{F}_{TbT}(\eta) - F_{TbT}(\eta)\right| \leq \frac{\epsilon^2 d\sigma^2}{n}$ for any fixed $\eta$. By Lemma 31, we know as long as $n \geq c \log(\frac{n}{\epsilon d})d$ for some constant $c$, there exists an $\frac{8\epsilon^2 d\sigma^2}{n}$-net $N$ for $F_{TbT}$ with size $O(\frac{tn}{\epsilon^2 d})$. By Lemma 32, we know with probability at least $1 - m\exp(-\Omega(n))$, there exists an $\frac{\epsilon^2 d\sigma^2}{n}$-net $N'$ for $\hat{F}_{TbT}$ with size $O(\frac{tn}{\epsilon^2 d} + m)$. It's not hard to verify that $N \cup N'$ is still an $\frac{8\epsilon^2 d\sigma^2}{n}$-net for $\hat{F}_{TbV}$ and $F_{TbV}$. That means, for any $\eta \in [0, \hat{\eta}]$, we have

$$|F_{TbT}(\eta) - F_{TbT}(\eta')|, |\hat{F}_{TbT}(\eta) - \hat{F}_{TbT}(\eta')| \leq \frac{8\epsilon^2 d\sigma^2}{n},$$

for $\eta' = \arg\min_{\eta'' \in N \cup N', \eta'' \leq \eta}(\eta - \eta'')$.

Taking a union bound over $N \cup N'$, we have with probability at least $1 - O(\frac{tn}{\epsilon^2 d} + m)\exp(-\Omega(m\epsilon^4 d^2/n^2))$,

$$\left|\hat{F}_{TbT}(\eta) - F_{TbT}(\eta)\right| \leq \frac{\epsilon^2 d\sigma^2}{n}$$

for all $\eta \in N \cup N'$.

Overall, we know with probability at least $1 - m\exp(-\Omega(n)) - O(\frac{tn}{\epsilon^2 d} + m)\exp(-\Omega(m\epsilon^4 d^2/n^2))$, for all $\eta \in [0, \hat{\eta}]$,

$$\begin{aligned}
&|F_{TbT}(\eta) - \hat{F}_{TbT}(\eta)| \\
\leq& |F_{TbT}(\eta) - F_{TbT}(\eta')| + |\hat{F}_{TbT}(\eta) - \hat{F}_{TbT}(\eta')| + |\hat{F}_{TbT}(\eta') - F_{TbT}(\eta')| \\
\leq& \frac{17\epsilon^2 d\sigma^2}{n},
\end{aligned}$$

where $\eta' = \arg\min_{\eta'' \in N \cup N', \eta'' \leq \eta}(\eta - \eta'')$. $\quad\square$

### C.4 PROOFS OF TECHNICAL LEMMAS

**Proof of Lemma 27.** According to Lemma 48, we know with probability at least $1 - 2\exp(-t^2/2)$,

$$\sqrt{n} - \sqrt{d} - t \leq \sigma_i(X_{\text{train}}) \leq \sqrt{n} + \sqrt{d} + t$$

for all $i \in [d]$. Since $d \leq \frac{\epsilon^2 n}{10}$, we have $\sqrt{n} - \frac{\epsilon\sqrt{n}}{\sqrt{10}} - t \leq \sigma_i(X_{\text{train}}) \leq \sqrt{n} + \frac{\epsilon\sqrt{n}}{\sqrt{10}} + t$. Choosing $t = (\frac{1}{3} - \frac{1}{\sqrt{10}})\epsilon\sqrt{n}$, we have with probability at least $1 - \exp(-\Omega(\epsilon^2 n))$,

$$(1 - \frac{\epsilon}{3})\sqrt{n} \leq \sigma_i(X_{\text{train}}) \leq (1 + \frac{\epsilon}{3})\sqrt{n}.$$

Since $\lambda_i(H_{\text{train}}) = 1/n\sigma_i^2(X_{\text{train}})$, we have $1 - \epsilon \leq \lambda_i(H_{\text{train}}) \leq 1 + \epsilon$. $\qquad\square$

**Proof of Lemma 28.** The proof is almost the same as in Lemma 5. We omit the details here. Basically, in Lemma 5, the only property we rely on is that the norm threshold is larger than $2\|w_{\text{train}}\|$ conditioning on $\mathcal{E}_1$. Conditioning on $\mathcal{E}_1$, we know $\|w_{\text{train}}\| \leq 5\sigma$. Recall that the norm threshold is still set as $40\sigma$. So this property is preserved and the previous proof works. $\qquad\square$

**Proof of Lemma 29.** The proof is very similar as in Lemma 6. Conditioning on $\mathcal{E}_1$, we know $\|H_{\text{train}}\| \leq 3/2$ and $\|w_{\text{train}}\| \leq 5\sigma$. So the GD sequence never exceeds the norm threshold $40\sigma$ for any $\eta \leq 4/3$. That means,

$$\mathbb{E}\frac{1}{2}\|w_{t,\eta} - w_{\text{train}}\|_{H_{\text{train}}}^2 \mathbb{1}\{\mathcal{E}_1 \cap \bar{\mathcal{E}}_2(\eta)\} = 0$$

for all $\eta \leq 4/3$.

To lower bound the loss for large step size, we need to first lower bound $\|w_{\text{train}}\|$. Recall that $w_{\text{train}} = w^* + (X_{\text{train}})^\dagger \xi_{\text{train}}$. Conditioning on $\mathcal{E}_1$, we know $\|\xi_{\text{train}}\| \leq 2\sqrt{n}\sigma$ and $\sigma_d(X_{\text{train}}) \geq \sqrt{n}/2$, which implies $\|(X_{\text{train}})^\dagger\| \leq 2/\sqrt{n}$. By Johnson-Lindenstrauss Lemma (Lemma 49), we have $\|\text{Proj}_{X_{\text{train}}}\xi_{\text{train}}\| \leq \frac{3}{2}\sqrt{d/n}\|\xi_{\text{train}}\|$ with probability at least $1 - \exp(-\Omega(d))$. Call this event $\mathcal{E}_3$. Conditioning on $\mathcal{E}_1 \cap \mathcal{E}_3$, we have

$$\|(X_{\text{train}})^\dagger \xi_{\text{train}}\| \leq 2\sqrt{n}\sigma\frac{2}{\sqrt{n}}\frac{3}{2}\sqrt{\frac{d}{n}} \leq 6\sqrt{\frac{d}{n}}\sigma,$$

which is smaller than $1/2$ as long as $n \geq 12^2 d\sigma^2$. Note that we assume $\sigma$ is a constant. This then implies $\|w_{\text{train}}\| \geq 1/2$.

Let $\{w'_{\tau,\eta}\}$ be the GD sequence without truncation. For any step size $\eta \in [6, \infty]$, conditioning on $\mathcal{E}_1 \cap \mathcal{E}_3$, we have

$$\|w'_{t,\eta}\| \geq \left((6 \times \frac{1}{2} - 1)^t - 1\right)\|w_{\text{train}}\| \geq (2^t - 1)\frac{1}{2} \geq 40\sigma,$$

where the last inequality holds as long as $t \geq c_2$ for some constant $c_2$. Therefore, we know when $\eta \in [6, \infty)$, $\mathbb{1}\{\mathcal{E}_1 \cap \bar{\mathcal{E}}_2(\eta)\} = \mathbb{1}\{\mathcal{E}_1 \cap \mathcal{E}_3\}$. Assuming $n \geq 40d$, we know $\mathcal{E}_1$ holds with probability at least $1 - \exp(-\Omega(n))$. Then, we have for any $\eta \geq 6$,

$$\mathbb{E}\frac{1}{2}\|w_{t,\eta} - w_{\text{train}}\|_{H_{\text{train}}}^2 \mathbb{1}\{\mathcal{E}_1 \cap \bar{\mathcal{E}}_2(\eta)\} \geq \frac{1}{4}(40\sigma - 5\sigma)^2 \Pr[\mathcal{E}_1 \cap \mathcal{E}_3] \geq \frac{\epsilon^2 d\sigma^2}{n},$$

where the last inequality assumes $n \geq c, d \geq c_4$ for some constant $c, c_4$.

Overall, we know $\mathbb{E}\frac{1}{2}\|w_{t,\eta} - w_{\text{train}}\|_{H_{\text{train}}}^2 \mathbb{1}\{\mathcal{E}_1 \cap \bar{\mathcal{E}}_2(\eta)\}$ equals zero for all $\eta \in [0, 4/3]$ and is at least $\frac{\epsilon^2 d\sigma^2}{n}$ for all $\eta \in [6, \infty)$. By definition, we know $\hat{\eta} \in (4/3, 6)$. $\qquad\square$

**Proof of Lemma 31.** By Lemma 29, we know $\hat{\eta}$ is a constant. The proof is very similar as in Lemma 8. Let $\mathcal{E}_1$ and $\bar{\mathcal{E}}_2(\eta)$ be as defined in Definition 2. For the simplicity of the proof, we assume $\mathbb{E}\frac{1}{2}\|w_{t,\hat{\eta}} - w_{\text{train}}\|_{H_{\text{train}}}^2 \mathbb{1}\{\mathcal{E}_1 \cap \bar{\mathcal{E}}_2(\hat{\eta})\} \leq \frac{\epsilon^2 d\sigma^2}{n}$. The other case can be resolved using techniques in the proof of Lemma 8.

Recall the population meta objective

$$F_{TbT}(\eta) = \mathbb{E}\frac{1}{2} \|w_{t,\eta} - w_{\text{train}}\|^2_{H_{\text{train}}} + \frac{n-d}{2n}\sigma^2.$$

Therefore, we only need to construct an $\epsilon$-net for the first term.

We can divide $\mathbb{E}\frac{1}{2} \|w_{t,\eta} - w_{\text{train}}\|^2_{H_{\text{train}}}$ as follows,

$$\mathbb{E}\frac{1}{2} \|w_{t,\eta} - w_{\text{train}}\|^2_{H_{\text{train}}}$$
$$=\mathbb{E}\frac{1}{2} \|w_{t,\eta} - w_{\text{train}}\|^2_{H_{\text{train}}} \mathbb{1}\{\mathcal{E}_1 \cap \mathcal{E}_2(\hat{\eta})\} + \mathbb{E}\frac{1}{2} \|w_{t,\eta} - w_{\text{train}}\|^2_{H_{\text{train}}} \mathbb{1}\{\mathcal{E}_1 \cap \bar{\mathcal{E}}_2(\hat{\eta})\}$$
$$+ \mathbb{E}\frac{1}{2} \|w_{t,\eta} - w_{\text{train}}\|^2_{H_{\text{train}}} \mathbb{1}\{\bar{\mathcal{E}}_1\}.$$

We will construct an $\epsilon$-net for the first term and show the other two terms are small. Let's first consider the third term. Assuming $n \geq 40d$, we know $\Pr[\mathcal{E}_1] \leq \exp(-\Omega(n))$. Since $\frac{1}{2} \|w_{t,\eta} - w_{\text{train}}\|^2_{H_{\text{train}}}$ is $O(1)$-subexponential, by Cauchy-Schwarz inequality, we have $\mathbb{E}\frac{1}{2} \|w_{t,\eta} - w_{\text{train}}\|^2_{H_{\text{train}}} \mathbb{1}\{\bar{\mathcal{E}}_1\} = O(1)\exp(-\Omega(n))$. Choosing $n \geq c\log(n/(\epsilon d))$ for some constant $c$, we know $\frac{1}{2} \|w_{t,\hat{\eta}} - w_{\text{train}}\|^2_{H_{\text{train}}} \mathbb{1}\{\bar{\mathcal{E}}_1\} \leq \frac{\epsilon^2 d\sigma^2}{n}$.

Then we upper bound the second term. Since $\mathbb{E}\frac{1}{2} \|w_{t,\hat{\eta}} - w_{\text{train}}\|^2_{H_{\text{train}}} \mathbb{1}\{\mathcal{E}_1 \cap \bar{\mathcal{E}}_2(\hat{\eta})\} \leq \frac{\epsilon^2 d\sigma^2}{n}$ and $\frac{1}{2} \|w_{t,\hat{\eta}} - w_{\text{train}}\|^2_{H_{\text{train}}} \geq \frac{35^2\sigma^2}{4}$ when $w_{t,\hat{\eta}}$ diverges, we know $\Pr[\mathcal{E}_1 \cap \bar{\mathcal{E}}_2(\hat{\eta})] \leq \frac{4\epsilon^2 d}{35^2 n}$. Then, we can upper bound the second term as follows,

$$\mathbb{E}\frac{1}{2} \|w_{t,\eta} - w_{\text{train}}\|^2_{H_{\text{train}}} \mathbb{1}\{\mathcal{E}_1 \cap \bar{\mathcal{E}}_2(\hat{\eta})\} \leq \frac{3 \times 45^2\sigma^2}{4}\frac{4\epsilon^2 d}{35^2 n} \leq \frac{6\epsilon^2 d\sigma^2}{n}$$

Next, similar as in Lemma 8, we can show the first term $\frac{1}{2} \|w_{t,\eta} - w_{\text{train}}\|^2_{H_{\text{train}}} \mathbb{1}\{\mathcal{E}_1 \cap \mathcal{E}_2(\hat{\eta})\}$ is $O(t)$-lipschitz. Therefore, there exists an $\frac{\epsilon^2 d\sigma^2}{n}$-net $N$ for $\mathbb{E}\frac{1}{2} \|w_{t,\eta} - w_{\text{train}}\|^2_{H_{\text{train}}} \mathbb{1}\{\mathcal{E}_1 \cap \mathcal{E}_2(\hat{\eta})\}$ with size $O(\frac{tn}{\epsilon^2 d})$. That means, for any $\eta \in [0, \hat{\eta}]$,

$$\left|\mathbb{E}\frac{1}{2} \|w_{t,\eta} - w_{\text{train}}\|^2_{H_{\text{train}}} \mathbb{1}\{\mathcal{E}_1 \cap \mathcal{E}_2(\hat{\eta})\} - \mathbb{E}\frac{1}{2} \|w_{t,\eta'} - w_{\text{train}}\|^2_{H_{\text{train}}} \mathbb{1}\{\mathcal{E}_1 \cap \mathcal{E}_2(\hat{\eta})\}\right| \leq \frac{\epsilon^2 d\sigma^2}{n}$$

for $\eta' = \arg\min_{\eta'' \in N, \eta'' \leq \eta}(\eta - \eta'')$.

Combing with the upper bounds on the second term and the third term, we have for any $\eta \in [0, \hat{\eta}]$,

$$|F_{TbT}(\eta) - F_{TbT}(\eta')| \leq \frac{8\epsilon^2 d\sigma^2}{n}$$

for $\eta' = \arg\min_{\eta'' \in N, \eta'' \leq \eta}(\eta - \eta'')$. □

**Proof of Lemma 32.** By Lemma 29, we know $\hat{\eta}$ is a constant. For each $k \in [m]$, let $\mathcal{E}_{1,k}$ be the event that $\sqrt{n}/2 \leq \sigma_i(X_{\text{train}}^{(k)}) \leq 3\sqrt{n}/2$ and $1/2 \leq \lambda_i(H_{\text{train}}^{(k)}) \leq 3/2$ for all $i \in [d]$ and $\sqrt{n}\sigma/2 \leq \left\|\xi_{\text{train}}^{(k)}\right\| \leq 2\sqrt{n}\sigma$. Assuming $n \geq 40d$, by Lemma 27, we know with probability at least $1 - m\exp(-\Omega(n))$, $\mathcal{E}_{1,k}$'s hold for all $k \in [m]$.

Then, similar as in Lemma 9, there exists an $\frac{\epsilon^2 d\sigma^2}{n}$-net $N'$ with $|N'| = O(\frac{nt}{\epsilon^2 d} + m)$ for $\hat{F}_{TbT}$. That means, for any $\eta \in [0, \hat{\eta}]$,

$$\left|\hat{F}_{TbT}(\eta) - \hat{F}_{TbT}(\eta')\right| \leq \frac{\epsilon^2 d\sigma^2}{n}$$

for $\eta' = \arg\min_{\eta'' \in N', \eta'' \leq \eta}(\eta - \eta'')$. □

# D    PROOFS OF TRAIN-BY-TRAIN V.S. TRAIN-BY-VALIDATION (SGD)

Previously, we have shown that train-by-validation generalizes better than train-by-train when the tasks are trained by GD and when the number of samples is small. In this section, we show a similar phenomenon also appears in the SGD setting.

In the train-by-train setting, each task $P$ contains a training set $S_{\text{train}} = \{(x_i, y_i)\}_{i=1}^n$. The inner objective is defined as $\hat{f}(w) = \frac{1}{2n} \sum_{(x,y) \in S_{\text{train}}} (\langle w, x \rangle - y)^2$. Let $\{w_{\tau,\eta}\}$ be the SGD sequence running on $\hat{f}(w)$ from initialization $0$ (without truncation). That means, $w_{\tau,\eta} = w_{\tau-1,\eta} - \eta \hat{\nabla} \hat{f}(w_{\tau-1,\eta})$, where $\hat{\nabla} \hat{f}(w_{\tau-1,\eta}) = (\langle w_{\tau-1,\eta}, x_{i(\tau-1)} \rangle - y_{i(\tau-1)}) x_{i(\tau-1)}$. Here index $i(\tau-1)$ is independently and uniformly sampled from $[n]$. We denote the SGD noise as $n_{\tau-1,\eta} := \hat{\nabla} \hat{f}(w_{\tau-1,\eta}) - \nabla \hat{f}(w_{\tau-1,\eta})$. The meta-loss on task $P$ is defined as follows,

$$\Delta_{TbT(n)}(\eta, P) = \mathbb{E}_{\text{SGD}} \hat{f}(w_{t,\eta}) = \mathbb{E}_{\text{SGD}} \frac{1}{2n} \sum_{(x,y) \in S_{\text{train}}} (\langle w_{t,\eta}, x \rangle - y)^2,$$

where the expectation is taken over the SGD noise. Note $w_{t,\eta}$ depends on the SGD noise along the trajectory. Then, the empirical meta objective $\hat{F}_{TbT(n)}(\eta)$ is the average of the meta-loss across $m$ different specific tasks

$$\hat{F}_{TbT(n)}(\eta) = \frac{1}{m} \sum_{k=1}^m \Delta_{TbT(n)}(\eta, P_k). \tag{4}$$

In order to control the SGD noise in expectation, we restrict the feasible set of step sizes into $O(1/d)$. We show within this range, the optimal step size under $\hat{F}_{TbT(n)}$ is $\Omega(1/d)$ and the learned weight is far from ground truth $w^*$ on new tasks. We prove Theorem 9 in Section D.1.

**Theorem 9.** *Let the meta objective $\hat{F}_{TbT(n)}$ be as defined in Equation 4 with $n \in [d/4, 3d/4]$. Suppose $\sigma$ is a constant. Assume unroll length $t \geq c_2 d$ and dimension $d \geq c_4 \log(m)$ for certain constants $c_2, c_4$. Then, with probability at least $0.99$ in the sampling of training tasks $P_1, \cdots, P_m$ and test task $P$,*

$$\eta_{train}^* = \Omega(1/d) \text{ and } \mathbb{E}_{SGD} \left\| w_{t,\eta_{train}^*} - w^* \right\|^2 = \Omega(\sigma^2),$$

*for all $\eta_{train}^* \in \arg\min_{0 \leq \eta \leq \frac{1}{2L^3 d}} \hat{F}_{TbT(n)}(\eta)$, where $L = 100$ and $w_{t,\eta_{train}^*}$ is trained by running SGD on test task $P$.*

In the train-by-validation setting, each task $P$ contains a training set $S_{\text{train}}$ with $n_1$ samples and a validation set with $n_2$ samples. The inner objective is defined as $\hat{f}(w) = \frac{1}{2n_1} \sum_{(x,y) \in S_{\text{train}}} (\langle w, x \rangle - y)^2$. Let $\{w_{\tau,\eta}\}$ be the SGD sequence running on $\hat{f}(w)$ from initialization $0$ (with the same truncation defined in Section 4). For each task $P$, the meta-loss $\Delta_{TbV(n_1, n_2)}(\eta, P)$ is defined as

$$\Delta_{TbV(n_1, n_2)}(\eta, P) = \mathbb{E}_{\text{SGD}} \frac{1}{2n_2} \sum_{(x,y) \in S_{\text{valid}}} (\langle w_{t,\eta}, x \rangle - y)^2.$$

The empirical meta objective $\hat{F}_{TbV(n_1, n_2)}(\eta)$ is the average of the meta-loss across $m$ different tasks $P_1, P_2, ..., P_m$,

$$\hat{F}_{TbV(n_1, n_2)}(\eta) = \frac{1}{m} \sum_{k=1}^m \Delta_{TbV(n_1, n_2)}(\eta, P_k). \tag{5}$$

In order to bound the SGD noise with high probability, we restrict the feasible set of the step sizes into $O(\frac{1}{d^2 \log^2 d})$. Within this range, we prove the optimal step size under $\hat{F}_{TbV(n_1, n_2)}$ is $\Theta(1/t)$ and the learned weight is better than initialization $0$ by a constant on new tasks. Theorem 10 is proved in Section D.2.

**Theorem 10.** *Let the meta objective $\hat{F}_{TbV(n_1, n_2)}$ be as defined in Equation 5 with $n_1, n_2 \in [d/4, 3d/4]$. Assume noise level $\sigma$ is a large constant $c_1$. Assume unroll length $t \geq c_2 d^2 \log^2(d)$,*

*number of training tasks $m \geq c_3$ and dimension $d \geq c_4$ for certain constants $c_2, c_3, c_4$. There exists constant $c_5$ such that with probability at least $0.99$ in the sampling of training tasks, we have*

$$\eta_{valid}^* = \Theta(1/t) \text{ and } \mathbb{E} \left\| w_{t,\eta_{valid}^*} - w^* \right\|^2 = \|w^*\|^2 - \Omega(1)$$

*for all $\eta_{valid}^* \in \arg\min_{0 \leq \eta \leq \frac{1}{c_5 d^2 \log^2(d)}} \hat{F}_{TbV(n_1,n_2)}(\eta)$, where the expectation is taken over the new tasks and SGD noise.*

**Notations:** In the following proofs, we use the same set of notations defined in Appendix B. We use $\mathbb{E}_{P \sim \mathcal{T}}$ to denote the expectation over the sampling of tasks and use $\mathbb{E}_{\text{SGD}}$ to denote the expectation over the SGD noise. We use $\mathbb{E}$ to denote $\mathbb{E}_{P \sim \mathcal{T}} \mathbb{E}_{\text{SGD}}$. Same as in Appendix B, we use letter $L$ to denote constant $100$, which upper bounds $\|H_{\text{train}}\|$ with high probability.

### D.1    TRAIN-BY-TRAIN (SGD)

Recall Theorem 9 as follows.

**Theorem 9.** *Let the meta objective $\hat{F}_{TbT(n)}$ be as defined in Equation 4 with $n \in [d/4, 3d/4]$. Suppose $\sigma$ is a constant. Assume unroll length $t \geq c_2 d$ and dimension $d \geq c_4 \log(m)$ for certain constants $c_2, c_4$. Then, with probability at least $0.99$ in the sampling of training tasks $P_1, \cdots, P_m$ and test task $P$,*

$$\eta_{train}^* = \Omega(1/d) \text{ and } \mathbb{E}_{SGD} \left\| w_{t,\eta_{train}^*} - w^* \right\|^2 = \Omega(\sigma^2),$$

*for all $\eta_{train}^* \in \arg\min_{0 \leq \eta \leq \frac{1}{2L^3 d}} \hat{F}_{TbT(n)}(\eta)$, where $L = 100$ and $w_{t,\eta_{train}^*}$ is trained by running SGD on test task $P$.*

In order to prove Theorem 9, we first show that $\eta_{\text{train}}^*$ is $\Omega(1/d)$ in Lemma 33. The proof is similar as in the GD setting. As long as $\eta = O(1/d)$, the SGD noise is dominated by the full gradient. Then, we can show that $\Delta_{TbT}(\eta, P)$ is roughly $(1 - \Theta(1)\eta)^t$, which implies that $\eta_{\text{train}}^* = \Omega(1/d)$. We leave the proof of Lemma 33 into Section D.1.1.

**Lemma 33.** *Assume $t \geq c_2 d$ with certain constant $c_2$. With probability at least $1 - m \exp(-\Omega(d))$ in the sampling of $m$ training tasks,*

$$\eta_{train}^* \geq \frac{1}{6L^5 d},$$

*for all $\eta_{train}^* \in \arg\min_{0 \leq \eta \leq \frac{1}{2L^3 d}} \hat{F}_{TbT}(\eta)$.*

Let $P = (\mathcal{D}(w^*), S_{\text{train}}, \ell)$ be an independently sampled test task with $|S_{\text{train}}| = n \in [d/4, 3d/4]$. For any step size $\eta \in [\frac{1}{6L^5 d}, \frac{1}{2L^3 d}]$, let $w_{t,\eta}$ be the weight obtained by running SGD on $\hat{f}(w)$ for $t$ steps. Next, we show $\mathbb{E}_{\text{SGD}} \|w_{t,\eta} - w^*\|^2 = \Omega(\sigma^2)$ with high probability in the sampling of $P$.

**Lemma 34.** *Suppose $\sigma$ is a constant. Assume unroll length $t \geq c_2 d$ for some constant $c_2$. With probability at least $1 - \exp(-\Omega(d))$ in the sampling of test task $P$,*

$$\mathbb{E}_{SGD} \|w_{t,\eta} - w^*\|^2 \geq \frac{\sigma^2}{128L},$$

*for all $\eta \in [\frac{1}{6L^5 d}, \frac{1}{2L^3 d}]$, where $w_{t,\eta}$ is obtained by running SGD on task $P$ for $t$ iterations.*

With Lemma Lemma 33 and Lemma 34, the proof of Theorem 9 is straightforward.

**Proof of Theorem 9.** Combing Lemma 33 and Lemma 34, we know as long as $\sigma$ is a constant, $t \geq c_2 d, d \geq c_4 \log(m)$, with probability at least $0.99$, $\eta_{\text{train}}^* = \Omega(1/d)$ and $\mathbb{E}_{\text{SGD}} \left\| w_{t,\eta_{\text{train}}^*} - w^* \right\|^2 = \Omega(\sigma^2)$, for all $\eta_{\text{train}}^* \in \arg\min_{0 \leq \eta \leq \frac{1}{2L^3 d}} \hat{F}_{TbT}(\eta)$. $\qquad\square$

### D.1.1    DETAILED PROOFS

**Proof of Lemma 33.** The proof is very similar to the proof of Lemma 2 except that we need to bound the SGD noise term. For each $k \in [m]$, let $\mathcal{E}_k$ be the event that $\sqrt{d}/\sqrt{L} \leq \sigma_i(X_{\text{train}}) \leq$

$\sqrt{Ld}$ and $1/L \leq \lambda_i(H_{\text{train}}) \leq L$ for all $i \in [n]$ and $\sqrt{d}\sigma/4 \leq \|\xi_{\text{train}}\| \leq \sqrt{d}\sigma$. According to Lemma 1 and Lemma 45, we know for each $k \in [m]$, $\mathcal{E}_k$ happens with probability at least $1 - \exp(-\Omega(d))$. Taking a union bound over all $k \in [m]$, we know $\cap_{k\in[m]}\mathcal{E}_k$ holds with probability at least $1 - m\exp(-\Omega(d))$. From now on, we assume $\cap_{k\in[m]}\mathcal{E}_k$ holds.

For each $k \in [m]$, we have

$$\Delta_{TbT}(\eta, P_k) := \frac{1}{2}\mathbb{E}_{\text{SGD}}\left\|w_{t,\eta}^{(k)} - w_{\text{train}}^{(k)}\right\|_{H_{\text{train}}^{(k)}}^2.$$

Since $1/L \leq \lambda_i(H_{\text{train}}^{(k)}) \leq L$ and $(w_{t,\eta}^{(k)} - w_{\text{train}}^{(k)})$ is in the span of $H_{\text{train}}^{(k)}$, we have

$$\frac{1}{2L}\mathbb{E}_{\text{SGD}}\left\|w_{t,\eta}^{(k)} - w_{\text{train}}^{(k)}\right\|^2 \leq \Delta_{TbT}(\eta, P_k) \leq \frac{L}{2}\mathbb{E}_{\text{SGD}}\left\|w_{t,\eta}^{(k)} - w_{\text{train}}^{(k)}\right\|^2.$$

Recall the updates of stochastic gradient descent,

$$w_{t,\eta}^{(k)} - w_{\text{train}}^{(k)} = (I - \eta H_{\text{train}}^{(k)})(w_{t-1,\eta}^{(k)} - w_{\text{train}}^{(k)}) - \eta n_{t-1,\eta}^{(k)}.$$

Therefore,

$$\mathbb{E}_{\text{SGD}}\left[\left\|w_{t,\eta}^{(k)} - w_{\text{train}}^{(k)}\right\|^2 |w_{t-1,\eta}^{(k)}\right] = \left\|(I - \eta H_{\text{train}}^{(k)})(w_{t-1,\eta}^{(k)} - w_{\text{train}}^{(k)})\right\|^2 + \eta^2 \mathbb{E}_{\text{SGD}}\left[\left\|n_{t-1,\eta}^{(k)}\right\|^2 |w_{t-1,\eta}^{(k)}\right].$$

We know for any $\eta \leq 1/L$,

$$(1 - 2\eta L)\left\|w_{t-1,\eta}^{(k)} - w_{\text{train}}^{(k)}\right\|^2 \leq \left\|(I - \eta H_{\text{train}}^{(k)})(w_{t-1,\eta}^{(k)} - w_{\text{train}}^{(k)})\right\|^2 \leq (1 - \frac{\eta}{L})\left\|w_{t-1,\eta}^{(k)} - w_{\text{train}}^{(k)}\right\|^2.$$

The noise can be bounded as follows,

$$\eta^2 \mathbb{E}_{\text{SGD}}\left[\left\|n_{t-1,\eta}^{(k)}\right\|^2 |w_{t-1,\eta}^{(k)}\right]$$

$$= \eta^2 \mathbb{E}_{\text{SGD}}\left[\left\|x_{i(t-1)}x_{i(t-1)}^{\top}(w_{t-1,\eta}^{(k)} - w_{\text{train}}^{(k)}) - H_{\text{train}}^{(k)}(w_{t-1,\eta}^{(k)} - w_{\text{train}}^{(k)})\right\|^2 |w_{t-1,\eta}^{(k)}\right]$$

$$\leq \eta^2 \mathbb{E}_{\text{SGD}}\left[\left\|x_{i(t-1)}x_{i(t-1)}^{\top}(w_{t-1,\eta}^{(k)} - w_{\text{train}}^{(k)})\right\|^2 |w_{t-1,\eta}^{(k)}\right]$$

$$\leq \eta^2 \max_{i(t-1)}\left\|x_{i(t-1)}\right\|^2 \left\|w_{t-1,\eta}^{(k)} - w_{\text{train}}^{(k)}\right\|_{H_{\text{train}}^{(k)}}^2.$$

Since $\|X_{\text{train}}\| \leq \sqrt{L}\sqrt{d}$, we immediately know $\max_{i(t-1)}\left\|x_{i(t-1)}\right\| \leq \sqrt{L}\sqrt{d}$. Therefore, we can bound the noise as follows,

$$\eta^2 \mathbb{E}_{\text{SGD}}\left[\left\|n_{t-1,\eta}^{(k)}\right\|^2 |w_{t-1,\eta}^{(k)}\right] \leq \eta^2 \max_{i(t-1)}\left\|x_{i(t-1)}\right\|^2 \left\|w_{t-1,\eta}^{(k)} - w_{\text{train}}^{(k)}\right\|_{H_{\text{train}}^{(k)}}^2$$

$$\leq L^2\eta^2 d \left\|w_{t-1,\eta}^{(k)} - w_{\text{train}}^{(k)}\right\|^2.$$

As long as $\eta \leq \frac{1}{2L^3 d}$, we have

$$(1 - \eta L)\left\|w_{t-1,\eta}^{(k)} - w_{\text{train}}^{(k)}\right\|^2 \leq \mathbb{E}_{\text{SGD}}\left[\left\|w_{t,\eta}^{(k)} - w_{\text{train}}^{(k)}\right\|^2 |w_{t-1,\eta}^{(k)}\right] \leq (1 - \frac{\eta}{2L})\left\|w_{t-1,\eta}^{(k)} - w_{\text{train}}^{(k)}\right\|^2.$$

This further implies

$$(1 - \eta L)^t \|w_{\text{train}}\|^2 \leq \mathbb{E}_{\text{SGD}}\left\|w_{t,\eta}^{(k)} - w_{\text{train}}^{(k)}\right\|^2 \leq (1 - \frac{\eta}{2L})^t \|w_{\text{train}}\|^2.$$

Let $\eta_2 := \frac{1}{2L^3 d}$, we have

$$\Delta_{TbT}(\eta, P_k) \leq \frac{L}{2}(1 - \frac{1}{4L^4 d})^t \|w_{\text{train}}\|^2$$

Let $\eta_1 := \frac{1}{6L^5 d}$, for all $\eta \in [0, \eta_1]$ we have

$$\Delta_{TbT}(\eta, P_k) \geq \frac{1}{2L}(1 - \frac{1}{6L^4 d})^t \|w_{\text{train}}\|^2 .$$

As long as $t \geq c_2 d$ for certain constant $c_2$, we know

$$\frac{1}{2L}(1 - \frac{1}{6L^4 d})^t \|w_{\text{train}}\|^2 > \frac{L}{2}(1 - \frac{1}{4L^4 d})^t \|w_{\text{train}}\|^2 .$$

As this holds for all $k \in [m]$ and $\hat{F}_{TbT} = 1/m \sum_{i=1}^{m} \Delta_{TbT}(\eta, P_k)$, we know the optimal step size $\eta_{\text{train}}^*$ is within $[\frac{1}{6L^5 d}, \frac{1}{2L^3 d}]$. $\square$

We rely the following technical lemma to prove Lemma 34.

**Lemma 35.** *Suppose $\sigma$ is a constant. Given any $\epsilon > 0$, with probability at least $1 - O(1/\epsilon) \exp(-\Omega(\epsilon^2 d))$,*

$$\left| \left\langle B_{t,\eta} w_{train}^* - w^*, B_{t,\eta}(X_{train})^\dagger \xi_{train} \right\rangle \right| \leq \epsilon,$$

*for all $\eta \in [0, \frac{1}{2L^3 d}]$.*

**Proof of Lemma 35.** By Lemma 1, with probability at least $1 - \exp(-\Omega(d))$, $\sqrt{d}/\sqrt{L} \leq \sigma_i(X_{\text{train}}) \leq \sqrt{Ld}$ and $1/L \leq \lambda_i(H_{\text{train}}) \leq L$ for all $i \in [n]$. Therefore $\left\| [(X_{\text{train}})^\dagger]^\top B_{t,\eta}(B_{t,\eta} w_{\text{train}}^* - w^*) \right\| \leq 2\sqrt{L}/\sqrt{d}$. Notice that $\xi_{\text{train}}$ is independent with $[(X_{\text{train}})^\dagger]^\top B_{t,\eta}(B_{t,\eta} w_{\text{train}}^* - w^*)$. By Hoeffding's inequality, with probability at least $1 - \exp(-\Omega(\epsilon^2 d))$,

$$\left| \left\langle [(X_{\text{train}})^\dagger]^\top B_{t,\eta}(B_{t,\eta} w_{\text{train}}^* - w^*), \xi_{\text{train}} \right\rangle \right| \leq \epsilon.$$

Next, we construct an $\epsilon$-net for $\eta$ and show the crossing term is small for all $\eta \in [0, \frac{1}{2L^3 d}]$. For simplicity, denote $g(\eta) := \left\langle B_{t,\eta} w_{\text{train}}^* - w^*, B_{t,\eta}(X_{\text{train}})^\dagger \xi_{\text{train}} \right\rangle$. Taking the derivative of $g(\eta)$, we have

$$\begin{aligned} g'(\eta) =& t \left\langle H_{\text{train}}(I - \eta H_{\text{train}})^{t-1} w_{\text{train}}^*, B_{t,\eta}(X_{\text{train}})^\dagger \xi_{\text{train}} \right\rangle \\ &+ t \left\langle B_{t,\eta} w_{\text{train}}^* - w^*, H_{\text{train}}(I - \eta H_{\text{train}})^{t-1}(X_{\text{train}})^\dagger \xi_{\text{train}} \right\rangle \end{aligned}$$

According to Lemma 45, we know with probability at least $1 - \exp(-\Omega(d))$, $\|\xi_{\text{train}}\| \leq \sqrt{d}\sigma$. Therefore, the derivative $g'(\eta)$ can be bounded as follows,

$$|g'(\eta)| = O(1)t(1 - \frac{\eta}{L})^{t-1}$$

Similar as in Lemma 14, there exists an $\epsilon$-net $N_\epsilon$ with size $O(1/\epsilon)$ such that for any $\eta \in [0, \frac{1}{3L^3 d}]$, there exists $\eta' \in N_\epsilon$ with $|g(\eta) - g(\eta')| \leq \epsilon$. Taking a union bound over $N_\epsilon$, we have with probability at least $1 - O(1/\epsilon) \exp(-\Omega(\epsilon^2 d))$, for every $\eta \in N_\epsilon$,

$$\left| \left\langle B_{t,\eta} w_{\text{train}}^* - w^*, B_{t,\eta}(X_{\text{train}})^\dagger \xi_{\text{train}} \right\rangle \right| \leq \epsilon.$$

which implies for every $\eta \in [0, \frac{1}{3L^3 d}]$.

$$\left| \left\langle B_{t,\eta} w_{\text{train}}^* - w^*, B_{t,\eta}(X_{\text{train}})^\dagger \xi_{\text{train}} \right\rangle \right| \leq 2\epsilon.$$

Changing $\epsilon$ to $\epsilon'/2$ finishes the proof. $\square$

**Proof of Lemma 34.** According to Lemma 1 and Lemma 45, we know with probability at least $1 - \exp(-\Omega(d))$, $\sqrt{d}/\sqrt{L} \leq \sigma_i(X_{\text{train}}) \leq \sqrt{Ld}$ and $1/L \leq \lambda_i(H_{\text{train}}) \leq L$ for all $i \in [n]$ and $\sqrt{d}\sigma/4 \leq \|\xi_{\text{train}}\| \leq \sqrt{d}\sigma$. We assume these properties hold in the proof and take a union bound at the end.

Recall that $\mathbb{E}_{\text{SGD}} \|w_{t,\eta} - w^*\|^2$ can be lower bounded as follows,

$$\begin{aligned} \mathbb{E}_{\text{SGD}} \|w_{t,\eta} - w^*\|^2 =& \mathbb{E}_{\text{SGD}} \left\| B_{t,\eta}(w_{\text{train}}^* + (X_{\text{train}})^\dagger \xi_{\text{train}}) - \eta \sum_{\tau=0}^{t-1}(I - \eta H_{\text{train}})^{t-1-\tau} n_{\tau,\eta} - w^* \right\|^2 \\ \geq& \left\| B_{t,\eta}(w_{\text{train}}^* + (X_{\text{train}})^\dagger \xi_{\text{train}}) - w^* \right\|^2 \\ \geq& \left\| B_{t,\eta}(X_{\text{train}})^\dagger \xi_{\text{train}} \right\|^2 + 2 \left\langle B_{t,\eta} w_{\text{train}}^* - w^*, B_{t,\eta}(X_{\text{train}})^\dagger \xi_{\text{train}} \right\rangle \end{aligned}$$

For any $\eta \in [\frac{1}{6L^5d}, \frac{1}{2L^3d}]$, we can lower bound the first term as follows,

$$
\begin{aligned}
\left\| B_{t,\eta}(X_{\text{train}})^\dagger \xi_{\text{train}} \right\|^2 &\geq \left( 1 - \exp\left( -\frac{\eta t}{L} \right) \right)^2 \frac{\sigma^2}{16L} \\
&\geq \left( 1 - \exp\left( -\frac{t}{6L^6 d} \right) \right)^2 \frac{\sigma^2}{16L} \\
&\geq \frac{\sigma^2}{64L},
\end{aligned}
$$

where the last inequality holds as long as $t \geq c_2 d$ for certain constant $c_2$.

Choosing $\epsilon = \frac{\sigma^2}{256L}$ in Lemma 35, we know with probability at least $1 - \exp(-\Omega(d))$,

$$
\left| \langle B_{t,\eta} w^*_{\text{train}} - w^*, B_{t,\eta}(X_{\text{train}})^\dagger \xi_{\text{train}} \rangle \right| \leq \frac{\sigma^2}{256L},
$$

for all $\eta \in [0, \frac{1}{2L^3 d}]$.

Overall, we have $\mathbb{E}_{\text{SGD}} \| w_{t,\eta} - w^* \|^2 \geq \frac{\sigma^2}{128L}$. Taking a union bound over all the bad events, we know this happens with probability at least $1 - \exp(-\Omega(d))$. $\qquad \square$

### D.2 TRAIN-BY-VALIDATION (SGD)

Recall Theorem 10 as follows.

**Theorem 10.** *Let the meta objective $\hat{F}_{TbV(n_1,n_2)}$ be as defined in Equation 5 with $n_1, n_2 \in [d/4, 3d/4]$. Assume noise level $\sigma$ is a large constant $c_1$. Assume unroll length $t \geq c_2 d^2 \log^2(d)$, number of training tasks $m \geq c_3$ and dimension $d \geq c_4$ for certain constants $c_2, c_3, c_4$. There exists constant $c_5$ such that with probability at least $0.99$ in the sampling of training tasks, we have*

$$
\eta^*_{valid} = \Theta(1/t) \text{ and } \mathbb{E} \left\| w_{t,\eta^*_{valid}} - w^* \right\|^2 = \| w^* \|^2 - \Omega(1)
$$

*for all $\eta^*_{valid} \in \arg\min_{0 \leq \eta \leq \frac{1}{c_5 d^2 \log^2(d)}} \hat{F}_{TbV(n_1,n_2)}(\eta)$, where the expectation is taken over the new tasks and SGD noise.*

To prove Theorem 10, we first study the behavior of the population meta objective $F_{TbV}$. That is,

$$
\begin{aligned}
F_{TbV}(\eta) := \mathbb{E}_{P \sim \mathcal{T}} \Delta_{TbV}(\eta, P) &= \mathbb{E}_{P \sim \mathcal{T}} \mathbb{E}_{\text{SGD}} \frac{1}{2} \left\| w_{t,\eta} - w^* - (X_{\text{valid}})^\dagger \xi_{\text{valid}} \right\|^2_{H_{\text{valid}}} \\
&= \mathbb{E}_{P \sim \mathcal{T}} \mathbb{E}_{\text{SGD}} \frac{1}{2} \| w_{t,\eta} - w^* \|^2 + \frac{\sigma^2}{2}.
\end{aligned}
$$

We show that the optimal step size for the population meta objective $F_{TbV}$ is $\Theta(1/t)$ and $\mathbb{E}_{P \sim \mathcal{T}} \mathbb{E}_{\text{SGD}} \| w_{t,\eta} - w^* \|^2 = \| w^* \|^2 - \Omega(1)$ under the optimal step size.

**Lemma 36.** *Suppose $\sigma$ is a large constant $c_1$. Assume $t \geq c_2 d^2 \log^2(d), d \geq c_4$ for some constants $c_2, c_4$. There exist $\eta_1, \eta_2, \eta_3 = \Theta(1/t)$ with $\eta_1 < \eta_2 < \eta_3$ and constant $c_5$ such that*

$$
\begin{aligned}
F_{TbV}(\eta_2) &\leq \frac{1}{2} \| w^* \|^2 - \frac{9}{10} C + \frac{\sigma^2}{2} \\
F_{TbV}(\eta) &\geq \frac{1}{2} \| w^* \|^2 - \frac{6}{10} C + \frac{\sigma^2}{2}, \forall \eta \in [0, \eta_1] \cup [\eta_3, \frac{1}{c_5 d^2 \log^2(d)}]
\end{aligned}
$$

*where $C$ is a positive constant.*

In order to relate the behavior of $F_{TbV}$ to $\hat{F}_{TbV}$, we show a generalization result from $\hat{F}_{TbV}$ to $F_{TbV}$ for $\eta \in [0, \frac{1}{c_5 d^2 \log^2(d/\epsilon)}]$.

**Lemma 37.** *For any $1 > \epsilon > 0$, assume $\sigma$ is a constant and $d \geq c_4 \log(1/\epsilon)$ for some constant $c_4$. There exists constant $c_5$ such that with probability at least $1 - O(1/\epsilon) \exp(-\Omega(\epsilon^2 m))$,*

$$
|\hat{F}_{TbV}(\eta) - F_{TbV}(\eta)| \leq \epsilon,
$$

*for all $\eta \in [0, \frac{1}{c_5 d^2 \log^2(d/\epsilon)}]$.*

Combining Lemma 36 and Lemma 37, we give the proof of Theorem 10.

**Proof of Theorem 10.** The proof is almost the same as in the GD setting (Theorem 8). We omit the details here. □

### D.2.1 Behavior of $F_{TbV}$ for $\eta \in [0, \frac{1}{c_5 d^2 \log^2 d}]$

In this section, we give the proof of Lemma 36. Recall the lemma as follows,

**Lemma 36.** *Suppose $\sigma$ is a large constant $c_1$. Assume $t \geq c_2 d^2 \log^2(d), d \geq c_4$ for some constants $c_2, c_4$. There exist $\eta_1, \eta_2, \eta_3 = \Theta(1/t)$ with $\eta_1 < \eta_2 < \eta_3$ and constant $c_5$ such that*

$$F_{TbV}(\eta_2) \leq \frac{1}{2} \|w^*\|^2 - \frac{9}{10} C + \frac{\sigma^2}{2}$$

$$F_{TbV}(\eta) \geq \frac{1}{2} \|w^*\|^2 - \frac{6}{10} C + \frac{\sigma^2}{2}, \forall \eta \in [0, \eta_1] \cup [\eta_3, \frac{1}{c_5 d^2 \log^2(d)}]$$

*where $C$ is a positive constant.*

Recall that $F_{TbV}(\eta) = \mathbb{E}_{P \sim \mathcal{T}} \mathbb{E}_{SGD} 1/2 \|w_{t,\eta} - w^*\|^2 + \sigma^2/2$. Denote $Q(\eta) := \mathbb{E}_{SGD} 1/2 \|w_{t,\eta} - w^*\|^2$. Recall that we truncate the SGD sequence once the weight norm exceeds $4\sqrt{L}\sigma$. Due to the truncation, the expectation of $1/2 \|w_{t,\eta} - w^*\|^2$ over SGD noise is very tricky to analyze.

Instead, we define an auxiliary sequence $\{w'_{\tau,\eta}\}$ that is obtained by running SGD on task $P$ without truncation and we first study $Q'(\eta) := 1/2 \mathbb{E}_{SGD} \|w'_{t,\eta} - w^*\|^2$. In Lemma 38, we show that with high probability in the sampling of task $P$, the minimizer of $Q'(\eta)$ is $\Theta(1/t)$. The proof is very similar as the proof of Lemma 13 except that we need to bound the SGD noise at step size $\eta_2$. We defer the proof into Section D.2.3.

**Lemma 38.** *Given a task $P$, let $\{w'_{\tau,\eta}\}$ be the weight obtained by running SGD on task $P$ without truncation. Choose $\sigma$ as a large constant $c_1$. Assume unroll length $t \geq c_2 d$ for some constant $c_2$. With probability at least $1 - \exp(-\Omega(d))$ over the sampling of task $P$, $\sqrt{d}/\sqrt{L} \leq \sigma_i(X_{train}) \leq \sqrt{Ld}$ and $1/L \leq \lambda_i(H_{train}) \leq L$ for all $i \in [n]$ and $\sqrt{d}\sigma/4 \leq \|\xi_{train}\| \leq \sqrt{d}\sigma$ and there exists $\eta_1, \eta_2, \eta_3 = \Theta(1/t)$ with $\eta_1 < \eta_2 < \eta_3$ such that*

$$Q'(\eta_2) := 1/2 \mathbb{E}_{SGD} \|w'_{t,\eta_2} - w^*\|^2 \leq \frac{1}{2} \|w^*\|^2 - C$$

$$Q'(\eta) := 1/2 \mathbb{E}_{SGD} \|w'_{t,\eta} - w^*\|^2 \geq \frac{1}{2} \|w^*\|^2 - \frac{C}{2}, \forall \eta \in [0, \eta_1] \cup [\eta_3, 1/L]$$

*where $C$ is a positive constant.*

To relate the behavior of $Q'(\eta)$ defined on $\{w'_{\tau,\eta}\}$ to the behavior of $Q(\eta)$ defined on $\{w_{\tau,\eta}\}$. We show when the step size is small enough, the SGD sequence gets truncated with very small probability so that sequence $\{w_{\tau,\eta}\}$ almost always coincides with sequence $\{w'_{\tau,\eta}\}$. The proof of Lemma 39 is deferred into Section D.2.3.

**Lemma 39.** *Given a task $P$, assume $\sqrt{d}/\sqrt{L} \leq \sigma_i(X_{train}) \leq \sqrt{Ld}$ and $1/L \leq \lambda_i(H_{train}) \leq L$ for all $i \in [n]$ and $\sqrt{d}\sigma/4 \leq \|\xi_{train}\| \leq \sqrt{d}\sigma$. Given any $\epsilon > 0$, suppose $\eta \leq \frac{1}{c_5 d^2 \log^2(d/\epsilon)}$ for some constant $c_5$, we have*

$$|Q(\eta) - Q'(\eta)| \leq \epsilon.$$

Combining Lemma 38 and Lemma 39, we give the proof of lemma 36.

**Proof of Lemma 36.** Recall that we define $Q(\eta) := 1/2 \mathbb{E}_{SGD} \|w_{t,\eta} - w^*\|^2$ and $Q'(\eta) = 1/2 \mathbb{E}_{SGD} \|w'_{t,\eta} - w^*\|^2$. Here, $\{w'_{\tau,\eta}\}$ is a SGD sequence running on task $P$ without truncation.

According to Lemma 38, with probability at least $1 - \exp(-\Omega(d))$ over the sampling of task $P$, $\sqrt{d}/\sqrt{L} \leq \sigma_i(X_{\text{train}}) \leq \sqrt{Ld}$ and $1/L \leq \lambda_i(H_{\text{train}}) \leq L$ for all $i \in [n]$ and $\sqrt{d}\sigma/4 \leq \|\xi_{\text{train}}\| \leq$

$\sqrt{d}\sigma$ and there exists $\eta_1, \eta_2, \eta_3 = \Theta(1/t)$ with $\eta_1 < \eta_2 < \eta_3$ such that

$$Q'(\eta_2) \leq \frac{1}{2}\|w^*\|^2 - C$$

$$Q'(\eta) \geq \frac{1}{2}\|w^*\|^2 - \frac{C}{2}, \forall \eta \in [0, \eta_1] \cup [\eta_3, 1/L]$$

where $C$ is a positive constant. Call this event $\mathcal{E}$. Suppose the probability that $\mathcal{E}$ happens is $1 - \delta$. We can write $\mathbb{E}_{P \sim \mathcal{T}} Q(\eta)$ as follows,

$$\mathbb{E}_{P \sim \mathcal{T}} Q(\eta) = \mathbb{E}_{P \sim \mathcal{T}}[Q(\eta)|\mathcal{E}] \Pr[\mathcal{E}] + \mathbb{E}_{P \sim \mathcal{T}}[Q(\eta)|\bar{\mathcal{E}}] \Pr[\bar{\mathcal{E}}].$$

According to the algorithm, we know $\|w_{t,\eta}\|$ is always bounded by $4\sqrt{L}\sigma$. Therefore, $Q(\eta) := 1/2 \|w_{t,\eta} - w^*\|^2 \leq 13L\sigma^2$. By Lemma 39, we know conditioning on $\mathcal{E}$, $|Q(\eta) - Q'(\eta)| \leq \epsilon$ for any $\eta \leq \frac{1}{c_5 d^2 \log^2(d/\epsilon)}$. As long as $t \geq c_2 d^2 \log^2(d/\epsilon)$ for certain constant $c_2$, we know $\eta_3 \leq \frac{1}{c_5 d^2 \log^2(d/\epsilon)}$.

When $\eta = \eta_2$, we have

$$\mathbb{E}_{P \sim \mathcal{T}} Q(\eta_2) \leq (Q'(\eta_2) + \epsilon)(1 - \delta) + 13L\sigma^2\delta$$

$$\leq \left(\frac{1}{2}\|w^*\|^2 - C + \epsilon\right)(1 - \delta) + 13L\sigma^2\delta$$

$$\leq \frac{1}{2}\|w^*\|^2 - C + 13L\sigma^2\delta + \epsilon \leq \frac{1}{2}\|w^*\|^2 - \frac{9C}{10},$$

where the last inequality assumes $\delta \leq \frac{C}{260L\sigma^2}$ and $\epsilon \leq \frac{C}{20}$.

When $\eta \in [0, \eta_1] \cup [\eta_3, \frac{1}{c_5 d^2 \log^2(d/\epsilon)}]$, we have

$$\mathbb{E}_{P \sim \mathcal{T}} Q(\eta_2) \geq (Q'(\eta) - \epsilon)(1 - \delta) - 13L\sigma^2\delta$$

$$\geq \left(\frac{1}{2}\|w^*\|^2 - \frac{C}{2} - \epsilon\right)(1 - \delta) - 13L\sigma^2\delta$$

$$\geq \frac{1}{2}\|w^*\|^2 - \frac{C}{2} - \frac{\delta}{2} - 13L\sigma^2\delta - \epsilon \geq \frac{1}{2}\|w^*\|^2 - \frac{6C}{10},$$

where the last inequality holds as long as $\delta \leq \frac{C}{280L\sigma^2}$ and $\epsilon \leq \frac{C}{20}$.

According to Lemma 38, we know $\delta \leq \exp(-\Omega(d))$. Therefore, the conditions for $\delta$ can be satisfied as long as $d$ is larger than certain constant. The condition on $\epsilon$ can be satisfied as long as $\eta \leq \frac{1}{c_5 d^2 \log^2(d)}$ for some constant $c_5$. $\square$

### D.2.2 GENERALIZATION FOR $\eta \in [0, \frac{1}{c_5 d^2 \log^2 d}]$

In this section, we prove Lemma 37 by showing that $\hat{F}_{TbV}(\eta)$ is point-wise close to $F_{TbV}(\eta)$ for all $\eta \in [0, \frac{1}{c_5 d^2 \log^2(d/\epsilon)}]$. Recall Lemma 37 as follows.

**Lemma 37.** *For any $1 > \epsilon > 0$, assume $\sigma$ is a constant and $d \geq c_4 \log(1/\epsilon)$ for some constant $c_4$. There exists constant $c_5$ such that with probability at least $1 - O(1/\epsilon)\exp(-\Omega(\epsilon^2 m))$,*

$$|\hat{F}_{TbV}(\eta) - F_{TbV}(\eta)| \leq \epsilon,$$

*for all $\eta \in [0, \frac{1}{c_5 d^2 \log^2(d/\epsilon)}]$.*

In order to prove Lemma 37, we first show that for a fixed $\eta$ with high probability $\hat{F}_{TbV}(\eta)$ is close to $F_{TbV}(\eta)$. Similar as in Lemma 16, we can still show that each $\Delta_{TbV}(\eta, P)$ is $O(1)$-subexponential. The proof is deferred into Section D.2.3.

**Lemma 40.** *Suppose $\sigma$ is a constant. Given any $1 > \epsilon > 0$, for any fixed $\eta$ with probability at least $1 - \exp(-\Omega(\epsilon^2 m))$,*

$$\left|\hat{F}_{TbV}(\eta) - F_{TbV}(\eta)\right| \leq \epsilon.$$

Next, we show that there exists an $\epsilon$-net for $F_{TbV}$ with size $O(1/\epsilon)$. By $\epsilon$-net, we mean there exists a finite set $N_\epsilon$ of step sizes such that $|F_{TbV}(\eta) - F_{TbV}(\eta')| \leq \epsilon$ for any $\eta$ and $\eta' \in \arg\min_{\eta \in N_\epsilon} |\eta - \eta'|$. The proof is very similar as in Lemma 17. We defer the proof of Lemma 41 into Section D.2.3.

**Lemma 41.** *Suppose $\sigma$ is a constant. For any $1 > \epsilon > 0$, assume $d \geq c_4 \log(1/\epsilon)$ for some $c_4$. There exists constant $c_5$ and an $\epsilon$-net $N_\epsilon \subset [0, \frac{1}{c_5 d^2 \log^2(d/\epsilon)}]$ for $F_{TbV}$ with $|N_\epsilon| = O(1/\epsilon)$. That means, for any $\eta \in [0, \frac{1}{c_5 d^2 \log^2(d/\epsilon)}]$,*

$$|F_{TbV}(\eta) - F_{TbV}(\eta')| \leq \epsilon,$$

*for $\eta' \in \arg\min_{\eta \in N_\epsilon} |\eta - \eta'|$.*

Next, we show that with high probability, there also exists an $\epsilon$-net for $\hat{F}_{TbV}$ with size $O(1/\epsilon)$. The proof is very similar as the proof of Lemma 18. We defer the proof into Section D.2.3.

**Lemma 42.** *Suppose $\sigma$ is a constant. For any $1 > \epsilon > 0$, assume $d \geq c_4 \log(1/\epsilon)$ for some $c_4$. With probability at least $1 - \exp(-\Omega(\epsilon^2 m))$, there exists constant $c_5$ and an $\epsilon$-net $N'_\epsilon \subset [0, \frac{1}{c_5 d^2 \log^2(d/\epsilon)}]$ for $\hat{F}_{TbV}$ with $|N_\epsilon| = O(1/\epsilon)$. That means, for any $\eta \in [0, \frac{1}{c_5 d^2 \log^2(d/\epsilon)}]$,*

$$|\hat{F}_{TbV}(\eta) - \hat{F}_{TbV}(\eta')| \leq \epsilon,$$

*for $\eta' \in \arg\min_{\eta \in N_\epsilon} |\eta - \eta'|$.*

Combing Lemma 40, Lemma 41 and Lemma 42, now we give the proof of Lemma 37.

**Proof of Lemma 37.** The proof is almost the same as the proof of Lemma 11. We omit the details here. $\qquad\square$

### D.2.3 PROOFS OF TECHNICAL LEMMAS

In Lemma 43, we show when the step size is small, the expected SGD noise square is well bounded. The proof follows from the analysis in Lemma 33.

**Lemma 43.** *Let $\{w'_{\tau,\eta}\}$ be an SGD sequence running on task $P$ without truncation. Let $n'_{\tau,\eta}$ be the SGD noise at $w'_{\tau,\eta}$. Assume $\sqrt{d}/\sqrt{L} \leq \sigma_i(X_{train}) \leq \sqrt{L}\sqrt{\sigma}$ for all $i \in [n]$ and $\|\xi_{train}\| \leq \sqrt{d}\sigma$. Suppose $\eta \in [0, \frac{1}{2L^3 d}]$, we have*

$$\mathbb{E}_{SGD} \left\| n'_{\tau,\eta} \right\|^2 \leq 4L^3 \sigma^2 d$$

*for all $\tau \leq t$.*

**Proof of Lemma 43.** Similar as the analysis in Lemma 33, for $\eta \leq \frac{1}{2L^3 d}$, we have

$$\mathbb{E}_{\text{SGD}} \left[ \left\| n'_{\tau,\eta} \right\|^2 |w'_{\tau-1,\eta} \right] \leq L^2 d \left\| w'_{\tau-1,\eta} - w_{\text{train}} \right\|^2 .$$

and

$$\mathbb{E}_{\text{SGD}} \left\| w'_{\tau-1,\eta} - w_{\text{train}} \right\|^2 \leq (1 - \frac{\eta}{2L})^{\tau-1} \left\| w_{\text{train}} \right\|^2 \leq \left\| w^*_{\text{train}} + (X_{\text{train}})^\dagger \xi_{\text{train}} \right\|^2 \leq 4L\sigma^2.$$

Therefore, we have

$$\mathbb{E}_{\text{SGD}} \left\| n'_{\tau,\eta} \right\|^2 \leq L^2 d \mathbb{E}_{\text{SGD}} \left\| w'_{\tau,\eta} - w_{\text{train}} \right\|^2 \leq 4L^3 \sigma^2 d.$$

$\qquad\square$

**Proof of Lemma 38.** We can expand $Q'(\eta)$ as follows,

$$Q'(\eta) := \frac{1}{2} \mathbb{E}_{\text{SGD}} \left\| w'_{t,\eta} - w^* \right\|^2$$

$$= \frac{1}{2} \mathbb{E}_{\text{SGD}} \left\| B_{t,\eta} w^*_{\text{train}} + B_{t,\eta}(X_{\text{train}})^\dagger \xi_{\text{train}} - \eta \sum_{\tau=0}^{t-1} (I - \eta H_{\text{train}})^{t-1-\tau} n'_{\tau,\eta} - w^* \right\|^2$$

$$= \frac{1}{2} \left\| B_{t,\eta} w^*_{\text{train}} - w^* \right\|^2 + \frac{1}{2} \left\| B_{t,\eta}(X_{\text{train}})^\dagger \xi_{\text{train}} \right\|^2 + \frac{\eta^2}{2} \mathbb{E}_{\text{SGD}} \left\| \sum_{\tau=0}^{t-1} (I - \eta H_{\text{train}})^{t-1-\tau} n'_{\tau,\eta} \right\|^2$$

$$+ \left\langle B_{t,\eta} w^*_{\text{train}} - w^*, B_{t,\eta}(X_{\text{train}})^\dagger \xi_{\text{train}} \right\rangle$$

Denote

$$G(\eta) := \frac{1}{2}\left\| B_{t,\eta} w^*_{\text{train}} - w^* \right\|^2 + \frac{1}{2}\left\| B_{t,\eta}(X_{\text{train}})^\dagger \xi_{\text{train}} \right\|^2 + \frac{\eta^2}{2}\mathbb{E}_{\text{SGD}}\left\| \sum_{\tau=0}^{t-1}(I - \eta H_{\text{train}})^{t-1-\tau} n'_{\tau,\eta} \right\|^2.$$

We first show that with probability at least $1 - \exp(-\Omega(d))$, there exist $\eta_1, \eta_2, \eta_3 = \Theta(1/t)$ with $\eta_1 < \eta_2 < \eta_3$ such that $G(\eta_2) \le 1/2\left\| w^* \right\|^2 - 5C/4$ and $G(\eta) \ge 1/2\left\| w^* \right\|^2 - C/4$ for all $\eta \in [0, \eta_1] \cup [\eta_3, 1/L]$.

According to Lemma 1, we know with probability at least $1 - \exp(-\Omega(d))$, $\sqrt{d}/\sqrt{L} \le \sigma_i(X_{\text{train}}) \le \sqrt{L}\sqrt{d}$ and $1/L \le \lambda_i(H_{\text{train}}) \le L$ for all $i \in [n]$. According to Lemma 45, we know with probability at least $1 - \exp(-\Omega(d))$, $\sqrt{d}\sigma/4 \le \left\| \xi_{\text{train}} \right\| \le \sqrt{d}\sigma$.

**Upper bounding $G(\eta_2)$:**  We can expand $G(\eta)$ as follows:

$$
\begin{aligned}
G(\eta) :=& \frac{1}{2}\left\| B_{t,\eta} w^*_{\text{train}} - w^* \right\|^2 + \frac{1}{2}\left\| B_{t,\eta}(X_{\text{train}})^\dagger \xi_{\text{train}} \right\|^2 + \frac{\eta^2}{2}\mathbb{E}_{\text{SGD}}\left\| \sum_{\tau=0}^{t-1}(I - \eta H_{\text{train}})^{t-1-\tau} n'_{\tau,\eta} \right\|^2 \\
=& \frac{1}{2}\left\| w^* \right\|^2 + \frac{1}{2}\left\| B_{t,\eta} w^*_{\text{train}} \right\|^2 + \frac{1}{2}\left\| B_{t,\eta}(X_{\text{train}})^\dagger \xi_{\text{train}} \right\|^2 + \frac{\eta^2}{2}\mathbb{E}_{\text{SGD}}\left\| \sum_{\tau=0}^{t-1}(I - \eta H_{\text{train}})^{t-1-\tau} n'_{\tau,\eta} \right\|^2 \\
& - \left\langle B_{t,\eta} w^*_{\text{train}}, w^* \right\rangle.
\end{aligned}
$$

Same as in Lemma 13, we know $\frac{1}{2}\left\| B_{t,\eta} w^*_{\text{train}} \right\|^2 + \frac{1}{2}\left\| B_{t,\eta}(X_{\text{train}})^\dagger \xi_{\text{train}} \right\|^2 \le L^3 \eta^2 t^2 \sigma^2$. For the SGD noise, by Lemma 43 we know $\mathbb{E}_{\text{SGD}}\left\| n'_{\tau,\eta} \right\|^2 \le 4L^3\sigma^2 d$ for all $\tau \le t$ as long as $\eta \le \frac{1}{2L^3 d}$. Therefore,

$$\frac{\eta^2}{2}\mathbb{E}_{\text{SGD}}\left\| \sum_{\tau=0}^{t-1}(I - \eta H_{\text{train}})^{t-1-\tau} n'_{\tau,\eta} \right\|^2 \le \frac{\eta^2}{2}\sum_{\tau=0}^{t-1}\mathbb{E}_{\text{SGD}}\left\| n'_{\tau,\eta} \right\|^2 \le 2L^3\eta^2\sigma^2 dt \le 2L^3\eta^2\sigma^2 t^2,$$

where the last inequality assumes $t \ge d$. According to Lemma 15, for any fixed $\eta \in [0, L/t]$, with probability at least $1 - \exp(-\Omega(d))$ over $X_{\text{train}}$,

$$\left\langle B_{t,\eta} w^*_{\text{train}}, w^* \right\rangle \ge \frac{\eta t}{16L}.$$

Therefore, for any step size $\eta \le \frac{1}{2L^3 d}$,

$$G(\eta) \le \frac{1}{2}\left\| w^* \right\|^2 + 3L^3\eta^2\sigma^2 t^2 - \frac{\eta t}{16L} \le \frac{1}{2}\left\| w^* \right\|^2 - \frac{\eta t}{32L},$$

where the second inequality holds as long as $\eta \le \frac{1}{96L^4\sigma^2 t}$. Choosing $\eta_2 := \frac{1}{96L^4\sigma^2 t}$ that is smaller than $\frac{1}{2L^3 d}$ assuming $t \ge d$. Then, we have

$$G(\eta_2) \le \frac{1}{2}\left\| w^* \right\|^2 - \frac{5C}{4},$$

where constant $C = \frac{1}{3072L^5\sigma^2}$.

**Lower bounding $G(\eta)$ for $\eta \in [0, \eta_1]$ :**  Now, we prove that there exists $\eta_1 = \Theta(1/t)$ with $\eta_1 < \eta_2$ such that for any $\eta \in [0, \eta_1]$, $G(\eta) \ge \frac{1}{2}\left\| w^* \right\|^2 - \frac{C}{4}$. Recall that

$$
\begin{aligned}
G(\eta) =& \frac{1}{2}\left\| w^* \right\|^2 + \frac{1}{2}\left\| B_{t,\eta} w^*_{\text{train}} \right\|^2 + \frac{1}{2}\left\| B_{t,\eta}(X_{\text{train}})^\dagger \xi_{\text{train}} \right\|^2 + \frac{\eta^2}{2}\mathbb{E}_{\text{SGD}}\left\| \sum_{\tau=0}^{t-1}(I - \eta H_{\text{train}})^{t-1-\tau} n'_{\tau,\eta} \right\|^2 \\
& - \left\langle B_{t,\eta} w^*_{\text{train}}, w^* \right\rangle. \\
\ge& \frac{1}{2}\left\| w^* \right\|^2 - \left\langle B_{t,\eta} w^*_{\text{train}}, w^* \right\rangle.
\end{aligned}
$$

Same as in Lemma 13, by choosing $\eta_1 = \frac{C}{4Lt}$, we have for any $\eta \in [0, \eta_1]$,

$$G(\eta) \ge \frac{1}{2}\left\| w^* \right\|^2 - \frac{C}{4}.$$

**Lower bounding $G(\eta)$ for $\eta \in [\eta_3, 1/L]$:** Now, we prove that there exists $\eta_3 = \Theta(1/t)$ with $\eta_3 > \eta_2$ such that for all $\eta \in [\eta_3, 1/L]$,

$$G(\eta) \geq \frac{1}{2}\|w^*\|^2 - \frac{C}{4}.$$

Recall that

$$G(\eta) = \frac{1}{2}\left\|B_{t,\eta}w^*_{\text{train}} - w^*\right\|^2 + \frac{1}{2}\left\|B_{t,\eta}(X_{\text{train}})^\dagger \xi_{\text{train}}\right\|^2 + \frac{\eta^2}{2}\mathbb{E}_{\text{SGD}}\left\|\sum_{\tau=0}^{t-1}(I - \eta H_{\text{train}})^{t-1-\tau} n'_{\tau,\eta}\right\|^2$$

$$\geq \frac{1}{2}\left\|B_{t,\eta}(X_{\text{train}})^\dagger \xi_{\text{train}}\right\|^2.$$

Same as in Lemma 13, by choosing $\eta_3 = \log(2)L/t$, as long as $\sigma \geq 8\sqrt{L}$, we have

$$G(\eta) \geq \frac{1}{2}\|w^*\|^2$$

for all $\eta \in [\eta_3, 1/L]$. Note $\eta_3 \leq 1/L$ as long as $t \geq \log(2)L^2$.

Overall, we have shown that there exist $\eta_1, \eta_2, \eta_3 = \Theta(1/t)$ with $\eta_1 < \eta_2 < \eta_3$ such that $G(\eta_2) \leq 1/2\|w^*\|^2 - 5C/4$ and $G(\eta) \geq 1/2\|w^*\|^2 - C/4$ for all $\eta \in [0, \eta_1] \cup [\eta_3, 1/L]$. Recall that $Q'(\eta) = G(\eta) + \langle B_{t,\eta}w^*_{\text{train}} - w^*, B_{t,\eta}(X_{\text{train}})^\dagger \xi_{\text{train}}\rangle$. Choosing $\epsilon = C/4$ in Lemma 14, we know with probability at least $1 - \exp(-\Omega(d))$, $\left|\langle B_{t,\eta}w^*_{\text{train}} - w^*, B_{t,\eta}(X_{\text{train}})^\dagger \xi_{\text{train}}\rangle\right| \leq C/4$ for all $\eta \in [0, 1/L]$. Therefore, we know $Q'(\eta_2) \leq 1/2\|w^*\|^2 - C$ and $Q'(\eta) \geq 1/2\|w^*\|^2 - C/2$ for all $\eta \in [0, \eta_1] \cup [\eta_3, 1/L]$. $\qquad\square$

In order to prove Lemma 39, we first construct a super-martingale to show that as long as task $P$ is well behaved, with high probability in SGD noise, the weight norm along the trajectory never exceeds $4\sqrt{L}\sigma$.

**Lemma 44.** *Assume $\sqrt{d}/\sqrt{L} \leq \sigma_i(X_{train}) \leq \sqrt{L}d$ and $1/L \leq \lambda_i(H_{train}) \leq L$ for all $i \in [n]$ and $\sqrt{d}\sigma/4 \leq \|\xi_{train}\| \leq \sqrt{d}\sigma$. Given any $1 > \delta > 0$, suppose $\eta \leq \frac{1}{c_5 d^2 \log^2(d/\delta)}$ for some constant $c_5$, with probability at least $1 - \delta$ in the SGD noise,*

$$\left\|w'_{\tau,\eta}\right\| < 4\sqrt{L}\sigma$$

*for all $\tau \leq t$.*

**Proof of Lemma 44.** According to the proofs of Lemma 43, as long as $\eta \leq \frac{1}{2L^3 d}$, we have

$$\mathbb{E}_{\text{SGD}}\left[\left\|w'_{t,\eta} - w_{\text{train}}\right\|^2 | w'_{t-1,\eta}\right] \leq (1 - \frac{\eta}{2L})\left\|w'_{t-1,\eta} - w_{\text{train}}\right\|^2.$$

Since $\log$ is a concave function, by Jenson's inequality, we know

$$\mathbb{E}_{\text{SGD}}\left[\log\left\|w'_{t,\eta} - w_{\text{train}}\right\|^2 | w'_{t-1,\eta}\right]$$

$$\leq \log \mathbb{E}_{\text{SGD}}\left[\left\|w'_{t,\eta} - w_{\text{train}}\right\|^2 | w'_{t-1,\eta}\right] \leq \log\left\|w'_{t-1,\eta} - w_{\text{train}}\right\|^2 + \log(1 - \frac{\eta}{2L}).$$

Defining $G_t = \log\left\|w'_{t,\eta} - w_{\text{train}}\right\|^2 - t\log(1 - \frac{\eta}{2L})$, we know $G_t$ is a super-martingale. Next, we bound the martingale differences.

We can bound $|G_t - \mathbb{E}_{\text{SGD}}[G_t|w'_{t-1,\eta}]|$ as follows,

$$|G_t - \mathbb{E}_{\text{SGD}}[G_t|w'_{t-1,\eta}]| \leq \max_{n'_{t-1,\eta}, n''_{t-1,\eta}} \log\left(\frac{\left\|(I - \eta H_{\text{train}})(w'_{t-1,\eta} - w_{\text{train}}) - \eta n'_{t-1,\eta}\right\|^2}{\left\|(I - \eta H_{\text{train}})(w'_{t-1,\eta} - w_{\text{train}}) - \eta n''_{t-1,\eta}\right\|^2}\right)$$

We can expand $\left\|(I - \eta H_{\text{train}})(w'_{t-1,\eta} - w_{\text{train}}) - \eta n'_{t-1,\eta}\right\|^2$ as follows,

$$\left\|(I - \eta H_{\text{train}})(w'_{t-1,\eta} - w_{\text{train}}) - \eta n'_{t-1,\eta}\right\|^2$$

$$= \left\|(I - \eta H_{\text{train}})(w'_{t-1,\eta} - w_{\text{train}})\right\|^2 - 2\eta\langle n'_{t-1,\eta}, (I - \eta H_{\text{train}})(w'_{t-1,\eta} - w_{\text{train}})\rangle + \eta^2\left\|n'_{t-1,\eta}\right\|^2$$

We can bound the norm of the noise as follows,

$$
\begin{aligned}
\left\| n'_{t-1,\eta} \right\| &= \left\| x_{i(t-1)} x_{i(t-1)}^\top (w'_{t-1,\eta} - w_{\text{train}}) - H_{\text{train}}(w'_{t-1,\eta} - w_{\text{train}}) \right\| \\
&\leq \left\| x_{i(t-1)} x_{i(t-1)}^\top (w'_{t-1,\eta} - w_{\text{train}}) \right\| + \left\| H_{\text{train}}(w'_{t-1,\eta} - w_{\text{train}}) \right\| \\
&\leq (Ld + L) \left\| w'_{t-1,\eta} - w_{\text{train}} \right\| \leq 2Ld \left\| w'_{t-1,\eta} - w_{\text{train}} \right\|,
\end{aligned}
$$

where the second inequality uses $\left\| x_{i(t-1)} \right\| \leq \sqrt{Ld}$. Therefore, we have

$$
\begin{aligned}
\left| 2\eta \left\langle n'_{t-1,\eta}, (I - \eta H_{\text{train}})(w'_{t-1,\eta} - w_{\text{train}}) \right\rangle \right| &\leq 4L\eta d \left\| w'_{t-1,\eta} - w_{\text{train}} \right\|^2, \\
\eta^2 \left\| n'_{t-1,\eta} \right\|^2 &\leq 4L^2\eta^2 d^2 \left\| w'_{t-1,\eta} - w_{\text{train}} \right\|^2.
\end{aligned}
$$

This further implies,

$$
\begin{aligned}
& \left| G_t - \mathbb{E}_{\text{SGD}}[G_t | w'_{t-1,\eta}] \right| \\
& \leq \log \left( \frac{\left\| (I - \eta H_{\text{train}})(w'_{t-1,\eta} - w_{\text{train}}) \right\|^2 + \left(4L\eta d + 4L^2\eta^2 d^2\right) \left\| w'_{t-1,\eta} - w_{\text{train}} \right\|^2}{\left\| (I - \eta H_{\text{train}})(w'_{t-1,\eta} - w_{\text{train}}) \right\|^2 - 4L\eta d \left\| w'_{t-1,\eta} - w_{\text{train}} \right\|^2} \right) \\
& \leq \log \left( 1 + \frac{8L\eta d + 4L^2\eta^2 d^2}{(1 - 2L\eta - 4L\eta d)} \right) \leq 16L\eta d + 8L^2\eta^2 d^2,
\end{aligned}
$$

where the second inequality uses $\left\| (I - \eta H_{\text{train}})(w'_{t-1,\eta} - w_{\text{train}}) \right\|^2 \geq (1 - 2L\eta) \left\| w'_{t-1,\eta} - w_{\text{train}} \right\|^2$. The last inequality assumes $\eta \leq \frac{1}{12Ld}$ and uses numerical inequality $\log(1 + x) \leq x$. Assuming $\eta \leq 1/(Ld)$, we further have $\left| G_t - \mathbb{E}_{\text{SGD}}[G_t | w'_{t-1,\eta}] \right| \leq L^2\eta d$.

By Azuma's inequality, we know with probability at least $1 - \delta/t$,

$$
G_t \leq G_0 + L^2 \sqrt{2t} \eta d \log(t/\delta).
$$

Plugging in $G_t = \log \left\| w'_{t,\eta} - w_{\text{train}} \right\|^2 - t \log(1 - \frac{\eta}{2L})$ and $G_0 = \log \left\| w_0 - w_{\text{train}} \right\|^2 = \log \left\| w_{\text{train}} \right\|^2$, we have

$$
\begin{aligned}
\log \left\| w'_{t,\eta} - w_{\text{train}} \right\|^2 &\leq \log \left\| w_{\text{train}} \right\|^2 + t \log(1 - \frac{\eta}{2L}) + L^2 \sqrt{2t} \eta d \log(t/\delta) \\
&\leq \log \left\| w_{\text{train}} \right\|^2 - \frac{\eta}{2L} t + L^2 \sqrt{2t} \eta d \log(t/\delta).
\end{aligned}
$$

This implies,

$$
\begin{aligned}
\left\| w'_{t,\eta} - w_{\text{train}} \right\|^2 &\leq \left\| w_{\text{train}} \right\|^2 \exp \left( \eta \left( -\frac{1}{2L} t + L^2 \sqrt{2} \log(t/\delta) d \sqrt{t} \right) \right) \\
&= \left\| w_{\text{train}} \right\|^2 \exp \left( O(d^2 \log^2(d/\delta)) \eta \right) \\
&\leq \left\| w_{\text{train}} \right\|^2 \exp \left( 2/3 \right),
\end{aligned}
$$

where the second inequality assumes $\eta \leq \frac{1}{c_5 d^2 log^2(d/\delta)}$ for some constant $c_5$. Furthermore, since $\left\| w_{\text{train}} \right\| \leq (1 + \sqrt{L})\sigma$, we have $\left\| w'_{t,\eta} \right\| \leq (1 + e^{1/3}) \left\| w_{\text{train}} \right\| < 4\sqrt{L}\sigma$.

Overall, we know as long as $\eta \leq \frac{1}{c_5 d^2 log^2(d/\delta)}$, with probability at least $1 - \delta/t$, $\left\| w'_{t,\eta} \right\| \leq 4\sqrt{L}\sigma$. Since this analysis also applies to any $\tau \leq t$, we know for any $\tau$, with probability at least $1 - \delta/t$, $\left\| w'_{\tau,\eta} \right\| < 4\sqrt{L}\sigma$. Taking a union bound over $\tau \leq t$, we have with probability at least $1 - \delta$, $\left\| w'_{\tau,\eta} \right\| < 4\sqrt{L}\sigma$ for all $\tau \leq t$. $\qquad \square$

**Proof of Lemma 39.** Let $\mathcal{E}$ be the event that $\left\| w'_{\tau,\eta} \right\| < 4\sqrt{L}\sigma$ for all $\tau \leq t$. We first show that $\mathbb{E}_{\text{SGD}} \left\| w_{t,\eta} - w^* \right\|^2$ is close to $\mathbb{E}_{\text{SGD}} \left\| w'_{t,\eta} - w^* \right\|^2 \mathbb{1}\{\mathcal{E}\}$. It's not hard to verify that

$$
\mathbb{E}_{\text{SGD}} \left\| w_{t,\eta} - w^* \right\|^2 = \mathbb{E}_{\text{SGD}} \left\| w'_{t,\eta} - w^* \right\|^2 \mathbb{1}\{\mathcal{E}\} + \left\| u - w^* \right\|^2 \Pr[\bar{\mathcal{E}}],
$$

where $u$ is a fixed vector with norm $4\sqrt{L}\sigma$. By Lemma 44, we know $\Pr[\bar{\mathcal{E}}] \leq \epsilon/(25L\sigma^2)$ as long as $\eta \leq \frac{1}{c_5 d^2 \log^2(d/\epsilon)}$ for some constant $c_5$. Therefore, we have

$$\left| \mathbb{E}_{\text{SGD}} \left\| w_{t,\eta} - w^* \right\|^2 - \mathbb{E}_{\text{SGD}} \left\| w'_{t,\eta} - w^* \right\|^2 \mathbb{1}\{\mathcal{E}\} \right| \leq \epsilon.$$

Next, we show that $\mathbb{E}_{\text{SGD}} \left\| w'_{t,\eta} - w^* \right\|^2 \mathbb{1}\{\mathcal{E}\}$ is close to $\mathbb{E}_{\text{SGD}} \left\| w'_{t,\eta} - w^* \right\|^2$. For any $1 \leq \tau \leq t$, let $\mathcal{E}_\tau$ be the event that $\left\| w'_{\tau,\eta} \right\| \geq 4\sqrt{L}\sigma$ and $\left\| w'_{\tau',\eta} \right\| < 4\sqrt{L}\sigma$ for all $\tau' < \tau$. Basically $\mathcal{E}_\tau$ means the weight norm exceeds the threshold at step $\tau$ for the first time. It's easy to see that $\cup_{\tau=1}^t \mathcal{E}_\tau = \bar{\mathcal{E}}$. Therefore, we have

$$\mathbb{E}_{\text{SGD}} \left\| w'_{t,\eta} - w^* \right\|^2 = \mathbb{E}_{\text{SGD}} \left\| w'_{t,\eta} - w^* \right\|^2 \mathbb{1}\{\mathcal{E}\} + \sum_{\tau=1}^t \mathbb{E}_{\text{SGD}} \left\| w'_{t,\eta} - w^* \right\|^2 \mathbb{1}\{\mathcal{E}_\tau\}.$$

Conditioning on $\mathcal{E}_\tau$, we know $\left\| w'_{\tau-1,\eta} \right\| < 4\sqrt{L}\sigma$. Since we assume $\frac{\sqrt{d}}{\sqrt{L}} \leq \sigma_i(X_{\text{train}}) \leq \sqrt{L}\sqrt{d}$ for all $i \in [n]$ and $\xi_{\text{train}} \leq \sqrt{d}\sigma$, we know $\left\| w_{\text{train}} \right\| \leq 2\sqrt{L}\sigma$. Therefore, we have $\left\| w'_{\tau-1,\eta} - w_{\text{train}} \right\| \leq 6\sqrt{L}\sigma$. Recall the SGD updates,

$$w'_{\tau,\eta} - w_{\text{train}} = (I - \eta H_{\text{train}})(w'_{\tau-1,\eta} - w_{\text{train}}) - \eta n'_{\tau-1,\eta}.$$

For the noise term, we have $\eta \left\| n'_{\tau-1,\eta} \right\| \leq 2\eta L d \left\| w'_{\tau-1,\eta} - w_{\text{train}} \right\|$ that is at most $\left\| w'_{\tau-1,\eta} - w_{\text{train}} \right\|$ assuming $\eta \leq \frac{1}{2Ld}$. Therefore, we have $\left\| w'_{\tau,\eta} - w_{\text{train}} \right\| \leq 2 \left\| w'_{\tau-1,\eta} - w_{\text{train}} \right\| \leq 12\sqrt{L}\sigma$. Note that event $\mathcal{E}_\tau$ is independent with the SGD noises after step $\tau$. Therefore, according to the previous analysis, we know as long as $\eta \leq \frac{1}{2L^3 d}$,

$$\mathbb{E}_{\text{SGD}} \left[ \left\| w'_{t,\eta} - w_{\text{train}} \right\|^2 | \mathcal{E}_\tau \right] \leq \left\| w'_{\tau,\eta} - w_{\text{train}} \right\|^2 \leq 2L^2\sigma^2.$$

Then, we can bound $\mathbb{E}_{\text{SGD}} \left[ \left\| w'_{t,\eta} - w^* \right\|^2 | \mathcal{E}_\tau \right]$ as follows,

$$\mathbb{E}_{\text{SGD}} \left[ \left\| w'_{t,\eta} - w^* \right\|^2 | \mathcal{E}_\tau \right]$$

$$= \mathbb{E}_{\text{SGD}} \left[ \left\| w'_{t,\eta} - w_{\text{train}} + w_{\text{train}} - w^* \right\|^2 | \mathcal{E}_\tau \right]$$

$$\leq \mathbb{E}_{\text{SGD}} \left[ \left\| w'_{t,\eta} - w_{\text{train}} \right\|^2 | \mathcal{E}_\tau \right] + 2\mathbb{E}_{\text{SGD}} \left[ \left\| w'_{t,\eta} - w_{\text{train}} \right\| | \mathcal{E}_\tau \right] \left\| w_{\text{train}} - w^* \right\| + \left\| w_{\text{train}} - w^* \right\|^2$$

$$\leq 2L^2\sigma^2 + 2 \cdot 2L\sigma \cdot 3\sqrt{L}\sigma + 9L\sigma^2 \leq 3L^2\sigma^2.$$

Therefore, we have

$$\sum_{\tau=1}^t \mathbb{E}_{\text{SGD}} \left\| w'_{t,\eta} - w^* \right\|^2 \mathbb{1}\{\mathcal{E}_\tau\} = \sum_{\tau=1}^t \mathbb{E}_{\text{SGD}} \left[ \left\| w'_{t,\eta} - w^* \right\|^2 | \mathcal{E}_\tau \right] \Pr[\mathcal{E}_\tau]$$

$$\leq 3L^2\sigma^2 \sum_{\tau=1}^t \Pr[\mathcal{E}_\tau] = 3L^2\sigma^2 \Pr[\bar{\mathcal{E}}] \leq 3L^2\sigma^2\epsilon.$$

This then implies that $\left| \mathbb{E}_{\text{SGD}} \left\| w'_{t,\eta} - w^* \right\|^2 - \mathbb{E}_{\text{SGD}} \left\| w'_{t,\eta} - w^* \right\|^2 \mathbb{1}\{\mathcal{E}\} \right| \leq 3L^2\sigma^2\epsilon$.

Finally, we have

$$\left| \mathbb{E}_{\text{SGD}} \left\| w_{t,\eta} - w^* \right\|^2 - \mathbb{E}_{\text{SGD}} \left\| w'_{t,\eta} - w^* \right\|^2 \right|$$

$$\leq \left| \mathbb{E}_{\text{SGD}} \left\| w_{t,\eta} - w^* \right\|^2 - \mathbb{E}_{\text{SGD}} \left\| w'_{t,\eta} - w^* \right\|^2 \mathbb{1}\{\mathcal{E}\} \right| + \left| \mathbb{E}_{\text{SGD}} \left\| w'_{t,\eta} - w^* \right\|^2 - \mathbb{E}_{\text{SGD}} \left\| w'_{t,\eta} - w^* \right\|^2 \mathbb{1}\{\mathcal{E}\} \right|$$

$$\leq \left( 3L^2\sigma^2 + 1 \right) \epsilon$$

as long as $\eta \leq \frac{1}{c_5 d^2 \log^2(d/\epsilon)}$. Therefore, $|Q(\eta) - Q'(\eta)| \leq \left( 3L^2\sigma^2 + 1 \right) \epsilon/2$. Choosing $\epsilon' = \frac{2\epsilon}{(3L^2\sigma^2+1)}$ finishes the proof. $\qquad\square$

**Proof of Lemma 40.** Recall that

$$\hat{F}_{TbV}(\eta) := \frac{1}{m}\sum_{k=1}^{m}\Delta_{TbV}(\eta, P) = \frac{1}{m}\sum_{k=1}^{m}\mathbb{E}_{\text{SGD}}\frac{1}{2}\left\|w_{t,\eta}^{(k)} - w_{\text{valid}}^{(k)}\right\|_{H_{\text{valid}}^{(k)}}^{2}.$$

Similar as in Lemma 11, we can show $\frac{1}{2}\left\|w_{t,\eta}^{(k)} - w_{\text{valid}}^{(k)}\right\|_{H_{\text{valid}}^{(k)}}^{2}$ is $O(1)$-subexponential, which implies $\mathbb{E}_{\text{SGD}}\frac{1}{2}\left\|w_{t,\eta}^{(k)} - w_{\text{valid}}^{(k)}\right\|_{H_{\text{valid}}^{(k)}}^{2}$ is $O(1)$-subexponential. Therefore, $\hat{F}_{TbV}(\eta)$ is the average of $m$ i.i.d. $O(1)$-subexponential random variables. By standard concentration inequality, we know for any $1 > \epsilon > 0$, with probability at least $1 - \exp(-\Omega(\epsilon^2 m))$,

$$\left|\hat{F}_{TbV}(\eta) - F_{TbV}(\eta)\right| \le \epsilon.$$

$\square$

**Proof of Lemma 41.** Recall that

$$F_{TbV}(\eta) = \mathbb{E}_{P\sim\mathcal{T}}\mathbb{E}_{\text{SGD}}\frac{1}{2}\left\|w_{t,\eta} - w^*\right\|^2 + \sigma^2/2$$

We only need to construct an $\epsilon$-net for $\mathbb{E}_{P\sim\mathcal{T}}\mathbb{E}_{\text{SGD}}\frac{1}{2}\left\|w_{t,\eta} - w^*\right\|^2$. Let $\mathcal{E}$ be the event that $\sqrt{d}/\sqrt{L} \le \sigma_i(X_{\text{train}}) \le \sqrt{Ld}$ and $1/L \le \lambda_i(H_{\text{train}}) \le L$ for all $i \in [n]$ and $\sqrt{d}\sigma/4 \le \|\xi_{\text{train}}\| \le \sqrt{d}\sigma$ We have

$$\mathbb{E}_{P\sim\mathcal{T}}\mathbb{E}_{\text{SGD}}\frac{1}{2}\left\|w_{t,\eta} - w^*\right\|^2$$

$$= \mathbb{E}_{P\sim\mathcal{T}}\left[\frac{1}{2}\mathbb{E}_{\text{SGD}}\left\|w_{t,\eta} - w^*\right\|^2 |\mathcal{E}\right]\Pr[\mathcal{E}] + \mathbb{E}_{P\sim\mathcal{T}}\left[\frac{1}{2}\mathbb{E}_{\text{SGD}}\left\|w_{t,\eta} - w^*\right\|^2 |\bar{\mathcal{E}}\right]\Pr[\bar{\mathcal{E}}]$$

According to Lemma 39, we know conditioning on $\mathcal{E}$,

$$\left|\frac{1}{2}\mathbb{E}_{\text{SGD}}\left\|w_{t,\eta} - w^*\right\|^2 - \frac{1}{2}\mathbb{E}_{\text{SGD}}\left\|w_{t,\eta}' - w^*\right\|^2\right| \le \epsilon,$$

as long as $\eta \le \frac{1}{c_5 d^2 \log^2(d/\epsilon)}$. Note $\{w_{\tau,\eta}'\}$ is the SGD sequence without truncation.

For the second term, we have

$$\mathbb{E}_{P\sim\mathcal{T}}\left[\frac{1}{2}\mathbb{E}_{\text{SGD}}\left\|w_{t,\eta} - w^*\right\|^2 |\bar{\mathcal{E}}\right]\Pr[\bar{\mathcal{E}}] \le 13L\sigma^2\Pr[\bar{\mathcal{E}}] \le \epsilon,$$

where the last inequality assumes $\Pr[\bar{\mathcal{E}}] \le \frac{\epsilon}{13L\sigma^2}$. According to Lemma 1 and Lemma 45, we know $\Pr[\bar{\mathcal{E}}] \le \exp(-\Omega(d))$. Therefore, given any $\epsilon > 0$, we have $\Pr[\bar{\mathcal{E}}] \le \frac{\epsilon}{13L\sigma^2}$ as long as $d \ge c_4 \log(1/\epsilon)$ for some constant $c_4$.

Then, we only need to construct an $\epsilon$-net for $\mathbb{E}_{P\sim\mathcal{T}}\left[\frac{1}{2}\mathbb{E}_{\text{SGD}}\left\|w_{t,\eta}' - w^*\right\|^2 |\mathcal{E}\right]\Pr[\mathcal{E}]$. By the analysis in Lemma 33, it's not hard to prove

$$\left|\frac{\partial}{\partial\eta}\mathbb{E}_{P\sim\mathcal{T}}\left[\frac{1}{2}\mathbb{E}_{\text{SGD}}\left\|w_{t,\eta}' - w^*\right\|^2 |\mathcal{E}\right]\Pr[\mathcal{E}]\right| = O(1)t(1 - \frac{\eta}{2L})^{t-1},$$

for all $\eta \in [0, \frac{1}{c_5 d^2 \log^2(d/\epsilon)}]$. Similar as in Lemma 14, for any $\epsilon > 0$, we know there exists an $\epsilon$-net $N_\epsilon$ with size $O(1/\epsilon)$ such that for any $\eta \in [0, \frac{1}{c_5 d^2 \log^2(d/\epsilon)}]$,

$$\left|\mathbb{E}_{P\sim\mathcal{T}}\left[\frac{1}{2}\mathbb{E}_{\text{SGD}}\left\|w_{t,\eta}' - w^*\right\|^2 |\mathcal{E}\right]\Pr[\mathcal{E}] - \mathbb{E}_{P\sim\mathcal{T}}\left[\frac{1}{2}\mathbb{E}_{\text{SGD}}\left\|w_{t,\eta'}' - w^*\right\|^2 |\mathcal{E}\right]\Pr[\mathcal{E}]\right| \le \epsilon$$

for $\eta' \in \arg\min_{\eta\in N_\epsilon}|\eta - \eta'|$.

Combing with the bounds on $\left|\frac{1}{2}\mathbb{E}_{\text{SGD}}\left\|w_{t,\eta} - w^*\right\|^2 \mathbb{1}\{\mathcal{E}\} - \frac{1}{2}\mathbb{E}_{\text{SGD}}\left\|w_{t,\eta}' - w^*\right\|^2 \mathbb{1}\{\mathcal{E}\}\right|$ and $\mathbb{E}_{P\sim\mathcal{T}}\left[\frac{1}{2}\mathbb{E}_{\text{SGD}}\left\|w_{t,\eta} - w^*\right\|^2 |\bar{\mathcal{E}}\right]\Pr[\bar{\mathcal{E}}]$, we have for any $\eta \in [0, \frac{1}{c_5 d^2 \log^2(d/\epsilon)}]$,

$$F_{TbV}(\eta) - F_{TbV}(\eta') \le 4\epsilon$$

for $\eta' \in \arg\min_{\eta \in N_\epsilon} |\eta - \eta'|$. We finish the proof by replacing $4\epsilon$ by $\epsilon'$. $\qquad\square$

**Proof of Lemma 42.** The proof is very similar as the proof of Lemma 18. The only difference is that we need to first relate the SGD sequence with truncation to the SGD sequence without truncation and then bound the Lipschitzness on the SGD sequence without truncation (as we did in Lemma 41). We omit the details here. $\qquad\square$

# E  TOOLS

## E.1  NORM OF RANDOM VECTORS

We use the following lemma to bound the noise in least squares model.

**Lemma 45** (Theorem 3.1.1 in Vershynin (2018)). *Let $X = (X_1, X_2, \cdots, X_n) \in \mathbb{R}^n$ be a random vector with each entry independently sampled from $\mathcal{N}(0, 1)$. Then*

$$\Pr[\big|\|x\| - \sqrt{n}\big| \geq t] \leq 2\exp(-t^2/C^2),$$

*where $C$ is an absolute constant.*

## E.2  SINGULAR VALUES OF GAUSSIAN MATRICES

Given a random Gaussian matrix, in expectation its smallest and largest singular value can be bounded as follows.

**Lemma 46** (Theorem 5.32 in Vershynin (2010)). *Let $A$ be an $N \times n$ matrix whose entries are independent standard normal random variables. Then*

$$\sqrt{N} - \sqrt{n} \leq \mathbb{E}s_{\min}(A) \leq \mathbb{E}s_{\max}(A) \leq \sqrt{N} + \sqrt{n}$$

Lemma 47 shows a lipchitz function over i.i.d. Gaussian variables concentrate well on its mean. We use this lemma to argue for any fixed step size, the empirical meta objective concentrates on the population meta objective.

**Lemma 47** (Proposition 5.34 in Vershynin (2010)). *Let $f$ be a real valued Lipschitz function on $\mathbb{R}^n$ with Lipschitz constant $K$. Let $X$ be the standard normal random vector in $\mathbb{R}^n$. Then for every $t \geq 0$ one has*

$$\Pr[f(X) - \mathbb{E}f(X) \geq t] \leq \exp(-\frac{t^2}{2K^2}).$$

The following lemma shows a tall random Gaussian matrix is well-conditioned with high probability. The proof follows from Lemma 46 and Lemma 47. We use Lemma 48 to show the covariance matrix is well conditioned in the least squares model.

**Lemma 48** (Corollary 5.35 in Vershynin (2010)). *Let $A$ be an $N \times n$ matrix whose entries are independent standard normal random variables. Then for every $t \geq 0$ with probability at least $1 - 2\exp(-t^2/2)$ one has*

$$\sqrt{N} - \sqrt{n} - t \leq s_{\min}(A) \leq s_{\max}(A) \leq \sqrt{N} + \sqrt{n} + t$$

## E.3  JOHNSON-LINDENSTRAUSS LEMMA

We also use Johnson-Lindenstrauss Lemma in some of the lemmas. Johnson-Lindenstrauss Lemma tells us the projection of a fixed vector on a random subspace concentrates well as long as the subspace is reasonably large.

**Lemma 49** (Johnson & Lindenstrauss (1984)). *Let $P$ be a projection in $\mathbb{R}^d$ onto a random $n$-dimensional subspace uniformly distributed in $G_{d,n}$. Let $z \in \mathbb{R}^d$ be a fixed point and $\epsilon > 0$, then with probability at least $1 - 2\exp(-c\epsilon^2 n)$,*

$$(1 - \epsilon)\sqrt{\frac{n}{d}}\|z\| \leq \|Pz\| \leq (1 + \epsilon)\sqrt{\frac{n}{d}}\|z\|.$$

## F EXPERIMENT DETAILS

We describe the detailed settings of our experiments in Section F.1 and give more experimental results in Section F.2.

### F.1 EXPERIMENT SETTINGS

**Optimizing step size for quadratic objective**  In this experiment, we meta-train a learning rate for gradient descent on a fixed quadratic objective. Our goal is to show that the autograd module in popular deep learning softwares, such as Tensorflow, can have numerical issues when using the log-transformed meta objective. Therefore, we first implement the meta-training process with Tensorflow to see the results. We then re-implement the meta-training using the hand-derived meta-gradient (see Eqn 3) to compare the result.

A general setting for both implementations is as follows. The inner problem is fixed as a 20-dimensional quadratic objective as described in Section 3, and we use the log-transformed meta objective for training. The positive semi-definite matrix $H$ is generated by first sampling a $20 \times 20$ matrix $X$ with all entries drawn from the standard normal distribution and then setting $H = X^T X$. The initial point $w_0$ is drawn from standard normal as well. Note that we use the same quadratic problem (i.e., the same $H$ and $w_0$) throughout the meta-training. We do 1000 meta-training iterations, and collect results for different settings of the initial learning rate $\eta_0$ and the unroll length $t$.

We first implement the meta-training code with Tensorflow. Our code is adapted from Wichrowska et al. (2017) [2]. We use their global learning rate optimizer and specify the problem set to have only one quadratic objective instance. We implemented the quadratic objective class ourselves (the "MyQuadratic" class). We also turned off multiple advanced features in the original code, such as attention and second derivatives, by assigning their flags as false. This ensures that the experiments have exactly the same settings as we described. The meta-training learning rate is set to be 0.001, which is of similar scale as our next experiment. We also try RMSProp as the meta optimizer, which alleviates some of the numerical issues as it renormalizes the gradient, but our experiments show that even RMSProp is still much worse than our implementation.

We then implement the meta-training by hand to show the accurate training results that avoid numerical issues. Specifically, we compute the meta-gradient using Eq (3), where we also scaled the numerator and denominator as described in Claim 2 to avoid numerical issues. We use the algorithm suggested in Theorem 4, except we choose the meta-step size to be $1/(100\sqrt{k})$ as the constants in Theorem 4 were not optimized.

**Train-by-train vs. train-by-validation, synthetic data**  In this experiment, we find the optimal learning rate $\eta^*$ for least-squares problems trained in train-by-train and train-by-validation settings and then see how the learning rate works on new tasks.

Specifically, we generate 300 different 1000-dimensional least-squares tasks with noise as defined in Section 4 for inner-training and then use the meta-objectives defined in Eq (1) and (2) to find the optimal learning rate. The inner-training number of steps $t$ is set as 40. We try different sample sizes and different noise levels for comparison. Subsequently, in order to test how the two $\eta^*$ (for train-by-train and train-by-validation respectively) work, we use them on 10 test tasks (the same setting as the inner-training problem) and compute training and testing root mean squared error (RMSE).

Note that since we only need the final optimal $\eta^*$ found under the two meta-objective settings (regardless of how we find it), we do not need to actually do the meta-training. Instead, we do a grid search on the interval $[10^{-6}, 1]$, which is divided log-linearly to 25 candidate points. For both the train-by-train and train-by-validation settings, we average the meta-objectives over the 300 inner problems and see which $\eta$ minimizes this averaged meta-objective.

**Train-by-train vs. train-by-validation, MLP optimizer on MNIST**  To observe the trade-off between train-by-train and train-by-validation in a broader and more realistic case, we also do ex-

---

[2]Their open source code is available at https://github.com/tensorflow/models/tree/master/research/learned_optimizer

periments to meta-train an MLP optimizer as in Metz et al. (2019) to solve the MNIST classification problem. We use part of their code [3] to integrate with our code in the first experiment, and we use exactly the same default setting as theirs, which is summarized below.

The MLP optimizer is a trainable optimizer that works on each parameter separately. When doing inner-training, for each parameter, we first compute some statistics of that parameter (explained below), which are combined into a feature vector, and then feed that feature vector to a Muti-Layer Perceptron (MLP) with ReLU activations, which outputs two scalars, the update direction and magnitude. The update is computed as the direction times the exponential of the magnitude. The feature vector is 31-dimensional, which includes gradient, parameter value, first-order moving averages (5-dim), second-order moving averages (5-dim), normalized gradient (5-dim), reciprocal of square root second-order moving averages (5-dim) and a step embedding (9-dim). All moving averages are computed using 5 different decay rates (0.5, 0.9, 0.99, 0.999, 0.9999), and the step embedding is $\tanh$ distortion of the current number of steps divided by 9 different scales (3, 10, 30, 100, 300, 1000, 3000, 10000, 300000). After expanding the 31-dimensional feature vector for each parameter, we also normalize the set of vectors dimension-wise across all the parameters to have mean 0 and standard deviation 1 (except for the step embedding part). More details can be found in their original paper and original implementation.

The inner-training problem is defined as using a two-layer fully connected network (i.e., another "MLP") with ReLU activations to solve the classic MNIST 10-class classification problem. We use a very small network for computational efficiency, and the two layers have 100 and 20 neurons. We fix the cross-entropy loss as the inner-objective and use mini-batches of 32 samples when inner-training.

When we meta-train the MLP optimizer, we use exactly the same process as fixed in experiments by Wichrowska et al. (2017). We use 100 different inner problems by shuffling the 10 classes and also sampling a new subset of data if we do not use the complete MNIST data set. We run each of the problems with three inner-training trajectories starting with different initialization. Each inner-training trajectory is divided into a certain number of unrolled segments, where we compute the meta-objective and update the meta-optimizer after each segment. The number of unrolled segments in each trajectory is sampled from $10 + \text{Exp}(30)$, and the length of each segment is sampled from $50 + \text{Exp}(100)$, where $\text{Exp}(\cdot)$ denotes the exponential distribution. Note that the meta-objective computed after each segment is defined as the average of all the inner-objectives (evaluated on the train/validation set for train-by-train/train-by-val) within that segment for a better convergence. We also do not need to log-transform the inner-objective this time because the cross entropy loss has a log operator itself. The meta-training, i.e. training the parameters of the MLP in the MLP optimzier, is completed using a classic RMSProp optimizer with meta learning rate 0.01.

For each settings of sample sizes and noise levels, we train two MLP optimizer: one for train-by-train, and one for train-by-validation. When we test the learned MLP optimizer, we use similar settings as the inner-training problem, and we run the trajectories longer for full convergence (4000 steps for small data sets; 40000 steps for the complete data set). We run 5 independent tests and collect training accuracy and test accuracy for evaluation. The plots show the mean of the 5 tests. We have also tuned a SGD optimizer (with the same mini-batch size) by doing a grid-search of the learning rate as baseline.

### F.2 ADDITIONAL RESULTS

**Optimizing step size for quadratic objective**  We try experiments for the same settings of the initial $\eta_0$ and inner training length $t$ for all of three implementations (our hand-derived GD version, Tensorflow GD version and the Tensorflow RMSProp version). We do 1000 meta-training steps for all the experiments.

For both Tensorflow versions, we always see infinite meta-objectives if $\eta_0$ is large or $t$ is large, whose meta-gradient is usually treated as zero, so the training get stuck and never converge. Even for the case that both $\eta_0$ and $t$ is small, it still has very large meta-objectives (the scale of a few hundreds), and that is why we also try RMSProp, which should be more robust against the gradient scales. Our

---

[3]Their code is available at https://github.com/google-research/google-research/tree/master/task_specific_learned_opt

Table 1: Whether the implementation converges for different $t$ (fixed $\eta_0 = 0.1$)

| $t$ | 10 | 20 | 40 | 80 |
|---|---|---|---|---|
| Ours | ✓ | ✓ | ✓ | ✓ |
| Tensorflow GD | × | × | × | × |
| Tensorflow RMSProp | ✓ | ✓ | × | × |

Table 2: Whether the implementation converges for different $\eta_0$ (fixed $t = 40$)

| $\eta_0$ | 0.001 | 0.01 | 0.1 | 1 |
|---|---|---|---|---|
| Ours | ✓ | ✓ | ✓ | ✓ |
| Tensorflow GD | × | × | × | × |
| Tensorflow RMSProp | ✓ | ✓ | × | × |

hand-derived version, however, does not have the numerical issues and can always converge to the optimal $\eta^*$. The detailed convergence is summarized in Tab 1 and Tab 2. Note that the optimal $\eta^*$ is usually around 0.03 under our settings.

**Train-by-train vs. train-by-validation, MLP optimizer on MNIST**  We also do additional experiments on training an MLP optimizer on the MNIST classification problem. We first try using all samples under the 20% noised setting. The results are shown in Fig 8. The train-by-train setting can perform well if we have a large data set, but since there is also noise in the data, the train-by-train model still overfits and is slightly worse than the train-by-validation model.

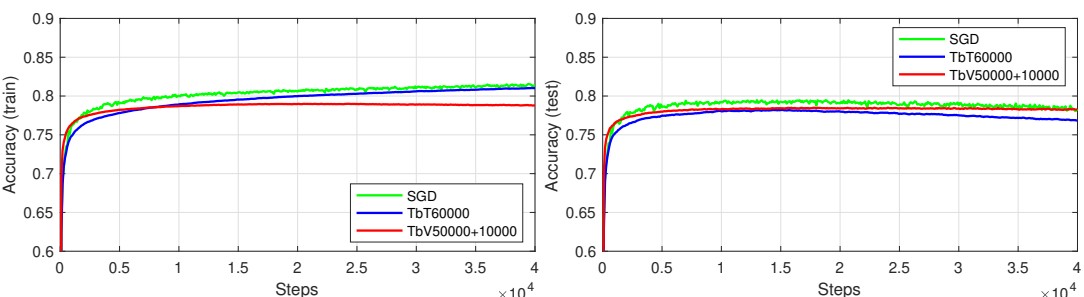

Figure 8: Training and testing accuracy for different models (all samples, 20% noise)

We then try an intermediate sample size 12000. The results are shown in Fig 9 (no noise) and Fig 10 (20% noise). We can see that as the theory predicts, as the amount of data increases (from 1000 samples to 12000 samples and then to 60000 samples) the gap between train-by-train and train-by-validation decreases. Also, when we condition on the same number of samples, having additional label noise always makes train-by-train model much worse compared to train-by-validation.

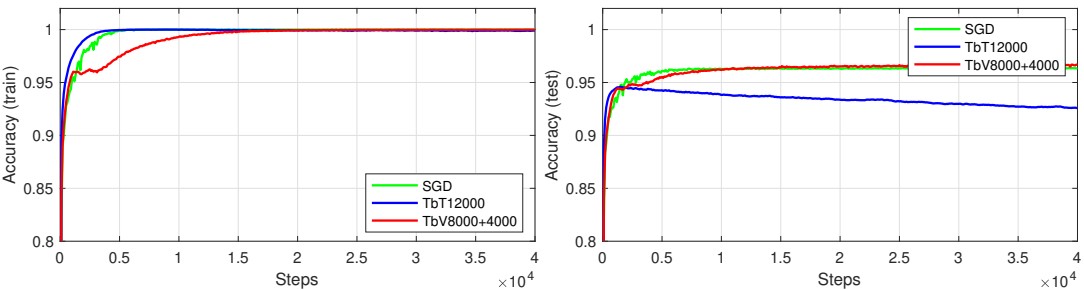

Figure 9: Training and testing accuracy for different models (12000 samples, no noise)

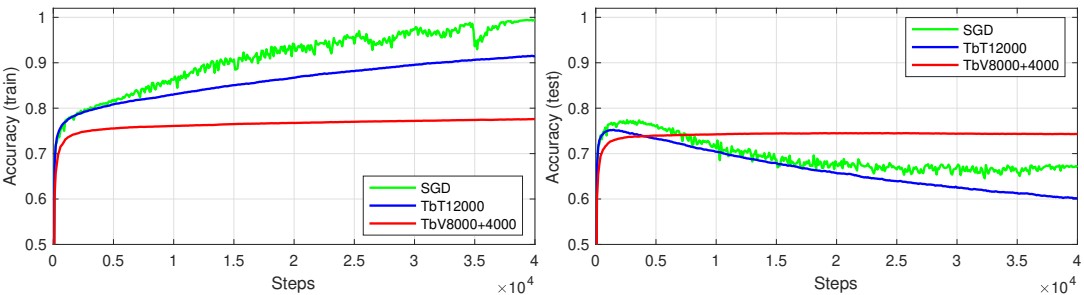

Figure 10: Training and testing accuracy for different models (12000 samples, 20% noise)

