# OpenReview forum: "Guarantees for Tuning the Step Size using a Learning-to-Learn Approach"
_ICLR.cc/2021/Conference — Reject_

### Official Review · AnonReviewer1 · 2020-10-27
**Great paper, long Appendix**

**Rating:** 8
**Confidence:** 3

**Review:**

## Overview
---

Meta-gradient descent is an approach to step-size adaptation in which the step-size is adapted by considering how it influences the loss function over time. Intuitively, one can think of the trajectory of parameters $(w_s)_{s=1}^t$ as being a function of the step-size $\eta$, and try to control the loss indirectly through the step-size's influence on the weight trajectory. This paper provides guarantees for this class of algorithms when applied to a quadradic loss function. It is shown that the meta-objective $\ell_t(\eta)=\frac{1}{2}w_t(\eta)^\top H w_t(\eta)$ contains no bad local solutions, but can suffer from vanishing/exploding gradients. It is then shown that this can be remedied simply considering the logarithm of this meta-objective, but that this too will have issues with numerical stability if approached with back-propagation. Finally, results related to the generalization ability of these methods are presented.

**Overall, I recommend the paper for acceptance.** Despite focusing on a simple quadratic loss setting, the results are quite non-trivial and I believe will be of interest to many in the community. The writing was clear throughout, and I found the proofs that I worked through (Appendix A) to be instructive and of high-quality. The experiments leave a bit to be desired but I think are generally successful in demonstrating the results suggested by the theory.

My main reservation about this paper is the length of the Appendix. On one hand, I think it's absolutely unreasonable to expect reviewers to work through and verify **60** pages of proofs, for free, on their own time, given a two-week deadline, so I'm inclined to say that this work is better-suited for a journal where the results can be verified properly by reviewers. However, I did work through Appendix A, and the material was well-explained and generally quite excellent, so I'm willing to believe that the rest of the appendix follows suit. But because I don't have time to work through the remaining 50 pages, I can't be sure of the quality and this is reflected in my score, which might otherwise be higher.

## Questions
---
- **Page 5**: *"Note we truncate a sequence and declare the meta loss is high once the weight norm exceeds certain threshold. Specifically, if at the $\tau^{\text{th}}$ step, $\|w_{\tau,\eta}\|\ge 40\sigma$, we freeze the training on this task and set $w_{\tau',\eta}= 40\sigma u$ for all $\tau\le \tau'\le t$, for some arbitrary vector $u$ with unit norm."* Why? Is there any specific justification for this, and how were these settings chosen?
- **Page 6**: *"Experiments show that in many settings (especially with large t and large $\eta_0$) the implementation does not converge."* Where? Everything seems to be converging in the results shown in this section. Is this demonstrated elsewhere in the paper that I'm missing?
- **Pages 7/8**: How many trials of these experiments were performed? Is there a reason why figures 5-7 don't include any measures of spread? Without some measure of spread it's hard to tell whether any of the differences are significant, which is problematic when these results are supposed to be verifying the theory presented.

## Minor Comments (which did not affect my score)
---
- **Page 2**: *"one needs to do the meta-training for an optimizer that runs for enough number of steps"* this reads awkwardly.
- **Page 2**: *"Another challenge is about the generalization performance of the learned optimizer"* This might read more clearly by rephrasing as "The generalization performance of the learned optimizer is another challenge"
- **Page 13**: *"where the second inequality holds"* I think this is supposed to read equality rather than inequality.

## Potentially Useful Citations
---
The following citations investigate methods which adapt a *vector* of step-sizes, which might be of interest
- **Incremental Delta-bar Delta (IDBD)**: Sutton, Richard S. "Adapting bias by gradient descent: An incremental version of delta-bar-delta." AAAI. 1992.
- **Stochastic Meta-descent (Generalization of IDBD)**: Schraudolph, Nicol N. "Local gain adaptation in stochastic gradient descent." (1999): 569-574.
- **TIDBD (IDBD for Reinforcement Learning algorithms)**:
  Kearney, Alex, et al. "Tidbd: Adapting temporal-difference step-sizes through stochastic meta-descent." arXiv preprint arXiv:1804.03334 (2018).

  Günther, Johannes, et al. "Meta-learning for Predictive Knowledge Architectures: A Case Study Using TIDBD on a Sensor-rich Robotic Arm." Proceedings of the 18th International Conference on Autonomous Agents and MultiAgent Systems. 2019.
- **AdaGain (Meta-descent for learning stability)**: Jacobsen, Andrew, et al. "Meta-descent for online, continual prediction." Proceedings of the AAAI Conference on Artificial Intelligence. Vol. 33. 2019.

---

> ### Author Response · Authors · 2020-11-19
> **response**
>
> Thanks for your positive review! Here we answer some of your questions.
>
> “Truncate a sequence when the weight norm is too large”
> In our setting, the ground truth weight has a unit norm, so if the model weight norm becomes much larger than one, we know the inner training has diverged and the learning rate is too high. Therefore, we simply terminate the inner training and declare the meta loss is high. Setting the weight to be a large fixed vector is just one way to declare the loss is high. We choose this particular way for some convenience in the proof. We have added more explanations in the revised version.
>
> “Where is the figure that shows the non-convergence of TensorFlow implementation?”
> In Figure 1, under TensorFlow implementation, the step size remains at initialization (which is non-optimal) through the meta training process. This happens because the meta gradient explodes and gives NaN value which fails the meta training. We have clarified this in the revised version.
>
> “How many trials of these experiments? Any measure of spread?”
> In the last paragraph of Appendix F.1, we mentioned that "we run 5 independent tests and collect training accuracy and test accuracy for evaluation. The plots show the mean of the 5 tests". We didn't show the measure of spread because these 5 curves are so close to each other, such that the range or standard deviation marks will not be readable in the plot. We have explained this in the new version.
>
> We have also fixed the phrasing issues and added the missing citations. Thanks for pointing them out.

---

> > ### Comment · AnonReviewer1 · 2020-11-23
> > **Follow-up**
> >
> > Thanks for the response; you've clarified most of my concerns. Based on the other reviewer's responses, I've decided to raise my score, to underscore the fact that I think that this paper is a worthy contribution.
> >
> > It seems to me that a lot of the pushback on this paper is due to the setting being too limited, but I would argue that this is exactly where the investigation into these guarantees *should* start. There's good reason why meta-descent has been around for so long without any rigorous guarantees: it's really quite difficult to produce guarantees for these algorithms. I think that the work provided in the appendix of this paper is a valid contribution to this line of work as it will provide other researchers a reference and set of tools moving forward, so that eventually we can tackle guarantees for more difficult objectives. It seems unreasonable to expect one of the first proper analyses of these methods to jump to general classes of loss functions right out of the gate.

---

### Official Review · AnonReviewer3 · 2020-10-28
**metalearning on linear regression**

**Rating:** 4
**Confidence:** 3

**Review:**

This paper considers algorithms that attempt to learn learning rates for gradient descent by gradient descent. Analysis is provided for a few specific quadratic losses showing that the gradient with respect to the learning rate may explode or vanish, and taking the logarithm is suggested to mitigate this. Further results suggest that implementing the gradient of the log comes with interesting numerical difficulties as *intermediate results* might explode or vanish even if the final answer does not.

Next, linear regression problems are analyzed in both the over-determined and under-determined settings. It shown that in the under-determined setting the optimal learning rate when tuned on the training set is very far from the optimal learning rate when tuned on the validation set.

Experiments are presented validating these theoretical findings, as well as comparisons to manual tuning on MNIST.


I felt that the claims in the abstract were a bit overblown here. I would not say that there has been a characterization of when to use validation vs train set, or that the proposed logarithmic method necessarily avoids vanishing/exploding metagradients. Instead, these questions have been addressed in the very specific setting of linear regression.

Theorems 5 and 6 appear to be making statements about the minimizers of the meta-objectives. Is there any guarantee that the meta-descent will efficiently approximate these minimizers so that the final algorithm actually comes with a guarantee?

Overall, I was a bit underwhelmed by the linear regression setting. This setting seems a bit limited since it is almost possible to write in closed form what the sgd iterates will do. I am willing to concede that there is some significant difficulty here, or that this setting is somehow necessarily broadly relevant, but I don’t think it has been clearly discussed. As for the empirical comparison, MNIST is frankly also a kind of toy dataset at this point, and even in this setting it does not seem that any gain is to be had over simply tuning SGD via a logarithmic grid.

---

> ### Author Response · Authors · 2020-11-19
> **response**
>
> Thanks for your review and suggestions. We take this opportunity to clarify some of your concerns.
>
> “Specific setting of linear regression”
> We agree that studying more complicated tasks/optimizers are important future works but we believe the current work is an important first step. There was no previous theoretical analysis even for linear regression, and even this simple setting already requires non-trivial techniques to analyze (see the proofs in Appendix).
>
> “Meta optimization for linear regression”
> For the simple quadratic objective, we prove the convergence of meta gradient descent in Theorem 4. In Theorem 5 and Theorem 6, we characterize the generalization performance of the optimal step size under the meta objective but didn’t address the optimization of meta training. We believe the optimization can be proved using similar techniques as in Theorem 4. We discussed this in the newly added conclusion section.
>
> “Any gain over simply tuning SGD for MNIST?”
> Figure 6 does show that in the noisy setting, train by validation is better than tuned SGD in terms of test loss. In the noiseless setting (Figure 5 and Figure 7), we don’t see a gain of trainable optimizer over tuned SGD, probably because MNIST is a simple dataset and SGD works well enough in such a setting.

---

### Official Review · AnonReviewer4 · 2020-10-28
**A 4+ paper because of the interesting problem it tries to address**

**Rating:** 4
**Confidence:** 5

**Review:**

This paper presents novel theoretical results on learning a step size for vanilla GD and SGD by unrolling the optimization steps and back-propagating, taking into account the simple problem minimizing quadratic functions and mean-square errors. The authors could demonstrate the occurrence of already-detected phenomena for learned optimizers, such as gradient explosion/vanishing and over-fitting, in the particular studied case. A few experiments illustrate what the developed theory predicts.

To the best of my knowledge, the literature lacks theoretical guarantees for learned optimizers and results as those within this manuscript can shed light on new ways to develop a solid learned optimizer theory.

Theorems 1 and 2, seem central to motivate the work. Yet, they are very imprecise.

In Theorem 1:

"For tuning the step size of gradient descent on a quadratic objective, if the meta-objective is the
loss of the last iteration, then the meta-gradient can explode/vanish."

Comment: Can is a loose expression — since it is a theorem, can it be shown that there are situations where they actually explode/ vanish? I believe this is discussed in Section 3, then why not state clearly that in the case of quadratics, the gradient vanishes?

In Theorem 2:

"For a simple least squares problem in d dimensions, if the number of samples n is a constant fraction of d (e.g., d/2), and the samples have large noise, then the train-by-train approach performs much worse than train-by-validation"

Comment: what is much worse? How is this shown?

"On the other hand, when number of samples n is large, train-by-train can get close to error dσ2/
n, which is optimal."

Comment: what is optimal?"


Section 3: After Equation (2), "We first show that when the number of samples is small (in particular n < d) and the
noise is a large enough constant, train-by-train can be much worse than train-by-validation"

Comment: Too many imprecise terms.

Until Section 3, the discussion emphasizes the use of logarithm for the meta-objective. But from section 4 onward, I see that the regular cost is used for the meta-objective. I am not sure if I am missing something here?

Figure 2 is invoked to support ‘In Figure 2, we verify the observation from Metz et al. (2019) that the optimal step size depends on inner training length.’— While the trend shows difference with length, I was unsure if the difference is significant—a comparison of setting a constant step size would be more revealing.

I also feel many of the proofs have not been properly explained in terms of the steps used in derivations, for example the various inequalities/upper bounds For Figures showing Training and Test RMSE, I would strongly recommend use of relative RMSE,
that is relative to the true value, expressed perhaps in dB plot. Currently it is very difficult to gauge the differences in terms of absolute RMSE.

Question: What is the overall conclusion after the experiments?

Is it that use of a logarithm helps remove vanishing gradient problem? From the experiments it seems the emphasis is that given data should be used with a separate validation data set.

I do not think that the quadratic analysis necessarily makes it insignificant, since it does aid understanding. Nevertheless, I think its rather poorly written and structured and I was left trying to find out what is being emphasized. As a result, even this simple-case analysis seems unclear to me. The imprecise nature of the Theorems added more to the confusion.


Pros:
- There are not many theoretical works on learned optimizers, thus this can be interesting to provide new insights.

- The proofs seem to be correct and rely upon interesting mathematical properties, as far as I could follow.

Cons:

- It seems to be hard to generalize the proposed analysis to more intricate (and practical) problems, such as training MLPs as optimizers. If this work intends to serve as a basis for further investigations, the authors should highlight which ideas could be still used in more general cases.

- The results of Theorem 3 were partially discussed in Metz et al. (2019) (Sec. 2.3). The authors could acknowledge this fact in the text.

- In Section 3, c_{min} is assumed to be positive, but what does it mean? Can w_0 be orthogonal to some eigenvector u_i of H?

- There are many typos and colloquial statements are presented. Sentences as "it is OK to use", "but didn’t give any", should be avoided. Moreover there are 21 "it's" throughout the paper.

- The statement "Therefore in order to train neural networks, it is better to use train-by-validation." is too strong and was not proved.

- Why is SGD noisier in Fig. 7 than the other methods? Are TbT and TbV calculating a full-batch gradient? This would not be a fair comparison if SGD had only access to an estimate of this gradient.



Minor concerns:

- What did the authors mean by "constant fraction" in Theorem 2?

- I think the task definition should also depend on the initial condition w_0 as different initializations provide different trajectories.

- The population losses were defined but not used in the text. Maybe it should be defined only in the appendix for clarity purposes.

- What is the relevance of "L > \alpha" to show that F(\eta) is strictly convex in the proof of Theorem 3?

- The constants in Theorem 5 and 6 are unused and could be omitted for the sake of concision. I recommend defining then in the proof only.

---

> ### Author Response · Authors · 2020-11-19
> **response**
>
> Thanks for your detailed reviews and valuable suggestions! We clarify some of your concerns as follows.
>
> “Theorem 1 and Theorem 2 are imprecise”
> The statements in Theorem 1 and Theorem 2 are imprecise because they are informal theorems. The corresponding formal versions are Theorem 3,4,5,6. We have added pointers below the informal theorems to the corresponding formal ones to further avoid confusion.
>
> “Regular cost is used in section 3”
> For the convenience of analysis, we studied vanilla meta objective without log transformation in section 3. We believe our results can be extended to log-transformed meta objectives because compositing the meta objective with a log function does not change its minimizer. We discussed this in the revised version.
>
> “Relative RMSE, dB plot”
> Thanks for the recommendation! We will consider dividing the norm of the true values to better show the scale of the differences. For the current Figure 3 and Figure 4, we also show the error range through multiple trials of the experiment, so it can still show that the differences are consistent and significant.
>
> “Overall conclusion after the experiments”
> Our experiments verify our theory in previous sections. First, it’s observed that the intermediate step in the auto-differentiation can vanish/explode even if the meta gradient is well behaved. Second, in the least-squares and MNIST experiments, it’s observed that when the number of samples is small and the noise is large, the step size obtained by train by validation generalizes much better than that obtained by train by train. We have added a new conclusion section.
>
> “Can w_0 be orthogonal with some eigenvector of H”
> We only require w_0 to have a non-zero correlation with the top eigenvector and the bottom eigenvector, so w_0 can be orthogonal with other eigenvectors.
>
> “Nosier SGD”
> 1. The SGD curve in Fig. 7 seems noisy because we run significantly more steps in Fig. 7 (40000 steps) compared to the others (4000 steps), so there are 10 times more data points in Fig. 7, making it seem noisier. If we sample 1/10 out of it, the curve will look as smooth as the others. (We run more steps because we use all data in that experiment setting, so the inner model needs more steps to converge)
> 2. We use mini-batches of 32 samples in all of these experiments (TbT, TbV and SGD), which is mentioned on the last three lines on page 7.
>
> “Constant fraction”
> This means the number of samples n is between d/4 and 3d/4. See more details in Theorem 5.
>
> “Task definition depends on initialization”
> Yes, generally a task should also depend on the initialization. For simplicity, in our analysis, we have fixed the initialization to the same point across different tasks.
>
> “Relevance of L>alpha in proving strict convexity”
> We need L>alpha to ensure that the second derivative of the meta objective is always positive. See more details in the proof of Theorem 3 (page 12).
>
> Also thanks for the suggestions on writing, we have tried to improve it in the new version.

---

### Official Review · AnonReviewer2 · 2020-10-29
**Somewhat interesting, but too much hidden in the supplemental**

**Rating:** 4
**Confidence:** 4

**Review:**

The authors analyze two different approaches to "learning to learn" which correct various pathologies, however primarily restricted to the quadratic setting. The first is the use of the log objective rather than the raw objective for the inner-loop optimization step, and the second is the differences between train-by-train and train-by-validation. Overall, however, I feel that this work relies much too heavily on its supplemental material, as the proofs themselves are not well spelled out within the text itself, and at 68 pages total this is perhaps too extensive for this venue.

The authors give a reasonably good overview of recent approaches to to learning to learn, or meta-learning for optimizers. However, it is worth pointing out that they omit much of the early work in this area, particularly the work of Hochreiter et al., 2001 and the various citations therein, which give this area its name.

Notationally, the work is also very dense. The authors might consider whether it is of value in introducing their approach whether to do away with the focus on supervised (e.g. input/output x/y data) models and instead focus on noisy gradients evaluated at each iteration. Otherwise it might be helpful to be more explicit about the indices x_i -> y_i when introducing these models, although this is perhaps a very nit-picky and aesthetic consideration.

Theorem 3 is itself interesting, however it seems to follow quite directly from Section 2.3 of Metz et al. Although the authors do provide more detail, this is restricted to the quadratic inner-loop setting, whereas the description of Metz doesn't go so far due to the more general problem setting (and changing Hessians).

Theorem 4, similarly is of interest. But it's not clear how much practical value it has. As far as I can tell the authors do not use this log objective when discussing the train-by-train/vs/train-by-validation setting. I may have misunderstood, but if this is not the case why not? Similarly, how does this compare empirically to the solution proposed by Metz et al? And lastly, the authors did not describe in detail how they dealt with the intermediate gradients which would have been most useful.

Finally, the remaining theorems are interesting, but mostly seem to confirm the results of Metz at al, and are restricted to the quadratic setting. So more work might be needed to show why this is particularly of use.

---

> ### Author Response · Authors · 2020-11-19
> **response**
>
> Thanks for your review and suggestions!
>
> “Proofs are not well spelled out within the text”
> Due to the space limit, the proof is left in the appendix. We will add a more detailed proof sketch in the main paper.
>
> “Theorem 4 is restricted to the quadratic inner-loop setting”
> Note that there was no previous theoretical analysis even for this simple setting, and even this simple quadratic setting already requires non-trivial techniques to analyze (see the proofs in Appendix A). We agree that studying more complicated tasks/optimizers are important future works but we believe the current work is an important first step.
>
> “Log objective is not used in the linear regression setting”
> For the convenience of analysis, we studied vanilla meta objective without log transformation in Theorem 5,6. We believe our results can be extended to log-transformed meta objectives because compositing with a log function does not change the optimal step size. We clarified this point in the new version.
>
> “How to deal with the intermediate gradient?”
> To avoid the intermediate gradient vanishing/explosion issue, we compute the meta objective using a formula which we derive in Appendix A. We added more explanation in the main text.
>
> Thanks for pointing out the missing citations, we have added them in the revised version.

---

### Decision · Program_Chairs · 2021-01-07
**Final Decision**

**Decision:**

Reject

**Comment:**

The paper studies the problem of learning the step size of gradient descent for quadratic loss. Interesting theoretical results are presented, which formally support the empirically observed problems of exploding/vanishing gradients, as well as another result showing that if meta-learning is done based on the validation performance, optimal performance can be achieved for a simple linear regression task.

On the negative side, there are several issues which preclude publication of the paper in its current stage:

1. The claims in the text seem to be much stronger than what is actually proved.
2. The contributions are not properly connected to the literature (e.g., the relation to Metz et al. 2013 is not properly discussed).
3. Not mentioned in the reviews, but the paper does not explore the connection to similar results coming from online learning/sequential optimization. Recently there has been a surge of papers analyzing meta-learning from an online learning perspective; as an example,  Khodak et al. (2019) presents an adaptive step-size tuning with guarantees for a much more general problem setting. It could also be interesting to explore if the exploding gradient problem is also related to issues with mirror descent as described in Section 4.1 of Orabona and Pal (2018).
4. The presentation in the main text does not provide enough insight about the results, as too much material is relegated to the appendix.
5. The presentation is often imprecise; it is somewhat questionable (though it is a matter of taste) if the informal theorems are useful (why call them theorems?), but Corollary 1 is not indicated to be informal, yet it is hard to interpret formally. There are other issues such as the statements of Theorems 5 and 6 where conditional expectations are used without explicitly showing the conditions, high probability bounds are stated although the error probability never appears, etc.
6. It is not clear how meta-learning helps in Theorem 6 compared to methods adaptively tuning the step size (as a recent work, see, e.g., Joulani et al. 2020 and the references therein).



M. Khodak, M-F. Balcan, A. Talwalkar. Adaptive Gradient-Based Meta-Learning Methods. NeurIPS 2019.
F. Orabona, D. Pal. Scale-free online learning. Theoretical Computer Science 716, 50-69, 2018.
P. Joulani, A. Raj, A. Gyorgy, C. Szepesvari. A simpler approach to accelerated optimization: iterative averaging meets optimism. ICML 2020.